# Finite-Time Regret of Thompson Sampling Algorithms for Exponential Family Multi-Armed Bandits

**Tianyuan Jin**
National University of Singapore
tianyuan@u.nus.edu

**Pan Xu**
Duke University
pan.xu@duke.edu

**Xiaokui Xiao**
National University of Singapore
xkxiao@nus.edu.sg

**Anima Anandkumar**
California Institute of Technology
anima@caltech.edu

## Abstract

We study the regret of Thompson sampling (TS) algorithms for exponential family bandits, where the reward distribution is from a one-dimensional exponential family, which covers many common reward distributions including Bernoulli, Gaussian, Gamma, Exponential, etc. We propose a Thompson sampling algorithm, termed ExpTS, which uses a novel sampling distribution to avoid the under-estimation of the optimal arm. We provide a tight regret analysis for ExpTS, which simultaneously yields both the finite-time regret bound as well as the asymptotic regret bound. In particular, for a $K$-armed bandit with exponential family rewards, ExpTS over a horizon $T$ is sub-UCB (a strong criterion for the finite-time regret that is problem-dependent), minimax optimal up to a factor $\sqrt{\log K}$, and asymptotically optimal, for exponential family rewards. Moreover, we propose ExpTS$^+$, by adding a greedy exploitation step in addition to the sampling distribution used in ExpTS, to avoid the over-estimation of sub-optimal arms. ExpTS$^+$ is an anytime bandit algorithm and achieves the minimax optimality and asymptotic optimality simultaneously for exponential family reward distributions. Our proof techniques are general and conceptually simple and can be easily applied to analyze standard Thompson sampling with specific reward distributions.

## 1  Introduction

The Multi-Armed Bandit (MAB) problem is centered around a fundamental model for balancing the exploration versus exploitation trade-off in many online decision problems. In this problem, the agent is given an environment with a set of $K$ arms $[K] = \{1, 2, \cdots, K\}$. At each time step $t$, the agent pulls an arm $A_t \in [K]$ based on observations of previous $t - 1$ time steps, and then a reward $r_t$ is revealed at the end of the step. In real-world applications, reward distributions often have different forms such as Bernoulli, Gaussian, etc. As suggested by Auer et al. [8, 9], Agrawal and Goyal [5], Lattimore [29], Garivier et al. [16], a good bandit strategy should be general enough to cover a sufficiently rich family of reward distributions. In this paper, we assume the reward $r_t$ is independently generated from some canonical one-parameter exponential family of distributions with a mean value $\mu_{A_t}$. It is a rich family that covers many common distributions including Bernoulli, Gaussian, Gamma, Exponential, and others.

The goal of a bandit strategy is usually to maximize the cumulative reward over $T$ time steps, which is equivalent to minimizing the regret, defined as the expected cumulative difference between playing

36th Conference on Neural Information Processing Systems (NeurIPS 2022).

the best arm and playing the arm according to the strategy: $R_\mu(T) = T \cdot \max_{i \in [K]} \mu_i - \mathbb{E}[\sum_{t=1}^{T} r_t]$. We assume, without loss of generality, $\mu_1 = \max_{i \in [K]} \mu_i$ is the best arm throughout this paper. For a fixed bandit instance (i.e., mean rewards $\mu_1, \cdots, \mu_K$ are fixed), Lai and Robbins [26] shows that for distributions that are continuously parameterized by their means,

$$\lim_{T \to \infty} \frac{R_\mu(T)}{\log T} \geq \sum_{i>1} \frac{\mu_1 - \mu_i}{\mathrm{kl}(\mu_i, \mu_1)}, \tag{1.1}$$

where $\mathrm{kl}(\mu_i, \mu_1)$ is the Kullback-Leibler divergence between two distributions with mean $\mu_i$ and $\mu_1$. A bandit strategy satisfying $\lim_{T \to \infty} R_\mu(T)/\log T = \sum_{i>1} \frac{\mu_1 - \mu_i}{\mathrm{kl}(\mu_i, \mu_1)}$ is said to be *asymptotically optimal* or achieve the asymptotic optimality in regret. The asymptotic optimality is one of the most important statistical properties in regret minimization, which shows that an algorithm is consistently good when it is played for infinite steps and thus should be a basic theoretical requirement of any good bandit strategy [8].

In practice, we can only run the bandit algorithm for a finite number $T$ steps, which is the time horizon of interest in real-world applications. Therefore, the finite-time regret is the ultimate property of a practical bandit strategy in regret minimization problems. A strong notion of finite-time regret bounds is called *the sub-UCB criteria* [29]. An algorithm is sub-UCB if there exist universal constants $C_1, C_2 > 0$ such that for any problem instances,

$$R_\mu(T) = C_1 \sum_{i \in [K]: \Delta_i > 0} \Delta_i + C_2 \sum_{i \in [K]: \Delta_i > 0} \frac{\log T}{\Delta_i}, \tag{1.2}$$

where $\Delta_i = \mu_1 - \mu_i$ is the sub-optimal gap between arm 1 and arm $i$. Note that the regret bound in (1.2) is a problem-dependent bound since it depends on the bandit instance and the sub-optimal gaps. Sub-UCB is an important metric for finite-time regret bound and has been adopted by recent work of Lattimore [29], Bian and Jun [10]. Another special type of finite-time bounds is called the worst-case regret, which is defined as the finite-time regret of an algorithm on any possible bandit instance within a bandit class. Specifically, for a finite time horizon $T$, Auer et al. [8] proves that any strategy has at least worst-case regret $\Omega(\sqrt{KT})$ for a $K$-armed bandit. We say the strategy that achieves a worst-case regret $O(\sqrt{KT})$ is *minimax optimal* or achieves the minimax optimality. Different from the asymptotic optimality, the minimax optimality characterizes the worst-case performance of the bandit strategy in finite steps.

A vast body of literature in multi-armed bandits [6, 5, 22, 32, 16, 29] have been pursing the afore-mentioned theoretical properties of bandit algorithms: generality, asymptotic optimality, problem-dependent finite-time regret, and minimax optimality. However, most of them focus on one or two properties and sacrifice the others. Moreover, many of existing theoretical analyses of bandit strategies are for optimism-based algorithm. The theoretical analysis of Thompson sampling (TS) is much less understood until recently, which has been shown to exhibit superior practical performances compared to the state-of-the-art methods [13, 34]. Specifically, its finite-time regret, asymptotic optimality, and near minimax optimality have been studied by Agrawal and Goyal [3, 4, 5] for Bernoulli rewards. Jin et al. [20] proved the minimax optimality of TS for sub-Gaussian rewards. For exponential family reward distributions, the asymptotic optimality is shown by Korda et al. [25], but no finite-time regret of TS is provided. See Table 1 for a comprehensive comparison of these results.

In this paper, we study the regret of Thompson sampling for exponential family reward distributions and address all the theoretical properties of TS. We propose a variant of TS algorithm with a general sampling distribution and a tight analysis for frequentist regret bounds. Our analysis simultaneously yields both the finite-time regret bound and the asymptotic regret bound.
Specifically, the **main contributions** of this paper are summarized as follows:

- We propose ExpTS, a general variant of Thompson sampling, that uses a novel sampling distribution with a tight anti-concentration bound to avoid the under-estimation of the optimal arm and a tight concentration bound to avoid the over-estimation of sub-optimal arms. For exponential family of reward distributions, we prove that ExpTS is the first Thompson sampling algorithm achieving the sub-UCB criteria, which is a strong notion of problem-dependent finite-time regret bounds. We further show that ExpTS is also simultaneously minimax optimal up to a factor of $\sqrt{\log K}$, as well as asymptotically optimal, where $K$ is the number of arms.
- We also propose ExpTS$^+$, which explores between the sample generated in ExpTS and the empirical mean reward for each arm, to get rid of the extra $\sqrt{\log K}$ factor in the worst-case regret.

Table 1: Comparisons of different Thompson sampling algorithms on $K$-armed bandits over a horizon $T$. For any algorithm, *Asym. Opt* is the indicator whether it is asymptotically optimal, *minimax ratio* is the scaling of its worst-case regret w.r.t. the minimax optimal regret $O(\sqrt{VKT})$, where $V$ is the variance of reward distributions, and sub-UCB is the indicator if it satisfies the sub-UCB criteria.

| Algorithm | Reward Type | Asym. Opt | Finite-Time Regret | | Anytime | Reference |
|---|---|---|---|---|---|---|
| | | | Minimax Ratio | Sub-UCB | | |
| TS | Bernoulli | yes | $\sqrt{\log T}$ | $-^*$ | yes | [4] |
| TS | Bernoulli | – | $\sqrt{\log K}$ | $-^*$ | yes | [5] |
| TS | Exponential Family | yes | – | – | yes | [25] |
| MOTS | sub-Gaussian | no | 1 | no | no | [20] |
| MOTS-$\mathcal{J}$ | Gaussian | yes | 1 | no | no | [20] |
| ExpTS | Exponential Family | yes | $\sqrt{\log K}$ | yes | yes | This paper |
| ExpTS$^+$ | Exponential Family | yes | 1 | no | yes | This paper |

*[4, 5] did not explicitly show that their regret bounds are sub-UCB. However, the intermediate results in their proofs might imply sub-UCB regret bounds.

Thus ExpTS$^+$ is the first Thompson sampling algorithm that is simultaneously minimax and asymptotically optimal for exponential family of reward distributions.

- Our regret analysis of ExpTS can be easily extended to analyze standard Thompson sampling with common reward distributions. We prove that standard Thompson sampling without inflating the posterior distribution[1] is minimax optimal up to a factor of $\sqrt{\log K}$, which matches the regret lower bound for standard Thompson sampling in Agrawal and Goyal [5]. Similar to the idea of ExpTS$^+$, we can add a greedy exploration step to the posterior distributions used in these variants of TS, and then the algorithms are simultaneously minimax and asymptotically optimal.

Our techniques are novel and conceptually simple. First, we introduce a lower confidence bound in the regret decomposition to avoid the under-estimation of the optimal arm, which is important in obtaining the finite-time regret bound. Specifically, Jin et al. [20] (Lemma 5 in their paper) shows that for Gaussian reward distributions, Gaussian-TS has a regret bound at least in the order of $\Omega(\sqrt{KT \log T})$ if the standard regret decomposition in existing analysis of Thompson sampling [5, 30, 20] is adopted. With our new regret decomposition that is conditioned on the lower confidence bound introduced in this paper, we improve the worst-case regret of Gaussian-TS for Gaussian reward distributions to $O(\sqrt{KT \log K})$.

Second, we do not require the closed form of the reward distribution, but only make use of the corresponding concentration bounds. This means our results can be readily extended to other reward distributions. For example, we can extend ExpTS$^+$ to sub-Gaussian reward distributions and the algorithm is simultaneously minimax and asymptotically optimal[2], which improve the results of MOTS proposed by Jin et al. [20] (see Table 1).

Third, the idea of ExpTS$^+$ is simple and can be used to remove the extra $\sqrt{\log K}$ factor in the worst-case regret. We note that MOTS [20] can also achieve the minimax optimal via the clipped Gaussian. However, it is not clear how to generalize the clipping idea to the exponential family of reward distribution. Moreover, it uses the MOSS [6] index for clipping, which needs to know the horizon $T$ in advance and thus cannot be extended to the anytime setting, while ExpTS$^+$ is an anytime bandit algorithm which does not need to know the horizon length in advance.

**Notations.** We let $T$ be the total number of time steps, $K$ be the total number of arms, and $[K] = \{1, 2, \cdots, K\}$. For simplicity, we assume arm 1 is the optimal throughout this paper, i.e., $\mu_1 = \max_{i \in [K]} \mu_i$. We denote $\log^+(x) = \max\{0, \log x\}$ and $\Delta_i := \mu_1 - \mu_i$, $i \in [K] \setminus \{1\}$ for the

---

[1]This is a common trick in the literature. In particular, for Bernoulli rewards, instead of using Beta posterior, Agrawal and Goyal [5] consider Thompson sampling with Gaussian posterior, whose variance is larger. Moreover, Jin et al. [20] inflate the variance of the sampling distribution by a factor $1/\rho$, where $\rho < 1$. However, both of these methods lose the asymptotic optimality.

[2]Note that sub-Gaussian is a non-parametric family and thus the lower bound (1.1) by Lai and Robbins [26] does not directly apply to a general sub-Gaussian distribution. Following similar work in the literature [20], in this paper, when we say an algorithm achieves the asymptotic optimality for sub-Gaussian rewards, we mean its regret matches the asymptotic lower bound for Gaussian rewards, which is a stronger notion.

gap between arm 1 and arm $i$. We let $T_i(t) := \sum_{j=1}^{t} \mathbb{1}\{A_t = i\}$ be the number of pulls of arm $i$ at the time step $t$, $\widehat{\mu}_i(t) := 1/T_i(t) \sum_{j=1}^{t} \left[ r_j \cdot \mathbb{1}\{A_t = i\} \right]$ be the average reward of arm $i$ at the time step $t$, and $\widehat{\mu}_{is}$ be the average reward of arm $i$ after its $s$-th pull. We reserve notations $C_1, C_2, \cdots$ to be positive universal constants that are independent of problem parameters.

## 2 Related Work

There are series of works pursuing the asymptotic regret bound and worst-case regret bound for MAB. For asymptotic optimality, UCB algorithms [15, 31, 5, 29], Thompson sampling [23, 25, 5, 20], Bayes-UCB [22], and other methods [21, 10] are all shown to be asymptotically optimal. Among them, only a few [15, 12, 25] can be extended to exponential families of distributions. One notable result in Cappé et al. [12] shows that for $[0, 1]$ bounded distribution, there exists an algorithm that has regret $\sum_{i>1} \Delta_i \log T/\mathrm{kl}(\mu_i, \mu_1) + O(\sum_{i>1}(\log T)^{4/5} \log \log T \cdot \Delta_i)^3$, which is better than (1.2). It is an interesting problem whether we can achieve such regret for unbounded reward distributions. For the worst-case regret, MOSS [6] is the first algorithm proved to be minimax optimal. Later, KL-UCB$^{++}$ [5], AdaUCB [29], MOTS [20] also join the family. The anytime version of MOSS is studied by Degenne and Perchet [14]. There are also some works that focus on the near optimal problem-dependent regret bound [27, 28]. As far as we know, no algorithm has been proved to achieve the sub-UCB criteria, asymptotic optimality, and minimax optimality simultaneously for exponential family reward distributions.

For Thompson sampling, Russo and Van Roy [33] studied the Bayesian regret. They show that the Bayesian regret of Thompson sampling is never worse than the regret of UCB. Bubeck and Liu [11] further showed the Bayesian regret of Thompson sampling is optimal using the regret analysis of MOSS. There are also a line of works focused on the frequentist regret of TS. Agrawal and Goyal [3] proposed the first finite time regret analysis for TS. Kaufmann et al. [23], Agrawal and Goyal [4] proved that TS with Beta posteriors is asymptotically optimal for Bernoulli reward distributions. Korda et al. [25] extended the asymptotic optimality to the exponential family of reward distributions. Subsequently, for Bernoulli rewards, Agrawal and Goyal [5] proved that TS with Beta prior is asymptotically optimal and has worst-case regret $O(\sqrt{KT \log T})$. Besides, they showed that TS with Gaussian posteriors can achieve a better worst-case regret bound $O(\sqrt{KT \log K})$. They also proved that for Bernoulli rewards, TS with Gaussian posteriors has a worst-case regret at least $\Omega(\sqrt{KT \log K})$. Very recently, Jin et al. [20] proposed the MOTS algorithm that can achieve the minimax optimal regret $O(\sqrt{KT})$ for multi-armed bandits with sub-Gaussian rewards but at the cost of losing the asymptotic optimality by a multiplicative factor of $1/\rho$, where $0 < \rho < 1$ is an arbitrarily fixed constant. For bandits with Gaussian rewards, Jin et al. [20] proved that MOTS combined with a Rayleigh distribution can achieve the minimax optimality and the asymptotic optimality simultaneously. We refer readers to Tables 1 and 2 for more details.

## 3 Preliminary on Exponential Family Distributions

A one-dimensional canonical exponential family [15, 17, 32] is a parametric set of probability distributions with respect to some reference measure, with the density function given by

$$p_\theta(x) = \exp(x\theta - b(\theta) + c(x)),$$

where $\theta$ is the model parameter, and $c$ is a real function. Denote the measure of $p_\theta(x)$ as $\nu_\theta$. Then, the above definition can be rewritten as

$$\frac{\mathrm{d}\nu_\theta}{\mathrm{d}\rho}(x) = \exp(x\theta - b(\theta)),$$

for some measure $\rho$ and $b(\theta) = \log(\int e^{x\theta} \mathrm{d}\rho(x))$. We make the classic assumption used by Garivier and Cappé [15], Ménard and Garivier [32] that $b(\theta)$ is twice differentiable with a continuous second derivative. Then, we can verify that exponential families have the following properties:

$$b'(\theta) = \mathbb{E}[\nu_\theta] \qquad \text{and} \qquad b''(\theta) = \mathrm{Var}[\nu_\theta] > 0.$$

---

[3]Cappé et al. [12] used a more general notation $\mathcal{K}_{\inf}(\cdot, \cdot)$ for any distribution supported in $[0, 1]$, which is equivalent to the $\mathrm{kl}(\cdot, \cdot)$ notation for the one-exponential family distribution studied in our paper.

Let $\mu = \mathbb{E}[\nu_\theta]$. The above equality means that the mapping between the mean value $\mu$ of $\nu(\theta)$ and the parameter $\theta$ is one-to-one. Hence, exponential family of distributions can also be parameterized by the mean value $\mu = b'(\theta)$. Note that $b''(\theta) > 0$ for all $\theta$, which implies $b'(\cdot)$ is invertible and its inverse function $b'^{-1}$ satisfies $\theta = b'^{-1}(\mu)$. In this paper, we will use the notion of Kullback-Leibler (KL) divergence. The KL divergence between two exponential family distributions with parameter $\theta$ and $\theta'$ respectively is defined as follows:

$$\mathrm{KL}(\nu_\theta, \nu_{\theta'}) = b(\theta') - b(\theta) - b'(\theta)(\theta' - \theta). \tag{3.1}$$

Recall that the mapping $\theta \mapsto \mu$ is one-to-one. We can define an equivalent notion of the KL divergence between random variables $\nu_\theta$ and $\nu_{\theta'}$ as a function of the mean values $\mu$ and $\mu'$ respectively:

$$\mathrm{kl}(\mu, \mu') = \mathrm{KL}(\nu_\theta, \nu_{\theta'}),$$

where $\mathbb{E}[\nu_\theta] = \mu$ and $\mathbb{E}[\nu_{\theta'}] = \mu'$. Similarly, we define $V(\mu) = \mathrm{Var}(\nu_{b'^{-1}(\mu)})$ as the variance of an exponential family random variable $\nu_\theta$ with mean $\mu$. We assume the variances of exponential family distributions used in this paper are bounded by a constant $V > 0$: $0 < V(\mu) \le V < +\infty$. We have the following property of the KL divergence between exponential family distributions.

**Proposition 3.1** (Harremoës [17]). *Let $\mu$ and $\mu'$ be the mean values of two exponential family distributions. The Kullback-Leibler divergence between them can be calculated as follows:*

$$kl(\mu, \mu') = \int_\mu^{\mu'} \frac{x - \mu}{V(x)} dx. \tag{3.2}$$

Based on Proposition 3.1, we can also verify the following properties.

**Proposition 3.2** (Jin et al. [19]). *For all $\mu$ and $\mu'$, we have*

$$kl(\mu, \mu') \ge (\mu - \mu')^2/(2V). \tag{3.3}$$

*In addition, for $\epsilon > 0$ and $\mu \le \mu' - \epsilon$, we can obtain that*

$$kl(\mu, \mu') \ge kl(\mu, \mu' - \epsilon) \quad and \quad kl(\mu, \mu') \le kl(\mu - \epsilon, \mu'). \tag{3.4}$$

Exponential families cover many of the most common distributions used in practice such as Bernoulli, exponential, Gamma, and Gaussian distributions. In particular, for two Gaussian distributions with the same known variance $\sigma^2$ but different means $\mu$ and $\mu'$, we can choose $V(\cdot) = \sigma^2$, and it holds that $\mathrm{kl}(\mu, \mu') = (\mu - \mu')^2/(2\sigma^2)$. For two Bernoulli distributions with means $\mu$ and $\mu'$ respectively, the variance upper bound is set as $V = 1/4$. Thus we can recover the result in Proposition 3.1 as $\mathrm{kl}(\mu, \mu') = \mu \log(\mu/\mu') + (1-\mu)\log((1-\mu)/(1-\mu'))$. For exponential and Gamma distributions, it suffices to ensure the variance id bounded as long as we assume the mean value is bounded.

The definition of one-dimensional exponential family in our paper is $p_\theta(x) = \exp(x\theta - b(\theta) + c(x))$, which is the same as that used by Garivier and Cappé [15], Harremoës [17], Jin et al. [19], Ménard and Garivier [32] as well as Cappé et al. [12]. The one-dimensional exponential family considered in Korda et al. [25] ($p_\theta(x) = \exp(T(x)\theta - b(\theta) + c(x))$) is more general than that in the aforementioned papers (see page 4 in Korda et al. [25]). Lai and Robbins [26] considers parametric distributions that satisfies some mild conditions, which is also more general than ours. Moreover, Cappé et al. [12] also considered the general reward distributions supported in $[0, 1]$, which is not compatible to ours.

## 4    Thompson Sampling for Exponential Family Reward Distributions

We present a general variant of Thompson sampling for exponential family rewards in Algorithm 1, named as ExpTS. At round $t$, ExpTS maintains an estimate of a sampling distribution for each arm, denoted as $\mathcal{P}$. The algorithm generates a sample parameter $\theta_i(t)$ for each arm $i$ independently from their sampling distribution and chooses the arm that attains the largest sample parameter. For each arm $i \in [K]$, the sampling distribution $\mathcal{P}$ is usually defined as a function of the total number of pulls $T_i(t)$ and the empirical average reward $\widehat{\mu}_i(t)$. After pulling the chosen arm, the algorithm updates $T_i(t)$ and $\widehat{\mu}_i(t)$ for each arm based on the reward $r_t$ it receives and proceeds to the next round.

Since we study the frequentist regret bound of Algorithm 1, ExpTS is not restricted as a Bayesian method. It has been shown [1, 20, 24, 36] that the sampling distribution does not have to be a posterior distribution derived from a pre-defined prior distribution. Therefore, we call $\mathcal{P}$ the sampling distribution instead of the posterior distribution as in Bayesian regret analysis of Thompson sampling [33, 11]. To obtain the finite-time regret bound of ExpTS for exponential family rewards, we will discuss the choice of a general sampling distribution and a new proof technique.

---
**Algorithm 1** Exponential Family Thompson Sampling (ExpTS)
---
1: **Input:** Arm set $[K]$
2: **Initialization:** Play each arm once and set $T_i(K) = 1$; let $\widehat{\mu}_i(K)$ be the observed reward of playing arm $i$
3: **for** $t = K + 1, K + 2, \cdots$ **do**
4:     For all $i \in [K]$, sample $\theta_i(t)$ independently from $\mathcal{P}(\widehat{\mu}_i(t), T_i(t))$
5:     Play arm $A_t = \arg\max_{i \in [K]} \theta_i(t)$ and observe the reward $r_t$
6:     For all $i \in [K]$, update the mean reward estimator and the number of pulls:

$$\hat{\mu}_i(t) = \frac{T_i(t-1) \cdot \widehat{\mu}_i(t-1) + r_t \mathbb{1}\{i = A_t\}}{T_i(t-1) + \mathbb{1}\{i = A_t\}}, \quad T_i(t) = T_i(t-1) + \mathbb{1}\{i = A_t\}$$

7: **end for**
---

### 4.1 Challenges in Regret Analysis for Exponential Family Bandits

Before we choose a specific sampling distribution $\mathcal{P}$ for ExpTS, we first discuss the main challenges in the finite-time regret analysis of Thompson sampling, which is the main motivation for our design of $\mathcal{P}$ in the next subsection.

**Under-Estimation of the Optimal Arm.** Denote $\widehat{\mu}_{is}$ as the average reward of arm $i$ after its $s$-th pull, $T_i(t)$ as the number of pulls of arm $i$ at time $t$, and $\mathcal{P}(\widehat{\mu}_{is}, s)$ as the sampling distribution of arm $i$ after its $s$-th pull. The regret of the algorithm contributed by pulling arm $i$ is $\Delta_i \mathbb{E}[T_i(T)]$, where $T_i(T)$ is the total number of pulls of arm $i$. All existing analyses of finite-time regret bounds for TS [3–5, 20] decompose this regret term as $\Delta_i \mathbb{E}[T_i(T)] \leq D_i + h_i(\Delta_i, T, \theta_i(1), \ldots, \theta_i(T))$, where $h_i()$ is a quantity characterizing the over-estimation of arm $i$ which can be easily dealt with by some concentration properties of the sampling distribution (see Lemma A.3 for more details). The term $D_i$ characterizes the under-estimation of the optimal arm 1, which is usually bounded as follows.

$$D_i = \Delta_i \sum_{s=1}^{T} \mathbb{E}_{\widehat{\mu}_{1s}} \left[ \frac{1}{G_{1s}(\epsilon)} - 1 \right], \tag{4.1}$$

where $G_{1s}(\epsilon) = 1 - F_{1s}(\mu_1 - \epsilon)$, $F_{1s}$ is the CDF of the sampling distribution $\mathcal{P}(\widehat{\mu}_{1s}, s)$, and $\epsilon = \Theta(\Delta_i)$. In other words, $G_{1s}(\epsilon) = \mathbb{P}(\theta_1(t) > \mu_1 - \epsilon)$ is the probability that the best arm will *not be under-estimated* from the mean reward by a margin $\epsilon$. Furthermore, we can interpret the quantity in (4.1) as the result of a union bound indicating how many samples TS requires to ensure that at least one sample of the best arm $\{\theta_1(t)\}_{t=1}^{T}$ is larger than $\mu_1 - \epsilon$. If $G_{1s}(\epsilon)$ is too small, arm 1 could be significantly under-estimated, and thus $D_i$ will be unbounded. In fact, as shown in Lemma 5 by Jin et al. [20], for MAB with Gaussian rewards, TS using Gaussian posteriors will unavoidably suffer from the lower bound of $K \cdot D_i = \Omega(\sqrt{KT \log T})$.

To address the above issue, we introduce a lower confidence bound for measuring the under-estimation problem. We use a new decomposition of the regret that bounds $D_i$ with the following term

$$\Delta_i \sum_{s=1}^{T} \mathbb{E}_{\widehat{\mu}_{1s}} \left[ \left( \frac{1}{G_{1s}(\epsilon)} - 1 \right) \cdot \mathbb{1}\{\widehat{\mu}_{1s} \geq Low_s\} \right], \tag{4.2}$$

where $Low_s$ is a lower confidence bound of $\widehat{\mu}_{1s}$. Intuitively, due to the concentration of arm 1's rewards, the probability of $\widehat{\mu}_{1s} \leq Low_s$ is very small. Thus, even when $G_{1s}(\epsilon)$ is small, the overall regret can be well controlled.

In the regret analysis of TS, we can bound (4.2) from two facets: (1) the lower confidence bound can be proved using the concentration property of the reward distribution; and (2) the term $G_{1s}(\epsilon) = \mathbb{P}(\theta_1(t) > \mu_1 - \epsilon)$ can be upper bounded by the anti-concentration property for the sampling distribution $\mathcal{P}$. To achieve an optimal regret, one needs to carefully balance the interplay between these two bounds. For a specific reward distribution (e.g., Gaussian, Bernoulli) as is studied by Agrawal and Goyal [5], Jin et al. [20], there are already tight anti-concentration inequalities for the reward distribution, and thus the lower confidence bound is tight. Therefore, by choosing Gaussian or Bernoulli as the prior (which leads to a Gaussian or Beta sampling distribution $\mathcal{P}$), we can use existing anti-concentration bounds for Gaussian [2, Formula 7.1.13] or Beta [18, Prop. A.4] distributions to obtain a tight bound of $G_{1s}(\epsilon)$.

In this paper, we study the general exponential family of reward distributions, which has no closed form. Thus we cannot obtain a tight concentration bound for $\hat{\mu}_{1s}$ as in special cases such as Gaussian or Bernoulli rewards. This increases the hardness of tightly bounding term (4.2) and it is imperative for us to design a sampling distribution $\mathcal{P}$ with a tight anti-concentration bound that can carefully control $G_{1s}(\epsilon)$ without any knowledge of the closed form distribution of the average reward $\hat{\mu}_{1s}$. Due to the generality of exponential family distributions, it is challenging and nontrivial to find such a sampling distribution to obtain a tight finite-time regret bound.

## 4.2 Sampling Distribution Design in Exponential Family Bandits

In this subsection, we show how to choose a sampling distribution $\mathcal{P}$ that has a tight anti-concentration bound to overcome the under-estimation of the optimal arm and concentration bound to overcome the over-estimation of the suboptimal arms.

For the simplicity of notation, we denote $\mathcal{P}(\mu, n)$ as the sampling distribution, where $\mu$ and $n$ are some input parameters. In particular, for ExpTS, we will choose $\mu = \hat{\mu}_i(t)$ and $n = T_i(t)$ for arm $i \in [K]$ at round $t$. We define $\mathcal{P}(\mu, n)$ as a distribution with PDF

$$f(x; \mu, n) = 1/2|(nb_n \cdot \text{kl}(\mu, x))'|e^{-nb_n \cdot \text{kl}(\mu, x)} = \frac{nb_n \cdot |x - \mu|}{2V(x)}e^{-nb_n \cdot \text{kl}(\mu, x)}, \qquad (4.3)$$

where $(\text{kl}(\mu, x))'$ denotes the derivative of $\text{kl}(\mu, x)$ with respect to $x$, and $b_n$ is a function of $n$ and will be chosen later.

We assume the reward is supported in $[R_{\min}, R_{\max}]$. Note that $R_{\min} = 0$, and $R_{\max} = 1$ for Bernoulli rewards, and $R_{\min} = -\infty$, and $R_{\max} = \infty$ for Gaussian rewards. Let $p(x)$ and $q(x)$ be the density functions of two exponential family distributions with mean values $\mu_p$ and $\mu_q$ respectively. By the definition in Section 3, we have $\text{kl}(\mu_p, \mu_q) = \text{KL}(p(x), q(x)) = \int_{R_{\min}}^{R_{\max}} p(x) \log \frac{p(x)}{q(x)} dx$.

**Proposition 4.1.** *If the mean reward of $q(x)$ is equal to the maximum value in its support, i.e., $\mu_q = R_{\max}$, we will have $kl(\mu, R_{\max}) = \infty$ for any $\mu < R_{\max}$.*

*Proof.* First consider the case that $R_{\max} < \infty$. Since the mean value concentrates on the maximum value, we must have $q(x) = 0$ for all $x < R_{\max}$, which immediately implies $\text{kl}(\mu, R_{\max}) = \infty$ for any $\mu < R_{\max}$. For the case that $R_{\max} = \infty$, from (3.3) and the assumption that $V < \infty$, we also have $\text{kl}(\mu, \infty) = (\infty - \mu)^2/V = \infty$. $\qquad\square$

Similarly, we can also prove that $\text{kl}(\mu, R_{\min}) = \infty$ for $\mu > R_{\min}$. Based on these properties, we can easily verify that a sample from the proposed sampling distribution $\theta \sim \mathcal{P}$ has the following tail bounds: for $z \in [\mu, R_{\max})$, it holds that

$$\mathbb{P}(\theta \geq z) = \int_z^{R_{\max}} f(x; \mu, n)dx = -1/2e^{-nb_n \cdot \text{kl}(\mu, x)}\Big|_z^{R_{\max}} = 1/2e^{-nb_n \cdot \text{kl}(\mu, z)}, \qquad (4.4)$$

and for $z \in (R_{\min}, \mu]$, it holds that

$$\mathbb{P}(\theta \leq z) = \int_{R_{\min}}^z f(x; \mu, n)dx = 1/2e^{-nb_n \cdot \text{kl}(\mu, x)}\Big|_{R_{\min}}^z = 1/2e^{-nb_n \cdot \text{kl}(\mu, z)}. \qquad (4.5)$$

Note that $\int_{R_{\min}}^{R_{\max}} f(x; \mu, n)dx = \int_{R_{\min}}^{\mu} f(x; \mu, n)dx + \int_{\mu}^{R_{\max}} f(x; \mu, n)dx = 1$, which indicates the PDF of $\mathcal{P}$ is well-defined.

**Intuition for the Design of the Sampling Distribution.** The tail bounds in (4.4) and (4.5) provide proper anti-concentration and concentration bounds for the sampling distribution $\mathcal{P}$ as long as we have corresponding lower and upper bounds of $e^{-nb_n \cdot \text{kl}(\mu, z)}$. When $n$ is large, we will choose $b_n$ to be close to 1, and thus (4.4) and (4.5) ensure that the sample of the corresponding arm concentrates in the interval $(\mu - \epsilon, \mu + \epsilon)$ with an exponentially small probability $e^{-n\text{kl}(\mu-\epsilon, \mu+\epsilon)}$, which is crucial for achieving a tight finite-time regret.

**How to Sample from $\mathcal{P}$.** We show that sampling from $\mathcal{P}$ is tractable when the CDF of $\mathcal{P}$ is invertible. In particular, according to (4.4) and (4.5), the CDF of $\mathcal{P}(\mu, n)$ is

$$F(x) = \begin{cases} 1 - 1/2e^{-nb_n \cdot \text{kl}(\mu, x)} & x \geq \mu, \\ 1/2e^{-nb_n \cdot \text{kl}(\mu, x)} & x \leq \mu. \end{cases}$$

Table 2: Comparisons of different algorithms on $K$-armed bandits over a horizon $T$. For any algorithm, *Asym. Opt* is the indicator whether it is asymptotically optimal, *minimax ratio* is the scaling of its worst-case regret w.r.t. the minimax optimal regret $O(\sqrt{VKT})$, *sub-UCB* indicates whether it satisfies the sub-UCB criteria, and *Anytime* indicates whether it needs the knowledge of the horizon length $T$ in advance.

| Algorithm | Reward Type | Asym. Opt | Finite-Time Regret | | Anytime | References |
|---|---|---|---|---|---|---|
| | | | Minimax Ratio | Sub-UCB | | |
| MOSS | $[0,1]$ | no | 1 | no | no | [6] |
| Anytime MOSS | $[0,1]$ | no | 1 | no | yes | [14] |
| KL-UCB$^{++}$ | Exponential Family | yes | 1 | no | no | [32] |
| OCUCB | sub-Gaussian | no | $\sqrt{\log\log T}$ | yes | yes | [28] |
| AdaUCB | Gaussian | yes | 1 | yes | no | [29] |
| MS | sub-Gaussian | yes | $\sqrt{\log K}$ | yes | yes | [10] |
| ExpTS | Exponential Family | yes | $\sqrt{\log K}$ | yes | yes | This paper |
| ExpTS$^+$ | Exponential Family | yes | 1 | no | yes | This paper |

To sample from $\mathcal{P}(\mu, n)$, we can first pick $y$ uniformly random from $[0, 1]$. Then, for $y \geq 1/2$, we solve the equation $y = 1 - 1/2e^{-nb_n \cdot \mathrm{kl}(\mu, x)}$ for $x$ ($x \geq \mu$), which is equivalent to solving $\log(1/(2(1-y)))/(nb_n) = \mathrm{kl}(\mu, x)$. For $y \leq 1/2$, we solve the equation $y = 1/2e^{-nb_n \cdot \mathrm{kl}(\mu, x)}$ for $x$ ($x \leq \mu$), which is equivalent to solving $\log(1/(2y))/(nb_n) = \mathrm{kl}(\mu, x)$. If $b(\theta)$ is reversible and the mapping $\theta \mapsto \mu$ is given[4], then according to (3.1), $\mathrm{kl}(\mu, x)$ is also reversible for $x$. We can obtain an exact sample from distribution $\mathcal{P}$ by solving $\mathrm{kl}(\mu, x) = \log(1/(2y))/(nb_n)$. Alternatively, we can also use approximate sampling methods such as Monte Carlo Markov Chain and Hastings-Metropolis [25] or gradient based Langevin Monte Carlo [35] to obtain samples from the target distribution.

### 4.3 Regret Analysis of ExpTS for Exponential Family Rewards

Now we present the regret bound of ExpTS for general exponential family bandits. The sampling distribution used in Algorithm 1 is defined in (4.3).

**Theorem 4.2.** *Let $b_n = (n-1)/n$. Let $\mathcal{P}$ be the sampling distribution defined in Section 4.2. There exist universal constants $C_0, C_1 > 0$ such that the regret of Algorithm 1 satisfies*

$$R_\mu(T) \leq C_0 \left( \sum_{i \in [K]:\Delta_i > \lambda} \Delta_i + \frac{V \log(T\Delta_i^2/V)}{\Delta_i} \right) + \max_{i \in [K], \Delta_i \leq \lambda} \Delta_i \cdot T, \tag{4.6}$$

$$R_\mu(T) \leq C_1 \left( \sum_{i=2}^{K} \Delta_i + \sqrt{VKT \log K} \right), \tag{4.7}$$

*where $\lambda \geq 16\sqrt{V/T}$, and also satisfies the following asymptotic bound simultaneously:*

$$\lim_{T \to \infty} \frac{R_\mu(T)}{\log T} = \sum_{i=2}^{K} \frac{\Delta_i}{kl(\mu_i, \mu_1)}. \tag{4.8}$$

**Remark 4.3.** *Similar to the argument by Auer and Ortner [7], we can see that the logarithm term in (4.6) is the main term for suitable $\lambda$. For instance, if we choose $\lambda = 16\sqrt{V/T}$, we will have $\max_{i \in [K], \Delta_i \leq \lambda} \Delta_i T \leq \sqrt{VT}$, which is in the order of $O(V/\Delta_i)$ due to $\Delta_i \leq \lambda$. Thus it is obvious to see that the regret in (4.6) satisfies the sub-UCB criteria.*

It is worth highlighting that ExpTS is an anytime algorithm and simultaneously satisfies the sub-UCB criteria in (1.2), the minimax optimal regret up to a factor $\sqrt{\log K}$, and the asymptotically optimal regret. ExpTS is also the first Thompson sampling algorithm that provides finite-time regret bounds for exponential family of rewards. Compared with state-of-the-art MAB algorithms listed in Table 2, ExpTS is comparable to the best known UCB algorithms that work for exponential family of reward distributions and no algorithms can dominate ExpTS. In particular, compared with MS [10] and

---

[4]This is true for distributions such as Gaussian with known variance, exponential distribution, and Bernoulli.

OCUCB [28], ExpTS is asymptotically optimal for exponential family of rewards, while MS is only asymptotically optimal for sub-Gaussian rewards and OCUCB is not asymptotically optimal. We note that Exponential Family does not cover the sub-Gaussian rewards. However, since we only use the tail bound to approximate the reward distribution, ExpTS can also be extended to solve sub-Gaussian reward bandits, which we leave as a future open direction.

### 4.4 Simple Variants for Gaussian and Bernoulli Reward Distributions

The choice of $\mathcal{P}$ in (4.3) seems complicated for a general exponential family reward distribution, even though we only need the sampling distribution to satisfy a nice tail bound derived from this reward distribution. When the reward distribution has a closed form such as Gaussian and Bernoulli distributions, we can replace $\mathcal{P}$ with the posterior in standard Thompson sampling and obtain the asymptotic and finite-time regrets in the previous section.

**Theorem 4.4.** *If the reward follows a Gaussian distribution with a known variance $V$, we can set the sampling distribution in Algorithm 1 as $\mathcal{N}(\widehat{\mu}_i(t), V/T_i(t))$. The resulting algorithm (denoted as Gaussian-TS) enjoys the same regret bounds presented in Theorem 4.2.*

**Remark 4.5.** *Lemma 5 in Jin et al. [20] shows for Gaussian rewards, Gaussian-TS has a regret bound at least $\Omega(\sqrt{VKT\log T})$ if the standard regret decomposition discussed in Section 4.1 is adopted in the proof [5, 30, 20]. With our new regret decomposition and the lower confidence bound introduced in (4.2), we improve it to $O(\sqrt{VKT\log K})$.*

*Jin et al. [20] also shows that their algorithms MOTS/MOTS-$\mathcal{J}$ can overcome the under-estimation issue of (4.1). However, they are either at the cost of sacrificing the asymptotic optimality or not generalizable to exponential family bandits. In specific, (1) For Gaussian rewards, MOTS [20] enlarges the variance of Gaussian posterior by a factor of $1/\rho$, where $\rho \in (0,1)$, which loses the asymptotic optimality by a factor of $1/\rho$ resultantly. (2) For Gaussian rewards, MOTS-$\mathcal{J}$ [20] introduces the Rayleigh posterior to overcome the under-estimation while maintaining the asymptotic optimality. However, it is not clear whether the idea can be generalized to exponential family rewards. Interestingly, their experimental results show that compared with Rayleigh posterior, Gaussian posterior actually has a smaller regret empirically. Therefore, to use a Gaussian sampling distribution, the new regret decomposition and the novel lower confidence bound in our paper is a better way to overcome the under-estimation issue of Gaussian-TS.*

**Theorem 4.6.** *If the reward distribution is Bernoulli, we can set the sampling distribution $\mathcal{P}$ in Algorithm 1 as Beta posterior $\mathcal{B}(S_i(t)+1, T_i(t)-S_i(t)+1)$, where $S_i(t)$ is the number of successes among the $T_i(t)$ plays of arm $i$. We denote the resulting algorithm as Bernoulli-TS, which enjoys the same regret bounds as in Theorem 4.2.*

Agrawal and Goyal [5] proved that for Bernoulli rewards, Thompson sampling with Beta posterior is asymptotically optimal and has a worst-case regret in the order of $O(\sqrt{KT\log T})$. Our regret analysis improves the worst-case regret to $O(\sqrt{KT\log K})$. They also proved that Gaussian-TS applied to the Bernoulli reward setting has a regret $O(\sqrt{KT\log K})$. However, no asymptotic regret was guaranteed in this setting.

## 5 Minimax Optimal Thompson Sampling for Exponential Family Rewards

In this section, in order to remove the extra logarithm term in the worst-case regret of ExpTS, we introduce a new sampling distribution that adds a greedy exploration step to the sampling distribution used in ExpTS. Specifically, the new algorithm ExpTS$^+$ is the same as ExpTS but uses a new sampling distribution $\mathcal{P}^+(\mu, n)$. A sample $\theta$ is generated from $\mathcal{P}^+(\mu, n)$ in the following way: $\theta = \mu$ with probability $1 - 1/K$ and $\theta \sim \mathcal{P}(\mu, n)$ with probability $1/K$.

**Over-Estimation of Sub-Optimal Arms.** We first elaborate the over-estimation issue of sub-optimal arms, which results in the extra $\sqrt{\log K}$ term in the worst-case regret of Thompson sampling. To explain, suppose that the sample of each arm $i$ has a probability $p = \mathbb{P}(\theta_i(t) \geq \theta_1(t))$ to become larger than the sample of arm 1. Note that when this event happens, the algorithm chooses the wrong arm and thus incurs a regret. Intuitively, the probability of making a mistake will be $K - 1$ times larger due to the union bound over $K - 1$ sub-optimal arms, which leads to an additional $\sqrt{\log K}$ factor in the worst-case regret. To reduce the probability $\mathbb{P}(\theta_i(t) \geq \theta_1(t))$, ExpTS$^+$ adds a greedy step that chooses the ExpTS sample with probability $1/K$ and chooses the arm with the largest

empirical average reward with probability $1 - 1/K$. Then we can prove that for sufficiently large $s$, with high probability we have $\hat{\mu}_{is} < \theta_1(t)$ and in this case it holds that $\mathbb{P}(\theta_i(t) \geq \theta_1(t)) = p/K$. Thus the extra factor $\sqrt{\log K}$ in regret is removed.

In specific, we have the following theorem showing that $\text{ExpTS}^+$ is asymptotically optimal and minimax optimal simultaneously.

**Theorem 5.1.** *Let $b_n = (n-1)/n$. There exists a constant $C_1 > 0$ such that $\text{ExpTS}^+$ satisfies*

$$R_\mu(T) \leq C_1 \left( \sum_{i=2}^{K} \Delta_i + \sqrt{VKT} \right), \quad and \quad \lim_{T \to \infty} \frac{R_\mu(T)}{\log T} = \sum_{i=2}^{K} \frac{\Delta_i}{kl(\mu_i, \mu_1)}.$$

This is the first time that the Thompson sampling algorithm achieves the minimax and asymptotically optimal regret for exponential family of reward distributions. Moreover, $\text{ExpTS}^+$ is also an anytime algorithm since it does not need to know the horizon $T$ in advance.

**Remark 5.2** (Sub-Gaussian Rewards). *In the proof of Theorem 5.1, we do not need the strict form of the PDF of the empirical mean reward $\hat{\mu}_{is}$, but only need the maximal inequality (Lemma H.1). This means that the proof can be straightforwardly extended to sub-Gaussian reward distributions, where similar maximal inequality holds [21].*

*It is worth noting that MOTS proposed by [20] (Thompson sampling with a clipped Gaussian posterior) also achieves the minimax optimal regret for sub-Gaussian rewards, but it can not keep the asymptotic optimality simultaneously with the same algorithm parameters. In particular, to achieve the minimax optimality, MOTS will have an additional $1/\rho$ factor in the asymptotic regret with $0 < \rho < 1$. Moreover, different from $\text{ExpTS}^+$, MOTS is only designed for fixed $T$ setting and thus is not an anytime algorithm.*

**Remark 5.3** (Gaussian and Bernoulli Rewards). *Following the idea in Section 4.4, we can derive new algorithms Gaussian-$\text{TS}^+$ and Bernoulli-$\text{TS}^+$ for Gaussian and Bernoulli rewards by replacing the sampling distribution in $\text{ExpTS}^+$. However, the posterior distribution does not fully satisfy the properties shown in Section 4.2. In particular, the factor $b_n < 1$ in Theorem 5.1 is an essential requirement for the asymptotic analyses whereas the posterior distribution does not have this factor. Due to these extra challenges, the proof techniques used for Theorem 5.1 can not be directly applied to these two new algorithms, and it is interesting to further investigate whether they are simultaneously minimax and asymptotically optimal.*

## 6 Conclusions

We studied Thompson sampling for exponential family of reward distributions. We proposed the ExpTS algorithm and proved it satisfies the sub-UCB criteria for problem-dependent finite-time regret, as well as achieves the asymptotic optimality and the minimax optimality up to a factor of $\sqrt{\log K}$ for exponential family rewards. Furthermore, we proposed a variant of ExpTS, dubbed $\text{ExpTS}^+$, that adds a greedy exploration step to balance between the sample generated in ExpTS and the empirical mean reward for each arm. We proved that $\text{ExpTS}^+$ is simultaneously minimax and asymptotically optimal. We also extended our proof techniques to standard Thompson sampling with common posterior distributions and improved existing results. This work is mainly focused on the theoretical optimality of Thompson sampling type algorithms. It would be an interesting future direction to investigate the empirical performance of ExpTS and $\text{ExpTS}^+$.

## Acknowledgement

We thank the anonymous reviewers and the area chair for their helpful comments. This research is supported by the National Research Foundation, Singapore under its AI Singapore Programme (AISG Award No: AISG-PhD/2021-01-004[T]), by ASTAR, Singapore under Grant A19E3b0099, and by Bren Named Chair Professorship at Caltech. In particular, T. Jin is supported by the National Research Foundation, Singapore under its AI Singapore Programme (AISG Award No: AISG-PhD/2021-01-004[T]). X. Xiao is supported by ASTAR, Singapore under Grant A19E3b0099. A. Anandkumar is partially supported by Bren Named Chair Professorship at Caltech. The views and conclusions contained in this paper are those of the authors and should not be interpreted as representing any funding agencies.

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
