# A  Proof of the Finite-Time Regret Bound of ExpTS

In this section, we prove the finite-time regret bound of ExpTS presented in Theorem 4.2. Specifically, we prove the sub-UCB property of ExpTS in (4.6) and the nearly minimax optimal regret of ExpTS in (4.7).

## A.1  Proof of the Main Results

We first focus on bounding the number of pulls of arm $i$ for the case that $\Delta_i > 16\sqrt{V/T}$. We start with the decomposition. Note that due to the warm start of Algorithm 1, each arm has been pulled once in the first $K$ steps. For any $\epsilon > 8\sqrt{V/T}$, define event $E_{i,\epsilon}(t) = \{\theta_i(t) \leq \mu_1 - \epsilon\}$, for all $i \in [K]$, which indicates that the estimate of arm $i$ at time step $t$ is smaller than the lower bound of the true mean reward of arm 1 ($\mu_1 - \epsilon \leq \mu_1$). The expected number of times that Algorithm 1 plays arms $i$ is bounded as follows.

$$\mathbb{E}[T_i(T)] = 1 + \mathbb{E}\left[\sum_{t=K+1}^{T} \mathbb{1}\{A_t = i, E_{i,\epsilon}(t)\} + \sum_{t=K+1}^{T} \mathbb{1}\{A_t = i, E_{i,\epsilon}^c(t)\}\right]$$

$$= 1 + \underbrace{\mathbb{E}\left[\sum_{t=K+1}^{T} \mathbb{1}\{A_t = i, E_{i,\epsilon}(t)\}\right]}_{A} + \underbrace{\mathbb{E}\left[\sum_{t=K+1}^{T} \mathbb{1}\{A_t = i, E_{i,\epsilon}^c(t)\}\right]}_{B}, \quad \text{(A.1)}$$

where $E^c$ is the complement of an event $E$, $\epsilon > 8\sqrt{V/T}$ is an arbitrary constant, and we used the fact $T_i(T) = \sum_{t=1}^{T} \mathbb{1}\{A_t = i\}$. In what follows, we bound these terms individually.

**Bounding Term $A$:**  Let us define

$$\alpha_s = \sup_{x \in [0, \mu_1 - \epsilon - R_{\min})} \text{kl}(\mu_1 - \epsilon - x, \mu_1) \leq 4\log(T/s)/s. \quad \text{(A.2)}$$

We decompose the term $\mathbb{E}\left[\sum_{t=K+1}^{T} \mathbb{1}\{A_t = i, E_{i,\epsilon}(t)\}\right]$ by the following lemma.

**Lemma A.1.** *Let $M = \lceil 16V\log(T\epsilon^2/V)/\epsilon^2 \rceil$ and $\alpha_s$ be the same as defined in* (A.2). *Then, there exists a universal constant $C_2 > 0$,*

$$\mathbb{E}\left[\sum_{t=K+1}^{T} \mathbb{1}\{A_t = i, E_{i,\epsilon}(t)\}\right] \leq \sum_{s=1}^{M} \mathbb{E}\left[\left(\frac{1}{G_{1s}(\epsilon)} - 1\right) \cdot \mathbb{1}\{\widehat{\mu}_{1s} \in L_s\}\right] + \frac{C_2 V}{\epsilon^2},$$

*where $G_{is}(\epsilon) = 1 - F_{is}(\mu_1 - \epsilon)$, $F_{is}$ is the CDF of $\mathcal{P}(\widehat{\mu}_{is}, s)$, and $L_s = (\mu_1 - \epsilon - \alpha_s, R_{\max}]$.*

The first term on the right hand side could be further bounded as follows.

**Lemma A.2.** *Let $M$, $G_{1s}(\epsilon)$, and $L_s$ be the same as defined in Lemma A.1. Then there a universal constant $C_3 > 0$, such that*

$$\sum_{s=1}^{M} \mathbb{E}_{\widehat{\mu}_{1s}}\left[\left(\frac{1}{G_{1s}(\epsilon)}\right) \cdot \mathbb{1}\{\widehat{\mu}_{1s} \in L_s\}\right] \leq \frac{C_3 \cdot V\log(T\epsilon^2/V)}{\epsilon^2}.$$

Combining Lemma A.1 and Lemma A.2 together, we have the upper bound of term $A$ in (A.1).

$$A = \frac{(C_3 + C_2)\log(T\epsilon^2/V)}{\epsilon^2}.$$

**Bounding Term $B$:**  To bound the second term in (A.1), we first prove the following lemma that bounds the number of time steps when the empirical average reward of arm $i$ deviates from its mean value.

**Lemma A.3.** *Let $N = \min\{1/(1 - \text{kl}(\mu_i + \rho_i, \mu_1 - \epsilon)/\log(T\epsilon^2/V)), 2\}$. For any $\rho_i, \epsilon > 0$ that satisfies $\epsilon + \rho_i < \Delta_i$, then*

$$\mathbb{E}\left[\sum_{t=K+1}^{T} \mathbb{1}\{A_t = i, E_{i,\epsilon}^c(t)\}\right] \leq 1 + \frac{2V}{\rho_i^2} + \frac{V}{\epsilon^2} + \frac{N\log(T\epsilon^2/V)}{\text{kl}(\mu_i + \rho_i, \mu_1 - \epsilon)}.$$

Applying Lemma A.3, we have the following bound for term $B$ in (A.1).

$$\mathbb{E}\left[\sum_{t=K+1}^{T}\mathbb{1}\{A_t = i, E_{i,\epsilon}^c(t)\}\right] \leq 1 + \frac{2V}{\rho_i^2} + \frac{V}{\epsilon^2} + \frac{N\log(T\epsilon^2/V)}{\mathrm{kl}(\mu_i + \rho_i, \mu_1 - \epsilon)}$$

$$\leq 1 + \frac{2V}{\rho_i^2} + \frac{V}{\epsilon^2} + \frac{4V\log(T\epsilon^2/V)}{(\Delta_i - \epsilon - \rho_i)^2},$$

where the last inequality is due to (3.3) and $N \leq 2$.

**Putting It Together:** Substituting the bounds of terms $A$ and $B$ back into (A.1), we have

$$\mathbb{E}[T_i(T)] = 1 + \frac{4V\log(T\epsilon^2/V)}{(\Delta_i - \epsilon - \rho_i)^2} + \frac{2V}{\rho_i^2} + \frac{V}{\epsilon^2} + \frac{(C_3 + C_2)V\log(T\epsilon^2/V)}{\epsilon^2}.$$

Let $\epsilon = \rho_i = \Delta_i/4$, we have

$$\mathbb{E}[T_i(T)] = 1 + \frac{(C_3 + C_2 + 64)V\log(T\Delta_i^2/V)}{\Delta_i^2}.$$

Note that we have assumed $\Delta_i > 16\sqrt{V/T}$ at the beginning of the proof. Therefore, there exists a universal constant $C_0 > 0$ such that

$$R_\mu(T) \leq C_0 \cdot \sum_{i \in [K]: \Delta_i > \lambda} O\left(\Delta_i + \frac{V\log(T\Delta_i^2/V)}{\Delta_i}\right) + \max_{i \in [K], \Delta_i \leq \lambda} \Delta_i \cdot T,$$

for any $\lambda \geq 16\sqrt{V/T}$. By choosing $\lambda = 16\sqrt{VK\log K/T}$, we obtain the following worst-case regret: $R_\mu(T) \leq C_1 \cdot \sqrt{VKT\log K}$ for some universal constant $C_1$. This completes the proof of the finite-time regret bounds of ExpTS.

## A.2 Proof of Supporting Lemmas

In this subsection, we prove the lemmas used in the proof of our main results in this section.

### A.2.1 Proof of Lemma A.1

Define $\mathcal{E}$ to be the event such that $\widehat{\mu}_{1s} \in L_s$ holds for all $s \in [T]$. The proof of Lemma A.1 needs the following lemma, which is used for bounding $\mathbb{P}(\mathcal{E}^c)$.

**Lemma A.4.** *Let $\epsilon > 0$, $b \in [K]$ and $f(\epsilon) = \lceil 16V\log(T\epsilon^2/(bV))/\epsilon^2 \rceil$. Assume $T \geq bf(\epsilon)$. Then, there exists a universal constant $C_2$ such that*

$$\mathbb{P}\left(\exists 1 \leq s \leq f(\epsilon): \hat{\mu}_{1s} \leq \mu_1 - \epsilon, kl(\hat{\mu}_{1s}, \mu_1) \geq 4\log(T/(bs))/s\right) \leq \frac{C_2 bV}{T\epsilon^2}.$$

The proof of Lemma A.4 could be found in Section G. Now, we are ready to prove Lemma A.1.

*Proof of Lemma A.1.* The indicator function can be decomposed based on $\mathcal{E}$, that is

$$\mathbb{E}\left[\sum_{t=K+1}^{T}\mathbb{1}\{A_t = i, E_{i,\epsilon}(t)\}\right]$$

$$\leq T \cdot \mathbb{P}(\mathcal{E}^c) + \mathbb{E}\left[\sum_{t=K+1}^{T}\left[\mathbb{1}\{A_t = i, E_{i,\epsilon}(t)\} \cdot \mathbb{1}\{\widehat{\mu}_{1T_i(t-1)} \in L_{T_i(t-1)}\}\right]\right]$$

$$\leq \frac{C_2 V}{T\epsilon^2} + \mathbb{E}\left[\sum_{t=K+1}^{T}\left[\mathbb{1}\{A_t = i, E_{i,\epsilon}(t)\} \cdot \mathbb{1}\{\widehat{\mu}_{1T_i(t-1)} \in L_{T_i(t-1)}\}\right]\right], \quad \text{(A.3)}$$

where the second inequality is due to Lemma A.4 with $b = 1$ and from the fact $\epsilon > 8\sqrt{V/T}$, $T \geq f(\epsilon)$. Let $\mathcal{F}_t = \sigma(A_1, r_1, \cdots, A_t, r_t)$ be the filtration. Note that $\theta_i(t)$ is sampled from

$\mathcal{P}(\hat{\mu}_i(t-1), T_i(t-1))$. Recall the definition, we know that $\hat{\mu}_i(t-1) = \hat{\mu}_{is}$ as long as $s = T_i(t-1)$. By the definition of $G_{is}(x)$, it holds that

$$G_{1T_1(t-1)}(\epsilon) = \mathbb{P}(\theta_1(t) \geq \mu_1 - \epsilon \mid \mathcal{F}_{t-1}). \tag{A.4}$$

Consider two cases. **Case 1:** $t : T_1(t-1) \leq M$. The proof of this case is similar to that of [30, Theorem 36.2]. Let $A'_t = \arg\max_{i \neq 1} \theta_i(t)$. Then

$$\begin{aligned}
\mathbb{P}(A_t = 1 \mid \mathcal{F}_{t-1}) &\geq \mathbb{P}(\{\theta_1(t) \geq \mu_1 - \epsilon\} \cap \{A'_t = i, E_{i,\epsilon}(t)\} \mid \mathcal{F}_{t-1}) \\
&= \mathbb{P}(\theta_1(t) \geq \mu_1 - \epsilon \mid \mathcal{F}_{t-1}) \cdot \mathbb{P}(A'_t = i, E_{i,\epsilon}(t) \mid \mathcal{F}_{t-1}) \\
&\geq \frac{G_{1T_1(t-1)}(\epsilon)}{1 - G_{1T_1(t-1)}(\epsilon)} \cdot \mathbb{P}(A_t = i, E_{i,\epsilon}(t) \mid \mathcal{F}_{t-1}), \tag{A.5}
\end{aligned}$$

The first inequality is due to the fact that when both event $\{\theta_1(t) \geq \mu_1 - \epsilon\}$ and event $\{A'_t = i, E_{i,\epsilon}(t)\}$ hold, we must have $\{A_t = 1\}$. The first equality is due to $\theta_1(t)$ is conditionally independent of $A'_t$ and $E_{i,\epsilon}(t)$ given $\mathcal{F}_{t-1}$. For the last inequality, let $C = \{A_t = i, E_{i,\epsilon}(t) \text{ occurs}\}$, $A = \{A'_t = i, E_{i,\epsilon}(t) \text{ occurs}\}$ and $B = \{\theta_1(t) \leq \mu_1 - \epsilon\}$. Then $A$ and $B$ are conditionally independent given $\mathcal{F}_{t-1}$. Besides, if $C$ happens, then $A_t = i$ and $\theta_i(t) \leq \mu_1 - \epsilon$. This implies the $\theta_1(t) \leq \mu_1 - \epsilon$ (otherwise, we will have $A_t \neq i$). Therefore, if $C$ happens, we must have $A'_t = i, E_{i,\epsilon}(t)$ occurs and $\theta_1(t) \leq \mu_1 - \epsilon$. Therefore, $C \subseteq A \cap B$ and

$$\begin{aligned}
\mathbb{P}(A_t &= i, E_{i,\epsilon}(t) \text{ occurs} \mid \mathcal{F}_{t-1}) \\
&= \mathbb{P}(C \mid \mathcal{F}_{t-1}) \\
&\leq \mathbb{P}(A \cap B \mid \mathcal{F}_{t-1}) \\
&\leq \mathbb{P}(A \mid \mathcal{F}_{t-1}) \cdot \mathbb{P}(B \mid \mathcal{F}_{t-1}) \\
&= \mathbb{P}(A'_t = i, E_{i,\epsilon}(t) \text{ occurs} \mid \mathcal{F}_{t-1}) \cdot \mathbb{P}(\theta_1(t) \leq \mu_1 - \epsilon \mid \mathcal{F}_{t-1}) \\
&= \mathbb{P}(A'_t = i, E_{i,\epsilon}(t) \text{ occurs} \mid \mathcal{F}_{t-1}) \cdot \left(1 - \mathbb{P}(\theta_1(t) > \mu_1 - \epsilon \mid \mathcal{F}_{t-1})\right).
\end{aligned}$$

Note that from (A.4), $G_{1T_1(t-1)}(\epsilon) = \mathbb{P}(\theta_1(t) \geq \mu_1 - \epsilon \mid \mathcal{F}_{t-1})$. Therefore, the above equation implies the last inequality of (A.5). Therefore, we have

$$\begin{aligned}
\mathbb{E}\left[\sum_{t:T_1(t-1)\leq M} \mathbb{1}\{A_t = i, E_{i,\epsilon}(t)\}\right] &\leq \mathbb{E}\left[\sum_{t:T_1(t-1)\leq M} \left(\frac{1}{G_{1T_1(t-1)}(\epsilon)} - 1\right)\mathbb{P}(A_t = 1 \mid \mathcal{F}_{t-1})\right] \\
&= \mathbb{E}\left[\sum_{t:T_1(t-1)\leq M} \left(\frac{1}{G_{1T_1(t-1)}(\epsilon)} - 1\right)\mathbb{1}\{A_t = 1\}\right] \\
&\leq \mathbb{E}\left[\sum_{s=1}^{M} \left(\frac{1}{G_{1s}(\epsilon)} - 1\right)\right]. \tag{A.6}
\end{aligned}$$

The first inequality is from (A.5). The first equality is due to $\mathbb{E}[\mathbb{1}\{A_t = 1\}] = \mathbb{P}(A_t = 1 \mid \mathcal{F}_{t-1})$. For the last inequality, note that due to the indicator function, the summation in first inequality is not zero only when $\mathbb{1}\{A_t = 1\} = 1$. And $\mathbb{1}\{A_t = 1\} = 1$ further means that we have pulled the best arm (arm 1) at time $t$. Therefore, the summation over all $T_1(t-1)$ conditional on $\mathbb{1}\{A_t = 1\} = 1$ is equivalent to the summation over $s$, which is the number of pulls of arm 1.

**Case 2:** $t : T - 1 \geq T_1(t-1) > M$. For this case, we have

$$\begin{aligned}
\mathbb{E}&\left[\sum_{t:T_1(t-1)>M}^{T-1} \mathbb{1}\{A_t = i, E_{i,\epsilon}(t)\}\right] \\
&\leq \mathbb{E}\left[\sum_{t:T_1(t-1)>M}^{T} \mathbb{1}\{\theta_1(t) < \mu_1 - \epsilon\}\right] \\
&\leq T \cdot \mathbb{P}\big(\exists s > M : \hat{\mu}_{1s} < \mu_1 - \epsilon/2\big) \\
&\quad + \mathbb{E}\left[\left(\sum_{t:T_1(t-1)>M}^{T} \mathbb{1}\{\theta_1(t) < \mu_1 - \epsilon)\}\right)\mathbb{1}\left\{\forall t \in \{t \mid T_1(t-1) > M\} : \hat{\mu}_{1T_1(t-1)} \geq \mu_1 - \epsilon/2\right\}\right]
\end{aligned}$$

$$\leq T \cdot e^{-M(\mu_1 - (\mu_1 - \epsilon/2))^2/(2V)}$$

$$+ \mathbb{E}\left[ \sum_{t:T_1(t-1)>M} \mathbb{P}\left[\theta_1(t) < \mu_1 - \epsilon \mid \hat{\mu}_{1T_1(t-1)} \geq \mu_1 - \epsilon/2 \right] \right]$$

$$\leq \frac{V}{\epsilon^2} + T \cdot e^{-M\epsilon^2/(16V)}$$

$$\leq \frac{2V}{\epsilon^2}. \tag{A.7}$$

In the first inequality, we use the fact that $\{A_t = i, E_{i,\epsilon}(t)\} \subseteq \{\theta_1(t) < \mu_1 - \epsilon\}$. In the second inequality, we decompose the term into two events. Event one: there exists a $t$ with $T_1(t-1) > M$ and $\hat{\mu}_{1T_1(t-1)} < \mu_1 - \epsilon/2$. Event two: for all $T_1(t-1) > M$, $\hat{\mu}_{1T_1(t-1)} \geq \mu_1 - \epsilon/2$. In the third inequality, we use Lemma H.1 and following facts

$$\mathbb{E}\left[ \left( \sum_{t:T_1(t-1)>M}^{T} \mathbb{1}\{\theta_1(t) < \mu_1 - \epsilon\} \right) \mathbb{1}\left\{ \forall t \in \{t \mid T_1(t-1) > M\} : \hat{\mu}_{1T_1(t-1)} \geq \mu_1 - \epsilon/2 \right\} \right]$$

$$= \mathbb{E}\left[ \sum_{t:T_1(t-1)>M}^{T} \left( \mathbb{1}\{\theta_1(t) < \mu_1 - \epsilon\} \cdot \mathbb{1}\left\{ \forall t \in \{t \mid T_1(t-1) > M\} : \hat{\mu}_{1T_1(t-1)} \geq \mu_1 - \epsilon/2 \right\} \right) \right]$$

$$\leq \mathbb{E}\left[ \sum_{t:T_1(t-1)>M}^{T} \left( \mathbb{1}\{\theta_1(t) < \mu_1 - \epsilon\} \cdot \mathbb{1}\{\hat{\mu}_{1T_1(t-1)} \geq \mu_1 - \epsilon/2\} \right) \right]$$

$$= \mathbb{E}\left[ \sum_{t:T_1(t-1)>M}^{T} \left( \mathbb{1}\left\{ \theta_1(t) < \mu_1 - \epsilon, \hat{\mu}_{1T_1(t-1)} \geq \mu_1 - \epsilon/2 \right\} \right) \right]$$

$$= \mathbb{E}\left[ \sum_{t:T_1(t-1)>M}^{T} \mathbb{E}\left[ \mathbb{1}\left\{ \theta_1(t) < \mu_1 - \epsilon, \hat{\mu}_{1T_1(t-1)} \geq \mu_1 - \epsilon/2 \right\} \,\Big|\, T_1(t-1) \right] \right]$$

$$= \mathbb{E}\left[ \sum_{t:T_1(t-1)>M}^{T} \mathbb{P}(\theta_1(t) < \mu_1 - \epsilon, \hat{\mu}_{1T_1(t-1)} \geq \mu_1 - \epsilon/2) \right]$$

$$\leq \mathbb{E}\left[ \sum_{t:T_1(t-1)>M}^{T} \mathbb{P}\left(\theta_1(t) < \mu_1 - \epsilon \mid \hat{\mu}_{1T_1(t-1)} \geq \mu_1 - \epsilon/2\right) \right],$$

where the first inequality is due to the fact $\mathbb{1}\left( \forall t \in T_1(t-1) > M : \hat{\mu}_{1T_1(t-1)} \geq \mu_1 - \epsilon/2 \right) \leq \mathbb{1}\left( \hat{\mu}_{1T_1(t-1)} \geq \mu_1 - \epsilon/2 \right)$ for any $t \in \{t \mid T_1(t-1) > M\}$, and the second inequality is due to $\mathbb{P}(A, B) = \mathbb{P}(A) \cdot \mathbb{P}(A \mid B) \leq \mathbb{P}(A \mid B)$.

In the fourth inequality of (A.7), we apply the following results

$$\mathbb{E}\left[ \sum_{t:T_1(t-1)>M} \mathbb{P}(\theta_1(t) \leq \mu_1 - \epsilon \mid \hat{\mu}_{1T_1(t-1)} \geq \mu_1 - \epsilon/2) \right]$$

$$= \mathbb{E}\left[ \sum_{t:T_1(t-1)>M} 1/2 e^{-T_1(t-1)b_{T_1(t-1)}\text{kl}(\hat{\mu}_{1T_1(t-1)}, \mu_1 - \epsilon)} \right]$$

$$\leq \mathbb{E}\left[ \sum_{t:T_1(t-1)>M} e^{-\frac{M}{2}\text{kl}(\hat{\mu}_{1T_1(t-1)}, \mu_1 - \epsilon)} \right]$$

$$\leq \mathbb{E}\left[ \sum_{t:T_1(t-1)>M} e^{-\frac{M}{2}\text{kl}(\mu_1 - \epsilon/2, \mu_1 - \epsilon)} \right]$$

$$\leq T \cdot e^{-M\epsilon^2/(16V)},$$

where the first equality is due to $\theta_1(t) \sim \mathcal{P}(\hat{\mu}_{1T_1(t-1)}, T_1(t-1))$ and (4.5), the first inequality is due to the fact that $b_s \geq 1/2$ for any $s > 1$ and $T_1(t-1) > M$, the second inequality is due to $\hat{\mu}_{1T_1(t-1)} \geq \mu_1 - \epsilon/2$ and the fact that from Proposition 3.2, $\mathrm{kl}(x, \mu_1 - \epsilon)$ is increasing for $x > \mu_1 - \epsilon/2$, and the last inequality is due to (3.3). Combining (A.3), (A.6), and (A.7) together, we finish the proof of Lemma A.1. $\qquad\square$

Note that in order to bound term $A$, we need the following lemma that states the upper bound of the first term in Lemma A.1.

### A.2.2 Proof of Lemma A.2

Let $p(x)$ be the PDF of $\hat{\mu}_{1s}$ and $\theta_{1s}$ be a sample from $\mathcal{P}(\hat{\mu}_{1s}, s)$. We have

$$
\sum_{s=1}^{M} \mathbb{E}_{\hat{\mu}_{1s}} \left[ \left( \frac{1}{G_{1s}(\epsilon)} - 1 \right) \cdot \mathbb{1}\{\hat{\mu}_{1s} \in L_s\} \right]
$$

$$
\leq \underbrace{\sum_{s=1}^{M} \left( \int_{\mu_1 - \epsilon/2}^{R_{\max}} p(x)/\mathbb{P}(\theta_{1s} \geq \mu_1 - \epsilon \mid \hat{\mu}_{1s} = x)\mathrm{d}x - 1 \right)}_{A_1}
$$

$$
+ \underbrace{\sum_{s=1}^{M} \int_{\mu_1 - \epsilon}^{\mu_1 - \epsilon/2} p(x)/\mathbb{P}(\theta_{1s} \geq \mu_1 - \epsilon \mid \hat{\mu}_{1s} = x)\mathrm{d}x}_{A_2}
$$

$$
+ \underbrace{\sum_{s=1}^{M} \int_{\mu_1 - \epsilon - \alpha_s}^{\mu_1 - \epsilon} \left[ p(x)/\mathbb{P}(\theta_{1s} \geq \mu_1 - \epsilon \mid \hat{\mu}_{1s} = x) \right]\mathrm{d}x}_{A_3}, \tag{A.8}
$$

where the inequality is due to the definition of $L_s$[5].

**Bounding term $A_1$.** For term $A_1$, we divide $\sum_{s=1}^{M}$ into two term, i.e., $\sum_{s=1}^{\lfloor 32V/\epsilon^2 \rfloor}$ and $\sum_{s=\lceil 32V/\epsilon^2 \rceil}^{M}$. Intuitively, for $s \geq 32V/\epsilon^2$, $\mathbb{P}(\theta_{1s} \geq \mu_1 - \epsilon \mid \hat{\mu}_{1s} \geq \mu_1 - \epsilon/2)$ will be large. We have

$$
A_1 = \sum_{s=1}^{M} \left( \int_{\mu_1 - \epsilon/2}^{R_{\max}} \frac{p(x)}{\mathbb{P}(\theta_{1s} \geq \mu_1 - \epsilon \mid \hat{\mu}_{1s} = x)}\mathrm{d}x - 1 \right)
$$

$$
\leq \frac{32V}{\epsilon^2} + \sum_{s=\lceil 32V/\epsilon^2 \rceil}^{M} \left( \int_{\mu_1 - \epsilon/2}^{R_{\max}} \frac{p(x)}{\mathbb{P}(\theta_{1s} \geq \mu_1 - \epsilon \mid \hat{\mu}_{1s} = x)}\mathrm{d}x - 1 \right)
$$

$$
\leq \frac{32V}{\epsilon^2} + \sum_{s=\lceil 32V/\epsilon^2 \rceil}^{M} \left( \frac{1}{1 - e^{-s/2 \cdot \mathrm{kl}(\mu_1 - \epsilon/2, \mu_1 - \epsilon)}} - 1 \right)
$$

$$
\leq \frac{32V}{\epsilon^2} + \sum_{s=\lceil 32V/\epsilon^2 \rceil}^{M} \left( \frac{1}{1 - e^{-s\epsilon^2/(16V)}} - 1 \right)
$$

$$
= \frac{16V}{\epsilon^2} + \sum_{s=\lceil 32V/\epsilon^2 \rceil}^{M} \frac{1}{e^{s\epsilon^2/(16V)} - 1}
$$

$$
\leq \frac{16V}{\epsilon^2} + \frac{16V}{\epsilon^2} \sum_{s=1}^{\infty} \frac{1}{e^{1+s} - 1}
$$

$$
\leq \frac{32V}{\epsilon^2}. \tag{A.9}
$$

---

[5] For the discrete reward distribution, we can use the Dirac delta function for the integral.

For the first inequality, we used the fact $\mathbb{P}(\theta_{1s} \geq \mu_1 - \epsilon \mid \hat{\mu}_{1s} \geq \mu_1 - \epsilon) \geq 1/2$, which is due to (4.4). The second inequality is due to (4.5) and the fact $b_s \geq 1/2$. The third inequality is due to (3.3).

**Bounding term $A_2$.** We have

$$A_2 = \sum_{s=1}^{M} \int_{\mu_1-\epsilon}^{\mu_1-\epsilon/2} \frac{p(x)}{\mathbb{P}(\theta_{1s} \geq \mu_1 - \epsilon \mid \hat{\mu}_{1s} = x)} \mathrm{d}x \leq 2 \sum_{s=1}^{\infty} e^{-s\epsilon^2/(8V)} \leq \frac{2}{e^{\epsilon^2/(8V)} - 1} \leq \frac{16V}{\epsilon^2},$$
(A.10)

where the first inequality is due to $\mathbb{P}(\theta_{1s} \geq \mu_1 - \epsilon \mid \hat{\mu}_{1s} \geq \mu_1 - \epsilon) \geq 1/2$ and from Lemma H.1, $\mathbb{P}(\hat{\mu}_{1s} \leq \mu_1 - \epsilon/2) \leq e^{-s\epsilon^2/(8V)}$, and the last inequality is due to $e^x - 1 \geq x$ for all $x > 0$.

**Bounding term $A_3$.** Note that the closed form of the probability density function of $\hat{\mu}_{1s}$ is hard to compute. Nevertheless, we only need to find an upper bound of the integration in $A_3$. In the following lemma, we show that it is possible to find such an upper bound with an explicit form.

**Lemma A.5.** *Let* $q(x) = |(s \cdot kl(x, \mu_1))'| e^{-s \cdot kl(x,\mu_1)} = s \int_x^{\mu_1} 1/V(t)dt \cdot e^{-s \cdot kl(x,\mu_1)}$, $g(x) = e^{sb_s \cdot kl(x,\mu_1-\epsilon)}$ *and* $p(x)$ *be the PDF of distribution of* $\hat{\mu}_{1s}$, *then*

$$\int_{\mu_1-\epsilon-\alpha_s}^{\mu_1-\epsilon} q(x)g(x)dx + e^{-s \cdot kl(\mu_1-\epsilon-\alpha_s,\mu_1)} \cdot g(\mu_1 - \epsilon - \alpha_s) \geq \int_{\mu_1-\epsilon-\alpha_s}^{\mu_1-\epsilon} p(x)g(x)dx.$$

The proof of Lemma A.5 could be found in Section G. Besides, we need the following inequality on kl-divergence, which resembles the three-point identity property. In particular, for $\mu_1 - \epsilon > x$, we have

$$-kl(x, \mu_1) + kl(x, \mu_1 - \epsilon) = -\int_x^{\mu_1} \frac{t - x}{V(t)}dt + \int_x^{\mu_1-\epsilon} \frac{t - x}{V(t)}dt$$

$$= -\int_{\mu_1-\epsilon}^{\mu_1} \frac{t - x}{V(t)}dt$$

$$\leq -\int_{\mu_1-\epsilon}^{\mu_1} \frac{t - (\mu_1 - \epsilon)}{V(t)}dt$$

$$= -kl(\mu_1 - \epsilon, \mu_1),$$
(A.11)

where the first and the last equality is due to (3.2). For term $A_3$, we have

$$A_3 \leq \sum_{s=1}^{M} \int_{\mu_1-\epsilon-\alpha_s}^{\mu_1-\epsilon} p(x)e^{sb_s \cdot kl(x,\mu_1-\epsilon)}\mathrm{d}x$$
(A.12)

$$\leq \sum_{s=1}^{M} \int_{\mu_1-\epsilon-\alpha_s}^{\mu_1-\epsilon} \left[q(x) \cdot e^{s \cdot kl(x,\mu_1-\epsilon)}\right]\mathrm{d}x + \sum_{s=1}^{M} e^{-s \cdot kl(\mu_1-\epsilon-\alpha_s,\mu_1)} \cdot e^{s \cdot kl(\mu_1-\epsilon-\alpha_s,\mu_1-\epsilon)}$$

$$\leq \sum_{s=1}^{M} \int_{\mu_1-\epsilon-\alpha_s}^{\mu_1-\epsilon} \left[|s \cdot kl(x,\mu_1)'| \cdot e^{-s \cdot kl(x,\mu_1)} \cdot e^{s \cdot kl(x,\mu_1-\epsilon)}\right]\mathrm{d}x + e^{-s\epsilon^2/(2V)} \cdot M$$

$$\leq \sum_{s=1}^{M} e^{-s \cdot kl(\mu_1-\epsilon,\mu_1)} \cdot \int_{\mu_1-\epsilon-\alpha_s}^{\mu_1} \left[|s \cdot kl(x,\mu_1)|'\right]\mathrm{d}x + e^{-s\epsilon^2/(2V)} \cdot M$$

$$\leq \sum_{s=1}^{M} e^{-s\epsilon^2/(2V)}(1 + s \cdot kl(\mu_1 - \epsilon - \alpha_s, \mu_1))$$

$$\leq \sum_{s=1}^{M} e^{-s\epsilon^2/(2V)}(1 + 4\log(T/s)),$$
(A.13)

where the first inequality is due to (4.4), the second inequality is due to Lemma A.5 and $b_s \leq 1$, the third inequality is due to (A.11), the fourth inequality is due to (A.11), and the last inequality is due to Lemma A.4 and the definition of $\alpha_s$. Let $d = \lceil V/\epsilon^2 \rceil$. For term $\sum_{s=1}^{d} \log(T/s)$, we have

$$\sum_{s=1}^{d} \log(T/s) = d\log T - \sum_{s=1}^{d} \log s$$

$$\leq d \log T - \left( \left. (s \log s - s) \right|_1^d - \log d \right)$$

$$\leq d \log(T/d) + d + \log d$$

$$\leq \frac{2V \log(T\epsilon^2/V)}{\epsilon^2}, \tag{A.14}$$

where the first inequality is due to $\sum_{x=a}^b f(x) \geq \int_a^b f(x)\mathrm{d}x - \max_{x \in [a,b]} f(x)$ for monotone function $f$. For term $\sum_{s=d}^M e^{-s\epsilon^2/(2V)} \log(T/s)$, we have

$$\sum_{s=d}^M e^{-s\epsilon^2/(2V)} \log(T/s) \leq \log(T/d) \sum_{s=d}^M e^{-s\epsilon^2/2V}$$

$$\leq \log(T/d) \sum_{s=1}^\infty e^{-s\epsilon^2/2V}$$

$$\leq \frac{\log(T/d)}{e^{\epsilon^2/(2V)} - 1}$$

$$\leq \frac{2V \log(T/d)}{\epsilon^2}$$

$$\leq \frac{2V \log(T\epsilon^2/V)}{\epsilon^2}, \tag{A.15}$$

where the fourth inequality is due to $e^x \geq 1 + x$ for $x > 0$. Substituting (A.15) and (A.14) to (A.13), we have $A_3 = 16V \log(T\epsilon^2/V)/\epsilon^2$. Substituting the bounds of $A_1$, $A_2$, and $A_3$ to (A.8), we have that there exists a constant $C_3$,

$$\sum_{s=1}^M \mathbb{E}_{\widehat{\mu}_{1s}} \left[ \left( \frac{1}{G_{1s}(\epsilon)} - 1 \right) \cdot \mathbb{1}\{\widehat{\mu}_{1s} \in L_s\} \right] \leq C_3 \cdot \left( \frac{V \log(T\epsilon^2/V)}{\epsilon^2} \right),$$

which completes the proof.

### A.2.3  Proof of Lemma A.3

Let $\mathcal{T} = \{t \in [K+1, T] : 1 - F_{iT_i(t-1)}(\mu_1 - \epsilon) > V/(T\epsilon^2)\}$. Then,

$$\mathbb{E}\left[ \sum_{t=K+1}^T \mathbb{1}\{A_t = i, E_{i,\epsilon}^c(t)\} \right]$$

$$\leq \mathbb{E}\left[ \sum_{t \in \mathcal{T}} \mathbb{1}\{A_t = i\} \right] + \mathbb{E}\left[ \sum_{t \notin \mathcal{T}} \mathbb{1}\{E_{i,\epsilon}^c(t)\} \right]$$

$$\leq \mathbb{E}\left[ \sum_{t \geq K+1} \left( \mathbb{1}\{A_t = i\} \cdot \mathbb{1}\{1 - F_{iT_i(t-1)}(\mu_1 - \epsilon) > V/(T\epsilon^2)\} \right) \right] + \mathbb{E}\left[ \sum_{t \notin \mathcal{T}} V/(T\epsilon^2) \right]$$

$$\leq \mathbb{E}\left[ \sum_{s \in [T]} \mathbb{1}\{1 - F_{is}(\mu_1 - \epsilon) > V/(T\epsilon^2)\} \right] + \frac{V}{\epsilon^2}$$

$$\leq \mathbb{E}\left[ \sum_{s=1}^T \mathbb{1}\{G_{is}(\epsilon) > V/(T\epsilon^2)\} \right] + \frac{V}{\epsilon^2}. \tag{A.16}$$

Let $s \geq N \log(T\epsilon^2/V)/\mathrm{kl}(\mu_i + \rho_i, \mu_1 - \epsilon)$. Note that

$$\frac{1}{N} = \max\left\{ 1 - \mathrm{kl}(\mu_i + \rho_i, \mu_1 - \epsilon)/\log(T\epsilon^2/V), \frac{1}{2} \right\}.$$

For case $1/N = 1/2$, we have

$$b_s \geq 1/2 = 1/N.$$

For case $1/N = 1 - \mathrm{kl}(\mu_i + \rho_i, \mu_1 - \epsilon)/\log(T\epsilon^2/V)$, we have

$$
\begin{aligned}
b_s &\geq 1 - 1/s \\
&\geq 1 - \mathrm{kl}(\mu_i + \rho_i, \mu_1 - \epsilon)/(N \log(T\epsilon^2/V)) \\
&\geq 1 - \mathrm{kl}(\mu_i + \rho_i, \mu_1 - \epsilon)/(\log(T\epsilon^2/V)) \\
&= 1/N,
\end{aligned}
$$

where the second inequality is due to $s \geq N \log(T\epsilon^2/V)/\mathrm{kl}(\mu_i + \rho_i, \mu_1 - \epsilon)$, the third inequality is due to $N > 1$. Let $X_{is}$ be a sample from the distribution $\mathcal{P}(\widehat{\mu}_{is}, s)$, if $\widehat{\mu}_{is} \leq \mu_i + \rho_i$, we have

$$
\mathbb{P}(X_{is} \geq \mu_1 - \epsilon) \leq \exp\left(-sb_s \mathrm{kl}(\widehat{\mu}_{is}, \mu_1 - \epsilon)\right) \leq \exp\left(-sb_s \mathrm{kl}(\mu_i + \rho_i, \mu_1 - \epsilon)\right) \leq \frac{V}{T\epsilon^2},
\tag{A.17}
$$

where the first inequality is from (4.4), the second inequality is due to the assumption $\widehat{\mu}_{is} \leq \mu_i + \rho_i$, and the last inequality is due to $s \geq N \log(T\epsilon^2/V)/\mathrm{kl}(\mu_i + \rho_i, \mu_1 - \epsilon)$ and $b_s \geq 1/N$. Note that when $\mathbb{P}(X_{is} \geq \mu_1 - \epsilon) \leq V/(T\epsilon^2)$ holds, $\mathbb{1}\{G_{is}(\epsilon) > V/(T\epsilon^2)\} = 0$. Now, we check the assumption $\widehat{\mu}_{is} \leq \mu_i + \rho_i$ that is needed for (A.17). From Lemma H.1, we have $\mathbb{P}(\widehat{\mu}_{is} > \mu_i + \rho_i) \leq \exp(-s\rho_i^2/(2V))$. Furthermore, it holds that

$$
\sum_{s=1}^{\infty} e^{-\frac{s\rho_i^2}{2V}} \leq \frac{1}{e^{\rho_i^2/(2V)} - 1} \leq \frac{2V}{\rho_i^2},
\tag{A.18}
$$

where the last inequality is due to the fact $1 + x \leq e^x$ for all $x$. Let $Y_{is}$ be the event that $\widehat{\mu}_{is} \leq \mu_i + \rho_i$ and $m = N \log(T\epsilon^2/V)/\mathrm{kl}(\mu_i + \rho_i, \mu_1 - \epsilon)$. We further obtain

$$
\begin{aligned}
\mathbb{E}\left[\sum_{s=1}^{T} \mathbb{1}\{G_{is}(\epsilon) > V/(T\epsilon^2)\}\right] &\leq \mathbb{E}\left[\sum_{s=1}^{T}[\mathbb{1}\{G_{is}(\epsilon) > V/(T\epsilon^2)\} \mid Y_{is}]\right] + \sum_{s=1}^{T}(1 - \mathbb{P}[Y_{is}]) \\
&\leq \mathbb{E}\left[\sum_{s=\lceil m \rceil}^{T} [\mathbb{1}\{\mathbb{P}(X_{is} > \mu_1 - \epsilon) > V/(T\epsilon^2))\} \mid Y_{is}]\right] \\
&\quad + \lceil m \rceil + \sum_{s=1}^{T}(1 - \mathbb{P}[Y_{is}]) \\
&\leq \lceil m \rceil + \sum_{s=1}^{T}(1 - \mathbb{P}[Y_{is}]) \\
&\leq 1 + \frac{2V}{\rho_i^2} + \frac{N \log(T\epsilon^2/V)}{\mathrm{kl}(\mu_i + \rho_i, \mu_1 - \epsilon)},
\end{aligned}
\tag{A.19}
$$

where the first inequality is due to the fact that $\mathbb{P}(A) \leq \mathbb{P}(A \mid B) + 1 - \mathbb{P}(B)$, the third inequality is due to (A.17) and the last inequality is due to (A.18). Substituting (A.19) into (A.16), we complete the proof.

# B  Proof of the Asymptotic Optimality of ExpTS

Now we prove the asymptotic regret bound (4.8) of ExpTS presented in Theorem 4.2.

## B.1  Proof of the Main Result

The proof in this section shares many components with the finite-time regret analysis presented in Section A. We reuse the decomposition (A.1) by specifying $\epsilon = 1/\log\log T$. In what follows, we bound terms $A$ and $B$, respectively.

**Bounding Term $A$:**   We reuse Lemma A.1. Then, it only remains term $\sum_{s=1}^{M} \mathbb{E}\left[(1/G_{1s}(\epsilon) - 1) \cdot \mathbb{1}\{\widehat{\mu}_{1s} \in L_s\}\right]$ to be bounded. We bound this term by the following lemma.

**Lemma B.1.** *Let $\epsilon = 1/\log\log T$. Let $M$, $G_{1s}(\epsilon)$, and $L_s$ be the same as defined in Lemma A.1. Then,*

$$\sum_{s=1}^{M} \mathbb{E}_{\widehat{\mu}_{1s}}\left[\left(\frac{1}{G_{1s}(\epsilon)} - 1\right) \cdot \mathbb{1}\{\widehat{\mu}_{1s} \in L_s\}\right] = O(V^2(\log\log T)^6 + V(\log\log T)^2 + 1).$$

Let $\epsilon = 1/\log\log T$. Combining Lemma B.1 and Lemma A.1 together, we have

$$A = O\big(V^2(\log\log T)^6 + V(\log\log T)^2 + 1\big).$$

**Bounding Term $B$:** Let $\rho_i = \epsilon = 1/\log\log T$. Applying Lemma A.3, we have

$$B = \mathbb{E}\left[\sum_{t=K+1}^{T} \mathbb{1}\{A_t = i, E_{i,\epsilon}^c(t)\}\right]$$

$$= O(1 + V(\log\log T)^2) + \frac{N\log(T/(V(\log\log T)^2))}{\mathrm{kl}(\mu_i + 1/\log\log T, \mu_1 - 1/\log\log T)}. \tag{B.1}$$

**Putting It Together:** Substituting the bound of term $A$ and $B$ into (A.1), we have

$$\mathbb{E}[T_i(T)] = O(1 + V^2(\log\log T)^6 + V(\log\log T)^2) + \frac{N\log(T/(V(\log\log T)^2))}{\mathrm{kl}(\mu_i + 1/\log\log T, \mu_1 - 1/\log\log T)}.$$

Note that for $T \to +\infty$, $N \to 1$. Therefore,

$$\lim_{T \to +\infty} \frac{\mathbb{E}[T_i(T)]}{\log T} = \frac{1}{\mathrm{kl}(\mu_i, \mu_1)}.$$

This completes the proof of asymptotic regret.

### B.2 Proof of Lemma B.1

The proof of this part shares many elements with the proof of Lemma A.2. The difference starts at bounding term $A_3$.

**Bounding term $A_3$.** We need to bound the term $\int_{\mu_1-\epsilon-\alpha_s}^{\mu_1-\epsilon} p(x)e^{s\mathrm{kl}(x,\mu_1-\epsilon)}\mathrm{d}x$. We divide the interval $[\mu_1 - \epsilon - \alpha_s, \mu_1 - \epsilon]$ into $n$ sub-intervals $[x_0, x_1), [x_1, x_2), \cdots, [x_{n-1}, x_n]$, such that $x_0 \leq x_1 \leq \cdots \leq x_n$. For $i \in [n-1]$, we let

$$x_i = \sup_{x:x \leq \mu_1 - \epsilon} 4\log(T/e^{i+1})/s < \mathrm{kl}(x, \mu_1) \leq 4\log(T/e^i)/s. \tag{B.2}$$

Let $n = \lceil \log T \rceil$ and $x_n = \mu_1$. Then, from definition of $\alpha_s$, $\mathrm{kl}(x_0, \mu_1) \geq \mathrm{kl}(\mu_1 - \epsilon - \alpha_s, \mu_1)$. Thus, $x_0 \leq \mu_1 - \epsilon - \alpha_s$. Now, continue on (A.12), we have

$$\int_{\mu_1-\epsilon-\alpha_s}^{\mu_1-\epsilon} p(x)e^{sb_s \cdot \mathrm{kl}(x,\mu_1-\epsilon)}\mathrm{d}x \leq \sum_{i=0}^{n} \int_{x_i}^{x_{i+1}} p(x)e^{sb_s\mathrm{kl}(x,\mu_1-\epsilon)}\mathrm{d}x$$

$$\leq \sum_{i=0}^{n} e^{sb_s\mathrm{kl}(x_i,\mu_1)} \int_{x_i}^{x_{i+1}} p(x)\mathrm{d}x$$

$$\leq \sum_{i=0}^{n} e^{sb_s\mathrm{kl}(x_i,\mu_1)}e^{-s \cdot \mathrm{kl}(x_{i+1},\mu_1)}$$

$$\leq \sum_{i=0}^{n} \left(\frac{T}{e^i}\right)^{b_s}\left(\frac{e^{i+1}}{T}\right)$$

$$= O\left(\int_0^{\ln T} \left(\frac{T}{e^x}\right)^{b_s} \cdot \frac{e^{x+1}}{T}\mathrm{d}x + e\right)$$

$$= O\left(\frac{1}{1-b_s}\right)$$

$$= O(s), \tag{B.3}$$

where the first inequality is due to $x_0 \leq \mu_1 - \epsilon - \alpha_s$ and $x_n = \mu_1 \geq \mu_1 - \epsilon$, the fourth inequality is due to the definition of $x_i$, and the first equality is due to $\sum_{x=a}^{b} f(x) \leq \int_a^b f(x)\mathrm{d}x + \max_{x \in [a,b]} f(x)$ for monotone function $f$. Now, we bound term $A_3$ as follows.

$$A_3 \leq \sum_{s=1}^{M} \int_{\mu_1 - \epsilon - \alpha_s}^{\mu_1 - \epsilon} p(x) e^{sb_s \cdot \mathrm{kl}(x, \mu_1 - \epsilon)} \mathrm{d}x$$

$$\leq \sum_{s=1}^{\lceil 4V(\log\log T)^3 \rceil} \int_{\mu_1 - \epsilon - \alpha_s}^{\mu_1 - \epsilon} p(x) e^{sb_s \cdot \mathrm{kl}(x, \mu_1 - \epsilon)} \mathrm{d}x$$

$$+ \sum_{s=\lceil 4V(\log\log T)^3 \rceil}^{M} \int_{\mu_1 - \epsilon - \alpha_s}^{\mu_1 - \epsilon} p(x) e^{sb_s \cdot \mathrm{kl}(x, \mu_1 - \epsilon)} \mathrm{d}x$$

$$\leq O\underbrace{\left( \sum_{s=1}^{\lceil 4V(\log\log T)^3 \rceil} s \right)}_{I_1} + \underbrace{\sum_{s=\lceil 4V(\log\log T)^3 \rceil}^{M} e^{-s\epsilon^2/(2V)}(1 + 4\log T)}_{I_2}, \tag{B.4}$$

where the first inequality is from (A.12) and the last inequality is from (B.3) and (A.13). For term $I_1$, we have $I_1 = O(V^2(\log\log T)^6 + 1)$. Let $\epsilon = 1/\log\log T$, then $M \leq O(V\log T \cdot (\log\log T)^2)$. For $s \geq 4V(\log\log T)^3$, we have $e^{-s\epsilon^2/(2V)} = 1/\log^2 T$. Thus, $I_2 = O(M/\log T) = O(V(\log\log T)^2)$. Therefore,

$$A_3 = O(V^2(\log\log T)^6 + V(\log\log T)^2 + 1). \tag{B.5}$$

From (A.9) and (A.10), we have

$$A_1 + A_2 = O(V(\log\log T)^2).$$

Substituting the bound of $A_1$, $A_2$ and $A_3$ to (A.8), we have

$$\sum_{s=1}^{M} \mathbb{E}_{\widehat{\mu}_{1s}} \left[ \left( \frac{1}{G_{1s}(\epsilon)} - 1 \right) \cdot \mathbb{1}\{\widehat{\mu}_{1s} \in L_s\} \right] \leq A_1 + A_2 + A_3 = O(V^2(\log\log T)^6 + V(\log\log T)^2 + 1).$$

This completes the proof.

## C    Proof of Theorem 4.4 (Gaussian-TS)

The proof of Theorem 4.4 is similar to that of Theorem 4.2. Thus we reuse the notation in the proofs of Theorem 4.2 presented in Sections A and F. However, the sampling distribution $\mathcal{P}$ in Theorem 4.4 is chosen as a Gaussian distribution, and therefore, the concentration and anti-concentration inequalities for Gaussian-TS are slightly different from those used in previous sections. This further affects the results of the supporting lemmas whose proofs depend on the concentration bound of $\mathcal{P}$. In this section, we will prove the regret bounds of Gaussian-TS by showing the existence of counterparts of these lemmas for Gaussian-TS.

### C.1    Proof of the Finite-Time Regret Bound

From Lemma H.1, the Gaussian posterior $\mathcal{N}(\mu, V/n)$ satisfies $\mathbb{P}(\theta \leq \mu - x) \leq e^{-nx^2/(2V)}$. Hence, A.1 also holds for Gaussian-TS. The proof of Lemma A.2 needs to call (4.4) and (4.5). However, the tail bound for Gaussian distribution has a different form. We need to replace Lemma A.2 with the following variant.

**Lemma C.1.** *Let $M$, $G_{1s}(\epsilon)$, and $L_s$ be the same as defined in Lemma A.1. Then, there exists a universal constant $C_3 > 0$ such that*

$$\sum_{s=1}^{M} \mathbb{E}_{\widehat{\mu}_{1s}} \left[ \left( \frac{1}{G_{1s}(\epsilon)} - 1 \right) \cdot \mathbb{1}\{\widehat{\mu}_{1s} \in L_s\} \right] \leq C_3 \cdot \left( \frac{V\log(T\epsilon^2/V)}{\epsilon^2} \right).$$

In Section A, the proof of Lemma A.3 only uses the following property of the sampling distribution: let $X_{is}$ be a sample from $\mathcal{P}(\widehat{\mu}_{is}, s)$ and if $\widehat{\mu}_{is} \leq \mu_1 - \epsilon$, then

$$\mathbb{P}(X_{is} \geq \mu_1 - \epsilon) \leq \exp(-sb_s \cdot \mathrm{kl}(\widehat{\mu}_{is}, \mu_1 - \epsilon)),$$

where the $\mathrm{kl}(\cdot)$ function is defined for Gaussian distribution with variance $V$. For Gaussian distribution, let $X_{is}$ be a sample from $\mathcal{N}(\widehat{\mu}_{is}, V/s)$. Then from Lemma H.1

$$\mathbb{P}(X_{is} \geq \mu_1 - \epsilon) \leq \exp(-s \cdot \mathrm{kl}(\widehat{\mu}_{is}, \mu_1 - \epsilon)) \leq \exp(-sb_s \cdot \mathrm{kl}(\widehat{\mu}_{is}, \mu_1 - \epsilon)),$$

where the last inequality is due to $b_s \leq 1$ The other parts of the proof of the finite-time bound are the same as that of Theorem 4.2 and thus are omitted.

## C.2 Proof of the Asymptotic Regret Bound

The proof of Lemma B.1 needs to call (4.4) and (4.5). However, the tail bound for Gaussian distribution has a different form. We need to replace Lemma A.2 with the following variant.

**Lemma C.2.** *Let $M$, $G_{1s}(\epsilon)$, and $L_s$ be the same as defined in Lemma A.1 and let $\epsilon = 1/\log\log T$. Then,*

$$\lim_{T \to \infty} \sum_{s=1}^{M} \mathbb{E}_{\widehat{\mu}_{1s}} \left[ \left( \frac{1}{G_{1s}(\epsilon)} - 1 \right) \cdot \mathbb{1}\{\widehat{\mu}_{1s} \in L_s\} \right] / \log T = 0.$$

The other parts of asymptotic regret bound are the same as that in Theorem 4.2 and are omitted.

## C.3 Proof of Supporting Lemmas

### C.3.1 Proof of Lemma C.1

Let $Z$ be a sample from $\mathcal{N}(\widehat{\mu}_{1s}, V/s)$ and $\widehat{\mu}_{1s} = \mu_1 + x$. For $x \leq -\epsilon$, applying Lemma H.2 with $z = -\sqrt{s/V}(\epsilon + x) > 0$ yields: for $0 < z \leq 1$,

$$G_{1s}(\epsilon) = \mathbb{P}(Z > \mu_1 - \epsilon) \geq \frac{1}{2\sqrt{2\pi}} \exp\left( -\frac{s(\epsilon + x)^2}{2V} \right). \tag{C.1}$$

Besides, for $z > 1$,

$$G_{1s}(\epsilon) \geq \frac{1}{\sqrt{2\pi}} \frac{z}{z^2 + 1} e^{-\frac{z^2}{2}} \geq \frac{1}{2\sqrt{2\pi} \cdot z} e^{-\frac{z^2}{2}} = \frac{\sqrt{V}}{-2\sqrt{2\pi}\sqrt{s}(\epsilon + x)} \exp\left( -\frac{s(\epsilon + x)^2}{2V} \right). \tag{C.2}$$

Since $\widehat{\mu}_{1s} \sim \mathcal{N}(\mu_1, V/s)$, $x \sim \mathcal{N}(0, V/s)$. Let $p(x)$ be the PDF of $\mathcal{N}(0, V/s)$. Note that $G_{1s}(\epsilon)$ is a random variable with respect to $\widehat{\mu}_{1s}$ and $\widehat{\mu}_{1s} = \mu_1 + x$. We have

$$\mathbb{E}_{\widehat{\mu}_{1s}} \left[ \left( \frac{1}{G_{1s}(\epsilon)} - 1 \right) \cdot \mathbb{1}\{\widehat{\mu}_{1s} \in L_s\} \right] \leq \int_{-\epsilon}^{+\infty} \frac{p(x)}{G_{1s}(\epsilon)} \mathrm{d}x - 1 + \int_{-\epsilon-\alpha_s}^{-\epsilon} \frac{p(x)}{G_{1s}(\epsilon)} \mathrm{d}x$$

$$\leq 1 + \int_{-\epsilon-\alpha_s}^{-\epsilon} \frac{p(x)}{G_{1s}(\epsilon)} \mathrm{d}x$$

$$\leq 1 + \underbrace{\int_{-\epsilon-\alpha_s}^{-\epsilon} p(x) \left( 2\sqrt{2\pi} \cdot \exp\left( \frac{s(\epsilon + x)^2}{2V} \right) \right) \mathrm{d}x}_{I_1}$$

$$+ \underbrace{\int_{-\epsilon-\alpha_s}^{-\epsilon} p(x) \left( 2\sqrt{2\pi}\sqrt{s/V}(-\epsilon - x) \cdot \exp\left( \frac{s(\epsilon + x)^2}{2V} \right) \right) \mathrm{d}x}_{I_2}. \tag{C.3}$$

The second inequality is due to the fact that for $\widehat{\mu}_{1s} \geq \mu_1 - \epsilon$, $G_{1s}(\epsilon) = \mathbb{P}(Z \geq \mu_1 - \epsilon) \geq 1/2$. The last inequality is due to (C.1) and (C.2). For term $I_1$, we have

$$I_1 = \int_{-\alpha_s-\epsilon}^{-\epsilon} \left( 2\sqrt{\frac{s}{V}} \exp\left( \frac{-sx^2}{2V} \right) \exp\left( \frac{s(\epsilon + x)^2}{2V} \right) \right) \mathrm{d}x$$

$$\leq 2\sqrt{\frac{s}{V}}\exp\left(\frac{s\epsilon^2}{2V}\right)\int_{-\infty}^{-\epsilon}\exp(s\epsilon x/V)\mathrm{d}x$$

$$= \frac{2\sqrt{V}e^{-s\epsilon^2/(2V)}}{\sqrt{s\epsilon}}. \tag{C.4}$$

For term $I_2$, we have

$$I_2 \leq \int_{-\alpha_s-\epsilon}^{-\epsilon}\left(2s/V(-\epsilon-x)\exp\left(\frac{-sx^2}{2V}\right)\exp\left(\frac{s(\epsilon+x)^2}{2V}\right)\right)\mathrm{d}x$$

$$\leq 2s/V\exp\left(\frac{s\epsilon^2}{2V}\right)\int_{-\alpha_s-\epsilon}^{-\epsilon}(-\epsilon-x)\exp(s\epsilon x/V)\mathrm{d}x$$

$$\leq 2s/V\exp\left(\frac{-s\epsilon^2}{2V}\right)\int_{-\alpha_s-2\epsilon}^{-2\epsilon}-x\exp(s\epsilon x/V)\mathrm{d}x$$

$$\leq 2e\cdot\exp\left(\frac{-s\epsilon^2}{2V}\right)\alpha_s/\epsilon, \tag{C.5}$$

where the last inequality is due to $h(x) = -x\exp(s\epsilon x/V)$ on $x < 0$ achieve is maximum at $x = -V/(s\epsilon)$. We further obtain that

$$\sum_{s=1}^{M}\mathbb{E}\left[\left(\frac{1}{G_{1s}(\epsilon)}-1\right)\cdot\mathbb{1}\{\widehat{\mu}_{1s}\in L_s\}\right] = 2e\left(\sum_{s=1}^{M}\alpha_s/\epsilon + \sum_{s=1}^{M}\frac{\sqrt{V}}{\sqrt{s\epsilon}} + M\right)$$

$$= 2e\left(\sum_{s=1}^{M}\alpha_s/\epsilon + \int_{s=1}^{M}\frac{\sqrt{V}}{\sqrt{s\epsilon}}\mathrm{d}s + M\right)$$

$$= 2e\left(\sum_{s=1}^{M}\alpha_s/\epsilon + \frac{\sqrt{VM}}{\epsilon} + M\right)$$

$$= 2e\left(\sum_{s=1}^{M}\alpha_s/\epsilon + \frac{V\log(T\epsilon^2/V)}{\epsilon^2}\right). \tag{C.6}$$

Note that

$$\mathrm{kl}(\mu_1-\epsilon-4\sqrt{V\log(T/s)/s},\mu_1) \geq \mathrm{kl}(\mu_1-\epsilon-4\sqrt{V\log(T/s)/s},\mu_1-\epsilon) = 8V\log(T/s)/s, \tag{C.7}$$

where the equality is due to (3.3). Thus, from the definition of $\alpha_s$ in (A.2), we have $\alpha_s \leq 4\sqrt{V\log(T/s)/s}$. For term $\sum_{s=1}^{M}\alpha_s$, we have

$$\sum_{s=1}^{M}\alpha_s/(4\sqrt{V}) \leq \sum_{s=1}^{M}\frac{\sqrt{\log(T/s)}}{\sqrt{s}}$$

$$\leq \sum_{j=0}^{\lceil\log M-1\rceil-1}\sum_{s=\lceil e^j\rceil}^{\lceil e^{j+1}\rceil}\frac{\sqrt{\log(T/e^j)}}{\sqrt{s}}$$

$$\leq \sum_{j=0}^{\lceil\log M-1\rceil}\sqrt{\log(T/e^j)}\int_{e^j}^{e^{j+1}}\frac{1}{\sqrt{s}}\mathrm{d}s + \sum_{j=0}^{\lceil\log M-1\rceil}\frac{\sqrt{\log(T/e^j)}}{e^{j/2}}$$

$$\leq 2\sum_{j=0}^{\lceil\log M-1\rceil}e^{(j+1)/2}\cdot\log(T/e^j)$$

$$\leq 2\sqrt{e}\int_{0}^{\log M}(\log T-x)e^{x/2}\mathrm{d}x + 2\sqrt{eM}\log(T/M)$$

$$= 2\sqrt{e}\left(2\log(e^2T)e^{x/2}-2xe^{x/2}\Big|_{0}^{\log M}\right) + 2\sqrt{eM}\log(T/M)$$

$$= 20\left(\frac{\sqrt{V}\log(T\epsilon^2/V)}{\epsilon}\right),\tag{C.8}$$

where the third and sixth inequality is due to $\sum_{x=a}^{b} f(x) \leq \int_a^b f(x)\mathrm{d}x + \max_{x\in[a,b]} f(x)$ for monotone function $f$. Substituting (C.8) to (C.6), we have that there exists a constant $C_3 > 0$ such that

$$\sum_{s=1}^{M}\mathbb{E}\left[\left(\frac{1}{G_{1s}(\epsilon)}-1\right)\cdot\mathbb{1}\{\widehat{\mu}_{1s}\in L_s\}\right] \leq C_3 \cdot \left(\frac{V\log(T\epsilon^2/V)}{\epsilon^2}\right).$$

This completes the proof.

### C.3.2 Proof of Lemma C.2

The proof of this part is similar to the proof of Lemma C.1. We reuse the notation defined in the Lemma C.1. Recall $Z$ is a sample from $\mathcal{N}(\widehat{\mu}_{1s}, V/s)$. For $\widehat{\mu}_{1s} = \mu_1 - \epsilon/2$, from (H.1)

$$\mathbb{P}(Z \leq \mu_1 - \epsilon) \leq \exp\left(-s\epsilon^2/(8V)\right).\tag{C.9}$$

We have

$$\mathbb{E}_{\widehat{\mu}_{1s}}\left[\left(\frac{1}{G_{1s}(\epsilon)}-1\right)\cdot\mathbb{1}\{\widehat{\mu}_{1s}\in L_s\}\right]$$

$$\leq \int_{-\epsilon/2}^{+\infty}\frac{p(x)}{G_{1s}(\epsilon)}\mathrm{d}x + \int_{-\epsilon}^{-\epsilon/2}\frac{p(x)}{G_{1s}(\epsilon)}\mathrm{d}x - 1 + \int_{-\epsilon-\alpha_s}^{-\epsilon}\frac{p(x)}{G_{1s}(\epsilon)}\mathrm{d}x$$

$$\leq e^{-s\epsilon^2/(8V)}\int_{-\epsilon/2}^{+\infty}p(x)\mathrm{d}x + 2\int_{-\epsilon}^{-\epsilon/2}p(x)\mathrm{d}x + \int_{-\epsilon-\alpha_s}^{-\epsilon}\frac{p(x)}{G_{1s}(\epsilon)}\mathrm{d}x$$

$$\leq e^{-s\epsilon^2/(8V)} + e^{-s\epsilon^2/(8V)} + \underbrace{\int_{-\epsilon-\alpha_s}^{-\epsilon}p(x)\left(2\sqrt{2\pi}\cdot\exp\left(\frac{s(\epsilon+x)^2}{2V}\right)\right)\mathrm{d}x}_{I_1}$$

$$+ \underbrace{\int_{-\epsilon-\alpha_s}^{-\epsilon}p(x)\left(2\sqrt{2\pi}\sqrt{s}(-\epsilon-x)\cdot\exp\left(\frac{s(\epsilon+x)^2}{2V}\right)\right)\mathrm{d}x}_{I_2},\tag{C.10}$$

where the second inequality is due to (C.9) and the fact that for $x \geq -\epsilon$, $G_{1s}(\epsilon) \geq 1/2$, the third inequality is due to $x \sim \mathcal{N}(0, V/s)$ and from (H.1), $\mathbb{P}(x \leq -\epsilon/2) \leq \exp(-s\epsilon^2/(8V))$. Further, we have

$$\sum_{s=1}^{\infty}\exp(-s\epsilon^2/(8V)) \leq \frac{1}{e^{\epsilon^2/(8V)}-1} \leq \frac{8V}{\epsilon^2}.$$

By applying (C.4) and (C.5) to bound term $I_1$ and $I_2$, we obtain

$$\sum_{s=1}^{M}\mathbb{E}_{\widehat{\mu}_{1s}}\left[\left(\frac{1}{G_{1s}(\epsilon)}-1\right)\cdot\mathbb{1}\{\widehat{\mu}_{1s}\in L_s\}\right] \leq \frac{8V}{\epsilon^2} + O\left(\sum_{s=1}^{M}\frac{e^{-s\epsilon^2/(2V)}}{\sqrt{s}\epsilon} + \sum_{s=1}^{M}\frac{e^{-s\epsilon^2/(2V)}\alpha_s}{\epsilon}\right)$$

$$= O\left(\frac{V}{\epsilon^2} + 2\sqrt{\log T}/\epsilon\sum_{s=1}^{\infty}e^{-s\epsilon^2/(2V)}\right)$$

$$= O\left(\frac{V}{\epsilon^2} + \frac{V\sqrt{\log T}}{\epsilon^3}\right),$$

where the first equality is due to (C.7). Let $\epsilon = 1/\log\log T$, we have

$$\lim_{T\to\infty}\sum_{s=1}^{M}\mathbb{E}_{\widehat{\mu}_{1s}}\left[\left(\frac{1}{G_{1s}(\epsilon)}-1\right)\cdot\mathbb{1}\{\widehat{\mu}_{1s}\in L_s\}\right]\bigg/\log T = 0,$$

which competes the proof.

# D Proof of Theorem 4.6 (Bernoulli-TS)

Similar to the proof strategy used in Section C, we will prove the regret bounds of Bernoulli-TS via providing a counterpart of the supporting lemma used in the proof of Theorem 4.6 that depends on the concentration bound of the sampling distribution $\mathcal{P}$.

## D.1 Proof of the Finite-Time Regret Bound

Due to the same reason shown in Section C.1, we only need to replace Lemma A.2 with the following variant. The rest of the proof remains the same as that of Theorem 4.2.

**Lemma D.1.** *Let $M$, $G_{1s}(\epsilon)$, and $L_s$ be the same as defined in Lemma A.1. Let $\epsilon = \Delta/4$. There exists a universal constant $C_3$ such that*

$$\sum_{s=1}^{M} \mathbb{E}_{\widehat{\mu}_{1s}} \left[ \left( \frac{1}{G_{1s}(\epsilon)} - 1 \right) \cdot \mathbb{1}\{\widehat{\mu}_{1s} \in L_s\} \right] \leq C_3 \cdot \left( \frac{\log(T\epsilon^2)}{\epsilon^2} \right).$$

## D.2 Proof of the Asymptotic Regret Bound

We note that Agrawal and Goyal [5] has proved the asymptotic optimality for Beta posteriors under Bernoulli rewards. One can find the details therein and we omit the proofs here.

## D.3 Proof of Lemma D.1

We first define some notations. Let $F_{n,p}^{B}(\cdot)$ denote the CDF, and $f_{n,p}^{B}(\cdot)$ denote the probability mass function of binomial distribution with parameters $n, p$ respectively. We also let $F_{\alpha,\beta}^{beta}(\cdot)$ denote the CDF of the beta distribution with parameters $\alpha, \beta$. The following equality gives the relationship between $F_{\alpha,\beta}^{beta}(\cdot)$ and $F_{n,p}^{B}(\cdot)$.

$$F_{\alpha,\beta}^{beta}(y) = 1 - F_{\alpha+\beta-1,y}^{B}(\alpha - 1). \tag{D.1}$$

Let $y = \mu_1 - \epsilon$. Let $j = S_i(t)$ and $s = T_i(t)$. From (D.1), we have $G_{1s}(\epsilon) = \mathbb{P}(\theta_1(t) > y) = F_{s+1,y}^{B}(j)$. Note that for Bernoulli distribution, we can set $V = 1/4$. Besides,

$$\mathrm{kl}(\mu_1 - \epsilon - 4\sqrt{V\log(T/s)/s}, \mu_1) \geq \mathrm{kl}(\mu_1 - \epsilon - 4\sqrt{V\log(T/s)/s}, \mu_1 - \epsilon) \geq 8V\log(T/s)/s,$$

where the inequality is due to (3.3). Thus, from the definition of $\alpha_s$ in (A.2), we have $\alpha_s \leq 2\sqrt{\log(T/s)/s}$. For $j/s \in L_s$, we have $j/s \geq \mu_1 - \epsilon - \sqrt{\frac{2\log(T/s)}{s}}$. Hence,

$$j \geq ys - 2\sqrt{s\log(T/s)}.$$

Let $\gamma_s = \lceil ys - 2\sqrt{s\log(T/s)} \rceil$. Therefore,

$$\mathbb{E}_{\widehat{\mu}_{1s}} \left[ \left( \frac{1}{G_{1s}(\epsilon)} \right) \cdot \mathbb{1}\{\widehat{\mu}_{1j} \in L_s\} \right] \leq \sum_{j=\gamma_s}^{s} \frac{f_{s,\mu_1}(j)}{F_{s+1,y}^{B}(j)}.$$

In the derivation below, we abbreviate $F_{s+1,y}^{B}(j)$ as $F_{s+1,y}(j)$.
**Case $s < 8/\epsilon$.** From Lemma 2.9 of Agrawal and Goyal [5], we have

$$\sum_{j=1}^{s} \frac{f_{s,\mu_1}(j)}{F_{s+1,y}^{B}(j)} \leq \frac{3}{\epsilon}. \tag{D.2}$$

**Case $s \geq 8/\epsilon$.** We divide the $Sum(\gamma_s, s) = \sum_{j=\gamma_s}^{s} \frac{f_{s,\mu_1}(j)}{F_{s+1,y}^{B}(j)}$ into four partial sums: $Sum(\gamma_s, \lfloor ys \rfloor)$, $Sum(\lfloor ys \rfloor, \lfloor ys \rfloor)$, $Sum(\lceil ys \rceil, \lfloor \mu_1 s - \frac{\epsilon}{2}s \rfloor)$, and $Sum(\lceil \mu_1 s - \frac{\epsilon}{2}s \rceil, \lfloor s \rfloor)$ and bound them respectively. We need the following bounds on the CDF of Binomial distribution [18] [Prop. A.4].
There exists a universal constant $C_4 > 0$ such that for $j \leq y(s+1) - \sqrt{(s+1)y(1-y)}$,

$$F_{s+1,y}(j) \leq C_4 \cdot \left( \frac{y(s+1-j)}{y(s+1)-j} \binom{s+1}{j} y^j (1-y)^{s+1-j} \right); \tag{D.3}$$

and for $j \geq y(s+1) - \sqrt{(s+1)y(1-y)}$,

$$F_{s+1,y}(j) \leq C_4.$$

**Bounding** $Sum(\gamma_s, \lfloor ys \rfloor)$. Let $R = \frac{\mu_1(1-y)}{y(1-\mu_1)}$. Then we have $R > 1$. Using the bounds above, we have for any $j$, there exists constant $C_4$ such that

$$\frac{f_{s,\mu_1}(j)}{F_{s+1,y}(j)} \leq C_4 \cdot \left( \frac{f_{s,\mu_1}(j)}{\frac{y(s+1-j)}{y(s+1)-j}\binom{s+1}{j}y^j(1-y)^{s+1-j}} \right) + C_4 \cdot f_{s,\mu_1}(j)$$

$$= C_4 \cdot \left( \left(1 - \frac{j}{y(s+1)}\right) R^j \frac{(1-\mu_1)^s}{(1-y)^{s+1}} \right) + C_4 \cdot f_{s,\mu_1}(j).$$

This applies that for $s \leq \lfloor ys \rfloor$,

$$\left(1 - \frac{j}{y(s+1)}\right) R^j \frac{(1-\mu_1)^s}{(1-y)^{s+1}} = \frac{y(s+1)-j}{y(1-y)(s+1)} R^{j-ys} R^{ys} \frac{(1-\mu_1)^s}{(1-y)^s}$$

$$= \frac{e^{-s \cdot \mathrm{kl}(y,\mu_1)}}{y(1-y)(s+1)}(y(s+1)-j)R^{j-ys}, \qquad \text{(D.4)}$$

where the last equality is due to the fact for Bernoulli distribution, $\mathrm{kl}(y, \mu_1) = y\log(y/\mu_1) + (1-y)\log((1-y)/(1-\mu_1))$. Next, we prove $(y(s+1)-j)R^{j-ys} \leq \frac{2R}{R-1} + e/\ln 2$. Consider the following two cases.

**Case 1:** $1/\ln R \leq y$. We have

$$(y(s+1)-j)R^{j-ys} = (y(s+1)-j)R^{j-y(s+1)}R^y \leq yR^{-y}R^y \leq 1,$$

where the inequality is due to $xR^{-x}$ is monotone increasing on $x \in (0, 1/\ln R)$ and $y(s+1)-j \geq y(s+1)-ys = y \geq 1/\ln R$.

**Case 2:** $1/\ln R \geq y$. We will divide it into the following three intervals of $R$:
For $R \geq e^2$, we have

$$(y(s+1)-j)R^{j-ys} = (y(s+1)-j)R^{j-y(s+1)}R^y$$

$$\leq \frac{1}{\ln R}R^{-1/\ln R}R^y$$

$$\leq \frac{1}{\ln R}R^{-y}R^y$$

$$\leq \frac{1}{\ln R}$$

$$\leq 1,$$

where the first inequality is due to $xa^{-x}$ achieve its maximum at $1/\ln a$.
For $2 < R < e^2$, we have

$$R^{-1/\ln R}/\ln R \leq 1/(e\ln 2) \Leftrightarrow -1 \leq \ln(\ln R/(e\ln 2)) \Leftrightarrow R \geq 2.$$

Therefore,

$$(y(s+1)-j)R^{j-ys} = (y(s+1)-j)R^{j-y(s+1)}R^y$$

$$\leq \frac{1}{\ln R}R^{-1/\ln R}R$$

$$\leq R/(e\ln 2)$$

$$\leq e/\ln 2.$$

For $1 < R < 2$, we have $\ln R \geq (R-1) - (R-1)^2/2$. Further,

$$\frac{R^{-1/\ln R}}{\ln R} \leq \frac{1}{\ln R} \leq \frac{1}{(R-1)-(R-1)^2/2} \leq \frac{1}{(R-1)(1-(R-1)/2)} \leq \frac{2}{R-1}.$$

We have

$$(y(s+1)-j)R^{j-ys} \leq (y(s+1)-j)R^{j-y(s+1)}R$$

$$\leq \frac{R}{\ln R} R^{-1/\ln R}$$
$$\leq \frac{2R}{R-1}.$$

Combining **Case 1** and **Case 2** together, we have $(y(s+1)-j)R^{j-ys} \leq \frac{2R}{R-1} + e/\ln 2$. Substituting this into (D.4), we have

$$\left(1 - \frac{j}{y(s+1)}\right)R^j \frac{(1-\mu_1)^s}{(1-y)^{s+1}} \leq \frac{e^{-s\cdot \mathrm{kl}(y,\mu_1)}}{y(1-y)(s+1)}\left(\frac{2R}{R-1} + e/\ln 2\right)$$
$$\leq \frac{2\mu_1 e^{-s\cdot \mathrm{kl}(y,\mu_1)}}{y(\mu_1-y)(s+1)} + \frac{8e^{-s\cdot \mathrm{kl}(y,\mu_1)}}{y(1-y)(s+1)}$$
$$\leq \frac{20 e^{-s\cdot \mathrm{kl}(y,\mu_1)}}{\epsilon(s+1)}. \tag{D.5}$$

The second inequality is due to $\frac{R}{R-1} = \frac{\mu_1(1-y)}{\mu_1-y}$. The last inequality is due to

$$\frac{\mu_1}{y} = \frac{\mu_1}{\mu_1 - \epsilon} = \frac{\mu_1}{\mu_1 - \Delta_i/4} \leq 4/3 < 2,$$

and

$$y(1-y) \geq \Delta_i/4(1 - \Delta_i/4) = \epsilon(1-\epsilon) \geq \epsilon/2,$$

where the first inequality is because $y(1-y)$ is decreasing for $y \geq 1/2$ and increasing for $y \leq 1/2$ and $y = \mu_1 - \Delta_i/4 \in [3/(4\Delta_i), 1 - \Delta_i/4]$, since $\mu_1 \in [0,1]$ and $\mu_1 \geq \Delta_i$ by definition, the last inequality is due to the fact $\epsilon = \Delta_i/4 \leq 1/4$. Therefore, we have

$$Sum(\gamma_s, \lfloor ys \rfloor) = \sum_{j=\gamma_s}^{\lfloor ys \rfloor} \frac{f_{s,\mu_1}(j)}{F^B_{s+1,y}(j)}$$
$$\leq C_4 \cdot \left(\sum_{j=\gamma_s}^{\lfloor ys \rfloor}\left(1 - \frac{j}{y(s+1)}\right)R^j \frac{(1-\mu_1)^s}{(1-y)^{s+1}}\right) + C_4 \cdot \sum_{j=1}^{s} f_{s,\mu_1}(j)$$
$$\leq 20 C_4 \cdot \left(\frac{e^{-s\cdot \mathrm{kl}(y,\mu_1)}(ys - \gamma_s)}{\epsilon(s+1)}\right) + C_4$$
$$= \frac{40 C_4 \sqrt{\log(T/s)/s}}{\epsilon} + C_4, \tag{D.6}$$

where the second equality is due to (D.5).

**Bounding** $Sum(\lfloor ys \rfloor, \lfloor ys \rfloor)$ **and** $Sum(\lceil ys \rceil, \lfloor \mu_1 s - \frac{\epsilon}{2}s \rfloor)$**.** From Lemma 2.9 of Agrawal and Goyal [5], we have

$$Sum(\lfloor ys \rfloor, \lfloor ys \rfloor) \leq 3e^{-s\mathrm{kl}(y,\mu_1)} \leq 3e^{-2s\epsilon^2}, \tag{D.7}$$

and there exist a universal constant $C_5 > 0$ such that

$$Sum\left(\lceil ys \rceil, \left\lfloor \mu_1 s - \frac{\epsilon}{2}s \right\rfloor\right) \leq C_5 \cdot e^{-s\epsilon^2/2}. \tag{D.8}$$

**Bounding** $Sum(\lceil \mu_1 s - \frac{\epsilon}{2}s \rceil, s)$**.** For $j \in [\lceil \mu_1 s - \frac{\epsilon}{2}s \rceil, s]$, $F_{s+1,y}(j) \leq C_4$. Hence,

$$Sum(\lceil \mu_1 s - \frac{\epsilon}{2}s \rceil, s) \leq C_4. \tag{D.9}$$

Combining (D.6), (D.7), (D.8) and (D.9) together, we have that for $s \geq 8/\epsilon$, there exists a universal constant $C_6 > 0$ such that

$$\mathbb{E}_{\hat{\mu}_{1s}}\left[\left(\frac{1}{G_{1s}(\epsilon)} - 1\right) \cdot \mathbb{1}\{\hat{\mu}_{1j} \in L_s\}\right] \leq C_6 \cdot \left(1 + e^{-s\epsilon^2/2} + e^{-2s\epsilon^2} + \frac{\sqrt{\log(T/s)/s}}{\epsilon}\right). \tag{D.10}$$

Combining (D.2) and (D.10) together, we have that there exists a universal constant $C_3 > 0$ such that

$$\sum_{s=1}^{M} \mathbb{E}_{\widehat{\mu}_{1s}}\left[\left(\frac{1}{G_{1s}(\epsilon)} - 1\right) \cdot \mathbb{1}\{\widehat{\mu}_{1s} \in L_s\}\right]$$

$$= \sum_{s:1 \leq s < 8/\epsilon} \mathbb{E}_{\widehat{\mu}_{1s}}\left[\left(\frac{1}{G_{1s}(\epsilon)} - 1\right) \cdot \mathbb{1}\{\widehat{\mu}_{1s} \in L_s\}\right]$$

$$+ \sum_{s:s \geq 8/\epsilon} \mathbb{E}_{\widehat{\mu}_{1s}}\left[\left(\frac{1}{G_{1s}(\epsilon)} - 1\right) \cdot \mathbb{1}\{\widehat{\mu}_{1s} \in L_s\}\right]$$

$$\leq \frac{24}{\epsilon^2} + C_6 \cdot \left(M + \sum_{s=1}^{\infty} e^{-2s\epsilon^2} + \sum_{s=1}^{\infty} e^{-s\epsilon^2/2} + \sum_{s=1}^{M} \frac{\sqrt{\log(T/s)/s}}{\epsilon}\right)$$

$$\leq C_3 \cdot \left(\frac{\log(T\epsilon^2)}{\epsilon^2}\right),$$

where the last inequality is due to the fact $\sum_{s=1}^{\infty} e^{-2s\epsilon^2} \leq 1/(e^{2\epsilon^2} - 1) \leq \frac{1}{2\epsilon^2}$ and $\sum_{s=1}^{M} \sqrt{\frac{\log(T/s)}{s}} \leq 80(\epsilon^{-1} \log(T\epsilon^2))$ from (C.8).

## E  Proof of the Minimax Optimality of ExpTS$^+$

In this section, we prove the worst case regret bound of ExpTS$^+$ presented in Theorem 5.1.

### E.1  Proof of the Main Result

**Regret Decomposition:**  For simplicity, we reuse the notations in Section A. Let $S_j = \{i \in [K] \mid 2^{-(j+1)} \leq \Delta_i < 2^{-j}\}$ be the set of arms whose gaps from the optimal arm are bounded in the interval $[2^{-(j+1)}, 2^{-j})$. Define $\gamma = 1/2 \log_2(T/(VK)) - 3$. Then we know that for any arm $i \in [K]$ that $\Delta_i > 4\sqrt{VK/T} = 2^{-(\gamma+1)}$, there must exist some $j \leq \gamma$ such that $i \in S_j$. Therefore, the regret of ExpTS$^+$ can be decomposed as follows.

$$R_\mu(T) = \sum_{i:\Delta_i > 0} \Delta_i \cdot \mathbb{E}[T_i(T)]$$

$$\leq \sum_{i:\Delta_i > 4\sqrt{VK/T}} \Delta_i \cdot \mathbb{E}[T_i(T)] + \max_{i:\Delta_i < 4\sqrt{VK/T}} \Delta_i \cdot T \tag{E.1}$$

$$< \sum_{j < \gamma} \sum_{i \in S_j} 2^{-j} \cdot \mathbb{E}[T_i(T)] + 4\sqrt{VKT}, \tag{E.2}$$

where in the first inequality we used the fact that $\sum_i \mathbb{E}[T_i(T)] = T$, and in the last inequality we used the fact that $\Delta_i < 2^{-j}$ for $\Delta_i \in S_j$. The expected number of times that Algorithm 1 plays arms in set $S_j$ with $j < \gamma$ is bounded as follows.

$$\sum_{i \in S_j} \mathbb{E}[T_i(T)] = |S_j| + \sum_{i \in S_j} \mathbb{E}\left[\sum_{t=K+1}^{T} \mathbb{1}\{A_t = i, E_{i,\epsilon_j}(t)\} + \sum_{t=K+1}^{T} \mathbb{1}\{A_t = i, E_{i,\epsilon_j}^c(t)\}\right]$$

$$= |S_j| + \underbrace{\sum_{i \in S_j} \mathbb{E}\left[\sum_{t=K+1}^{T} \mathbb{1}\{A_t = i, E_{i,\epsilon_j}(t)\}\right]}_{A} + \underbrace{\sum_{i \in S_j} \mathbb{E}\left[\sum_{t=K+1}^{T} \mathbb{1}\{A_t = i, E_{i,\epsilon_j}^c(t)\}\right]}_{B},$$

$$\tag{E.3}$$

where $\epsilon_j > \sqrt{8VK/T}$ is an arbitrary constant.

**Bounding Term $A$:**   Define

$$\alpha_s = \sup_{x \in [0, \mu_1 - \epsilon - R_{\min})} \mathrm{kl}(\mu_1 - \epsilon - x, \mu_1) \leq 4 \log^+(T/(Ks))/s, \tag{E.4}$$

where $\log^+(x) = \max\{0, \log x\}$. We decompose the term $\sum_{i \in S_j} \mathbb{E}\left[\sum_{t=K+1}^{T} \mathbb{1}\{A_t = i, E_{i,\epsilon}(t)\}\right]$ by the following lemma.

**Lemma E.1.** *Let $\epsilon_j = 2^{-j-2}$. Let $M_j = \lceil 16V \log(T\epsilon_j^2/(KV))/\epsilon_j^2 \rceil$. Then, there exists a universal constant $C_2 > 0$,*

$$\sum_{i \in S_j} \mathbb{E}\left[\sum_{t=K+1}^{T} \mathbb{1}\{A_t = i, E_{i,\epsilon_j}(t)\}\right] \leq \sum_{s=1}^{M_j} \mathbb{E}\left[\left(\frac{1}{G_{1s}(\epsilon_j)} - 1\right) \cdot \mathbb{1}\{\widehat{\mu}_{1s} \in L_s\}\right] + \frac{C_2 V K}{\epsilon_j^2},$$

*where $G_{is}(\epsilon) = 1 - F_{is}(\mu_1 - \epsilon)$, $F_{is}$ is the CDF of $\mathcal{P}(\widehat{\mu}_{is}, s)$, and $L_s = \left(\mu_1 - \epsilon - \alpha_s, R_{\max}\right]$.*

Now, we bound the remaining term in Lemma E.1.

**Lemma E.2.** *Let $M_j$, $G_{1s}(\epsilon_j)$, and $L_s$ be the same as defined in Lemma E.1. Then, there exists a universal constant $C_3 > 0$ such that*

$$\sum_{s=1}^{M_j} \mathbb{E}_{\widehat{\mu}_{1s}}\left[\left(\frac{1}{G_{1s}(\epsilon_j)} - 1\right) \cdot \mathbb{1}\{\widehat{\mu}_{1s} \in L_s\}\right] \leq C_3 \cdot \left(\frac{V K \log(T\epsilon_j^2/(KV))}{\epsilon_j^2}\right).$$

Combining Lemma E.1 and Lemma E.2 together, we have

$$A \leq \left(\frac{(C_3 + C_2) V K \log(T\epsilon_j^2/(KV))}{\epsilon_j^2}\right).$$

**Bounding Term $B$:**   We have the following lemma that bounds the second term in (E.3).

**Lemma E.3.** *Let $N_i = \min\{1/(1 - (\mathrm{kl}(\mu_i + \rho_i, \mu_1 - \epsilon_j))/\log(T\epsilon_j^2/V)), 2\}$. For any $\rho_i, \epsilon_j > 0$ that satisfies $\epsilon_j + \rho_i < \Delta_i$, then*

$$\mathbb{E}\left[\sum_{t=K+1}^{T} \mathbb{1}\{A_t = i, E_{i,\epsilon_j}^c(t)\}\right] \leq 1 + \frac{2V}{\rho_i^2} + \frac{V}{\epsilon_j^2} + \frac{N_i \log(T\epsilon_j^2/(VK))}{\mathrm{kl}(\mu_i + \rho_i, \mu_1 - \epsilon_j)}.$$

**Putting it Together:**   Let $\rho_i = \epsilon_j$. Substituting Lemma E.1 and Lemma E.3 to the regret decomposition (E.2), we obtain

$$\begin{aligned}
R_\mu(T) &\leq \sum_{j<\gamma} \sum_{i \in S_j} \epsilon_j \cdot \mathbb{E}[T_i(T)] + 4\sqrt{VKT} \\
&= \sum_{j<\gamma} \frac{(C_3 + C_2 + 64)KV \log(T\epsilon_j^2/(VK))}{\epsilon_j} + \sqrt{VKT} + \sum_{i \geq 2} \Delta_i \\
&= 8(C_3 + C_2 + 64)\sqrt{VKT} \cdot \sum_{n=0}^{\infty} \frac{\log 64 + n \log 2}{2^n} + \sum_{i \geq 2} \Delta_i,
\end{aligned}$$

which completes the proof of the minimax optimality. Therefore, there exists a universal constant $C_1 > 0$ such that

$$R_\mu(T) \leq C_1 \cdot \left(\sum_{i>1} \Delta_i + \sqrt{VKT}\right).$$

### E.2 Proof of Supporting Lemmas

#### E.2.1 Proof of Lemma E.1

The proof of this lemma shares many element with that of Lemma A.1. Let $\mathcal{F}_t = \sigma(A_1, r_1, \cdots, A_t, r_t)$ be the filtration. By the definition of $G_{is}(x)$, it holds that

$$G_{1T_1(t-1)}(\epsilon_j) = \mathbb{P}(\theta_1(t) \geq \mu_1 - \epsilon_j \mid \mathcal{F}_{t-1}). \tag{E.5}$$

Define $\mathcal{E}$ to be the event such that $\widehat{\mu}_{1s} \in L_s$ holds for all $s \in [T]$. The indicator function can be decomposed based on $\mathcal{E}$.

$$\sum_{i \in S_j} \mathbb{E}\left[ \sum_{t=K+1}^{T} \mathbb{1}\{A_t = i, E_{i,\epsilon_j}(t)\} \right]$$

$$\leq T \cdot \mathbb{P}(\mathcal{E}^c) + \sum_{i \in S_j} \mathbb{E}\left[ \sum_{t=K+1}^{T} \left[ \mathbb{1}\{A_t = i, E_{i,\epsilon_j}(t)\} \cdot \mathbb{1}\{\widehat{\mu}_{1T_1(t-1)} \in L_{T_i(t-1)}\} \right] \right]$$

$$\leq \Theta\left( \frac{VK}{\epsilon_j^2} \right) + \sum_{i \in S_j} \mathbb{E}\left[ \sum_{t=K+1}^{T} \left[ \mathbb{1}\{A_t = i, E_{i,\epsilon_j}(t)\} \cdot \mathbb{1}\{\widehat{\mu}_{1T_1(t-1)} \in L_{T_i(t-1)}\} \right] \right], \tag{E.6}$$

where the second inequality is from Lemma A.4 with $b = K$. Let $A'_t = \arg\max_{i \neq 1} \theta_i(t)$. Then

$$\mathbb{P}(A_t = 1 \mid \mathcal{F}_{t-1}) \geq \mathbb{P}(\{\theta_1(t) \geq \mu_1 - \epsilon_j\} \cap \{\exists i \in S_j : A'_t = i, E_{i,\epsilon_j}(t)\} \mid \mathcal{F}_{t-1})$$

$$= \mathbb{P}(\theta_1(t) \geq \mu_1 - \epsilon_j \mid \mathcal{F}_{t-1}) \mathbb{P}\left( \bigcup_{i \in S_j} \{A'_t = i, E_{i,\epsilon_j}(t)\} \right)$$

$$= \mathbb{P}(\theta_1(t) \geq \mu_1 - \epsilon_j \mid \mathcal{F}_{t-1}) \cdot \sum_{i \in S_j} \mathbb{P}(A'_t = i, E_{i,\epsilon_j}(t) \mid \mathcal{F}_{t-1})$$

$$\geq \frac{G_{1T_1(t-1)}}{1 - G_{1T_1(t-1)}} \cdot \sum_{i \in S_j} \mathbb{P}(A_t = i, E_{i,\epsilon_j}(t) \mid \mathcal{F}_{t-1}). \tag{E.7}$$

The first inequality is due to the fact when both event $\{\theta_1(t) \geq \mu_1 - \epsilon\}$ and event $\{\exists i \in S_j : A'_t = i, E_{i,\epsilon_j}(t)\}$ hold, we must have $\{A_t = 1\}$. The first equality is due to $\theta_1(t)$ is conditionally independent of $A'_t$ and $E_{i,\epsilon_j}(t)$ given $\mathcal{F}_{t-1}$. The second equality is due to that these events are mutually exclusive. For the last inequality, let $C = \{\exists i \in S_j : A_t = i, E_{i,\epsilon}(t) \text{ occurs}\}$, $A = \{\exists i \in S_j : A'_t = i, E_{i,\epsilon}(t) \text{ occurs}\}$ and $B = \{\theta_1(t) \leq \mu_1 - \epsilon\}$. Then $A$ and $B$ are conditionally independent given $\mathcal{F}_{t-1}$. Besides, if $C$ happens, then $A_t = i$ and $\theta_i(t) \leq \mu_1 - \epsilon$ for some $i \in S_j$. This implies $\theta_1(t) \leq \mu_1 - \epsilon$. Therefore, if $C$ happens, we must have $A'_t = i, E_{i,\epsilon}(t)$ occurs for some $i \in S_j$ and $\theta_1(t) \leq \mu_1 - \epsilon$. Therefore, $C \subseteq A \cap B$ and

$$\sum_{i \in S_j} \mathbb{P}(A_t = i, E_{i,\epsilon_j}(t) \mid \mathcal{F}_{t-1})$$
$$= \mathbb{P}(\exists i \in S_j : A_t = i, E_{i,\epsilon}(t) \text{ occurs} \mid \mathcal{F}_{t-1})$$
$$= \mathbb{P}(C \mid \mathcal{F}_{t-1})$$
$$\leq \mathbb{P}(A \cap B \mid \mathcal{F}_{t-1})$$
$$\leq \mathbb{P}(A \mid \mathcal{F}_{t-1}) \cdot \mathbb{P}(B \mid \mathcal{F}_{t-1})$$
$$= \mathbb{P}(\exists i \in S_j : A'_t = i, E_{i,\epsilon}(t) \text{ occurs} \mid \mathcal{F}_{t-1}) \cdot \mathbb{P}(\theta_1(t) \leq \mu_1 - \epsilon \mid \mathcal{F}_{t-1})$$
$$= \sum_{i \in S_j} \mathbb{P}(A'_t = i, E_{i,\epsilon}(t) \text{ occurs} \mid \mathcal{F}_{t-1}) \cdot \left( 1 - \mathbb{P}(\theta_1(t) > \mu_1 - \epsilon \mid \mathcal{F}_{t-1}) \right). \tag{E.8}$$

Note that $G_{1T_1(t-1)}(\epsilon) = \mathbb{P}(\theta_1(t) \geq \mu_1 - \epsilon \mid \mathcal{F}_{t-1})$. (E.8) implies the last inequality of (E.7).

Consider two cases. **Case 1:** $t : T_1(t-1) \leq M_j$. We have

$$\mathbb{E}\left[ \sum_{t:T_1(t-1) \leq M_j} \sum_{i \in S_j} \mathbb{P}(A_t = i, E_{i,\epsilon_j}(t)) \right] \leq \mathbb{E}\left[ \sum_{t:T_1(t-1) \leq M_j} \left( \frac{1}{G_{1T_1(t-1)}(\epsilon_j)} - 1 \right) \mathbb{P}(A_t = 1 \mid \mathcal{F}_{t-1}) \right]$$

$$= \mathbb{E}\left[\sum_{t:T_1(t-1)\le M_j}\left(\frac{1}{G_{1T_1(t-1)}(\epsilon_j)}-1\right)\mathbb{1}\{A_t=1\}\right]$$

$$\le \mathbb{E}\left[\sum_{s=1}^{M_j}\left(\frac{1}{G_{1s}(\epsilon_j)}-1\right)\right]. \tag{E.9}$$

The first inequality is from (E.7) The first equality is due to $\mathbb{E}[\mathbb{1}\{A_t=1\}]=\mathbb{P}(A_t=1\mid\mathcal{F}_{t-1})$. For the last inequality, note that due to the indicator function, the summation in first equality is not zero only when $\mathbb{1}\{A_t=1\}=1$. And $\mathbb{1}\{A_t=1\}=1$ further means that we have pulled the best arm (arm 1) at time $t$. Therefore, the summation over all $T_1(t-1)$ conditional on $\mathbb{1}\{A_t=1\}=1$ is equivalent to the summation over $s$, which is the number of pulls of arm 1.

**Case 2:** $t:T\ge T_1(t-1)>M_j$. For this case, we have

$$\mathbb{E}\left[\sum_{t:T_1(t-1)>M_j}^{T}\mathbb{1}\{A_t=i,E_{i,\epsilon_j}(t)\}\right]$$

$$\le \mathbb{E}\left[\sum_{t:T_1(t-1)>M_j}^{T}\mathbb{1}\{\theta_1(t)<\mu_1-\epsilon_j\}\right]$$

$$\le T\cdot\mathbb{P}\big(\exists s>M_j:\widehat{\mu}_{1s}<\mu_1-\epsilon_j/2\big)$$

$$+\mathbb{E}\left[\sum_{t:T_1(t-1)>M_j}\mathbb{P}\Big(\{\theta_1(t)<\mu_1-\epsilon_j\mid\widehat{\mu}_{1T_1(t-1)}\ge\mu_1-\epsilon_j/2\}\Big)\right]$$

$$\le T\cdot e^{-M_j(\mu_1-(\mu_1-\epsilon_j/2))^2/(2V)}+T\cdot e^{-M_j\epsilon^2/(16V)}$$

$$\le \frac{2VK}{\epsilon_j^2}, \tag{E.10}$$

In the first inequality, we use the fact that $\{A_t=i,E_{i,\epsilon}(t)\}\subseteq\{\theta_1(t)<\mu_1-\epsilon\}$. In the second inequality we use the same argument as the third inequality of (A.7). In the third inequality, we use Lemma H.1 and the following results

$$\mathbb{E}\left[\sum_{t:T_1(t-1)>M_j}\mathbb{P}(\theta_1(t)\le\mu_1-\epsilon_j\mid\hat{\mu}_{1T_1(t-1)}\ge\mu_1-\epsilon_j/2)\right]$$

$$\le \mathbb{E}\left[\sum_{t:T_1(t-1)>M_j}1/2e^{-T_1(t-1)b_{T_1(t-1)}\mathrm{kl}(\hat{\mu}_{1T_1(t-1)},\mu_1)}\right]$$

$$\le \mathbb{E}\left[\sum_{t:T_1(t-1)>M_j}e^{-\frac{M_j}{2}\mathrm{kl}(\hat{\mu}_{1T_1(t-1)},\mu_1)}\right]$$

$$\le \mathbb{E}\left[\sum_{t:T_1(t-1)>M_j}e^{-\frac{M_j}{2}\mathrm{kl}(\mu_1-\epsilon_j/2,\mu_1)}\right]$$

$$\le T\cdot e^{-M_j\epsilon_j^2/(16V)},$$

where the first inequality is due to the facts that $\theta_1(t)\sim\mathcal{P}^+(\hat{\mu}_{1T_1(t-1)},T_1(t-1))$, $\hat{\mu}_{1T_1(t-1)}\ge\mu_1-\epsilon_j$, and (4.5), the second inequality is due to the facts that $b_s\ge1/2$ for any $s>1$ and $T_1(t-1)>M_j$, the third inequality is due to $\hat{\mu}_{1T_1(t-1)}\ge\mu_1-\epsilon_j/2$ and the fact that from Proposition 3.2, $\mathrm{kl}(x,\mu_1-\epsilon)$ is increasing for $x>\mu_1-\epsilon/2$, and the last inequality is due to (3.3). Combining (E.6), (E.9), and (E.10) together, we complete the proof of this lemma.

### E.2.2 Proof of Lemma E.2

Let $p(x)$ be the PDF of $\widehat{\mu}_{1s}$ and $\theta_{1s}$ be a sample from $\mathcal{P}(\widehat{\mu}_{1s}, s)$. We have

$$\sum_{s=1}^{M_j} \mathbb{E}_{\widehat{\mu}_{1s}} \left[ \left( \frac{1}{G_{1s}(\epsilon)} - 1 \right) \cdot \mathbb{1}\{\widehat{\mu}_{1s} \in L_s\} \right]$$

$$\leq \underbrace{\sum_{s=1}^{M_j} \left( \int_{\mu_1-\epsilon_j/2}^{R_{\max}} p(x)/\mathbb{P}(\theta_{1s} \geq \mu_1 - \epsilon_j \mid \widehat{\mu}_{1s} = x)\mathrm{d}x - 1 \right)}_{A_1}$$

$$+ \underbrace{\sum_{s=1}^{M_j} \left( \int_{\mu_1-\epsilon_j}^{\mu_1-\epsilon_j/2} p(x)/\mathbb{P}(\theta_{1s} \geq \mu_1 - \epsilon_j \mid \widehat{\mu}_{1s} = x)\mathrm{d}x \right)}_{A_2}$$

$$+ \underbrace{\sum_{s=1}^{M_j} \int_{\mu_1-\epsilon_j-\alpha_s}^{\mu_1-\epsilon_j} \left[ p(x)/\mathbb{P}(\theta_{1s} \geq \mu_1 - \epsilon_j \mid \widehat{\mu}_{1s} = x) \right]\mathrm{d}x,}_{A_2} \tag{E.11}$$

where the inequality is due to the definition of $L_s$.

**Bounding term $A_1$.** Similar to the bounding term $A_1$ in Lemma A.2, we divide $\sum_{s=1}^{M_j}$ into two term, i.e., $\sum_{s=1}^{\lfloor 32V/\epsilon_j^2 \rfloor}$ and $\sum_{s=\lceil 32V/\epsilon_j^2 \rceil}^{M_j}$. We have

$$A_1 = \sum_{s=1}^{M_j} \left( \int_{\mu_1-\epsilon_j/2}^{R_{\max}} \frac{p(x)}{\mathbb{P}(\theta_{1s} \geq \mu_1 - \epsilon_j \mid \widehat{\mu}_{1s} = x)} \mathrm{d}x - 1 \right)$$

$$\leq \frac{32V}{\epsilon_j^2} + \sum_{s=\lceil 32V/\epsilon_j^2 \rceil}^{M_j} \left( \int_{\mu_1-\epsilon_j/2}^{R_{\max}} \frac{p(x)}{\mathbb{P}(\theta_{1s} \geq \mu_1 - \epsilon_j \mid \widehat{\mu}_{1s} = x)} \mathrm{d}x - 1 \right)$$

$$\leq \frac{32V}{\epsilon_j^2} + \sum_{s=\lceil 32V/\epsilon_j^2 \rceil}^{M_j} \left( \frac{1}{1 - e^{-s/2 \cdot \mathrm{kl}(\mu_1-\epsilon_j/2, \mu_1-\epsilon_j)}} - 1 \right)$$

$$\leq \frac{32V}{\epsilon_j^2} + \sum_{s=\lceil 32V/\epsilon_j^2 \rceil}^{M_j} \left( \frac{1}{1 - e^{-s\epsilon_j^2/(16V)}} - 1 \right)$$

$$= \frac{16V}{\epsilon_j^2} + \sum_{s=\lceil 32V/\epsilon_j^2 \rceil}^{M_j} \frac{1}{e^{s\epsilon_j^2/(16V)} - 1}$$

$$\leq \frac{32V}{\epsilon_j^2}, \tag{E.12}$$

For the first inequality, we use the fact that with probability at least $1 - 1/K \geq 1/2$, $\theta_{1s} = \hat{\mu}_{1s} \geq \mu_1 - \epsilon$. For second inequality we use the fact that for $\theta_{1s} = \hat{\mu}_{1s}$, $\theta_{1s} \geq \mu_1 - \epsilon$; for $\theta_{1s} \sim \mathcal{P}$, from (4.5), $\mathbb{P}(\theta_{1s} \geq \mu_1 - \epsilon \mid \widehat{\mu}_{1s} = x) \geq 1 - e^{-sb_s\epsilon^2/(16V)}$. The third inequality is due to (3.3).

**Bounding term $A_2$.** This part is the same as the bounding term $A_2$ in Lemma A.2, thus we omit the details.

**Bounding term $A_3$.** The proof of bounding term $A_3$ is similar to that of proofs in Lemma A.2. We also omit the details. The results are as follows.

$$A_3 \leq K \sum_{s=1}^{M_j} \int_{\mu_1-\epsilon_j-\alpha_s}^{\mu_1-\epsilon_j} p(x)e^{sb_s \cdot \mathrm{kl}(x,\mu_1-\epsilon_j)}\mathrm{d}x \tag{E.13}$$

$$\leq K \sum_{s=1}^{M_j} e^{-s\epsilon_j^2/(2V)} \left( 1 + 4\log(T/(Ks)) \right) \tag{E.14}$$

$$= 16 \left( \frac{VK \log(T\epsilon_j^2/(KV))}{\epsilon_j^2} \right), \tag{E.15}$$

where the first inequality is due to the fact that with probability $1/K$, we sample from $\mathcal{P}$. Substituting the bound of $A_1$, $A_1$ and $A_3$ to (E.11), we have that there exists a universal constant $C_3$ such that

$$\sum_{s=1}^{M_j} \mathbb{E}_{\widehat{\mu}_{1s}} \left[ \left( \frac{1}{G_{1s}(\epsilon_j)} - 1 \right) \cdot \mathbb{1}\{\widehat{\mu}_{1s} \in L_s\} \right] \leq C_3 \cdot \left( \frac{VK \log(T\epsilon_j^2/(VK))}{\epsilon_j^2} \right), \tag{E.16}$$

which completes the proof.

### E.2.3 Proof of Lemma E.3

Similar to (A.16), we can obtain

$$\mathbb{E}\left[ \sum_{t=K+1}^{T} \mathbb{1}\{A_t = i, E_{i,\epsilon_j}^c(t)\} \right] \leq \mathbb{E}\left[ \sum_{s=1}^{T} \mathbb{1}\{G_{is}(\epsilon_j) > V/(T\epsilon_j^2)\} \right] + \frac{V}{\epsilon_j^2}. \tag{E.17}$$

Let $s \geq N_i \log(T\epsilon_j^2/(VK))/\mathrm{kl}(\mu_i + \rho_i, \mu_1 - \epsilon_j)$. Let $s \geq N \log(T\epsilon^2/V)/\mathrm{kl}(\mu_i + \rho_i, \mu_1 - \epsilon)$. Note that

$$\frac{1}{N_i} = \max \left\{ 1 - \mathrm{kl}(\mu_i + \rho_i, \mu_1 - \epsilon_j)/\log(T\epsilon_j^2/V), \frac{1}{2} \right\}.$$

For case $1/N_i = 1/2$, we have

$$b_s \geq 1/2 = 1/N_i. \tag{E.18}$$

For case $1/N_i = 1 - \mathrm{kl}(\mu_i + \rho_i, \mu_1 - \epsilon_j)/\log(T\epsilon_j^2/V)$, we have

$$\begin{aligned} b_s &\geq 1 - 1/s \\ &\geq 1 - \mathrm{kl}(\mu_i + \rho_i, \mu_1 - \epsilon_j)/(N \log(T\epsilon_j^2/V)) \\ &\geq 1 - \mathrm{kl}(\mu_i + \rho_i, \mu_1 - \epsilon_j)/(\log(T\epsilon_j^2/V)) \\ &= 1/N_i, \end{aligned} \tag{E.19}$$

where the second inequality is due to $s \geq N_i \log(T\epsilon_j^2/V)/\mathrm{kl}(\mu_i + \rho_i, \mu_1 - \epsilon_j)$ and the third inequality is due to $N_i > 1$. Let $X_{is}$ be a sample from the distribution $\mathcal{P}^+(\widehat{\mu}_{is}, s)$. Assume $\widehat{\mu}_{is} \leq \mu_i + \rho_i$. Then from definition of $\mathcal{P}^+(\widehat{\mu}_{is}, s)$, with probability $1 - 1/K$, $\widehat{\mu}_{is} \leq \mu_i + \rho_i$; with probability $1/K$, $X_{is}$ is a random sample from $\mathcal{P}(\widehat{\mu}_{is}, s)$. Therefore if $\widehat{\mu}_{is} \leq \mu_i + \rho_i$ and $s \geq N_i$, we have

$$\begin{aligned} \mathbb{P}(X_{is} \geq \mu_1 - \epsilon_j) &\leq \exp(-sb_s\mathrm{kl}(\widehat{\mu}_{is}, \mu_1 - \epsilon_j))/K \\ &\leq \exp(-sb_s\mathrm{kl}(\mu_i + \rho_i, \mu_1 - \epsilon_j))/K \\ &\leq \frac{V}{T\epsilon_j^2}, \end{aligned} \tag{E.20}$$

where the first inequality is from (4.4) and the definition of $\mathcal{P}^+(\mu, n)$, the second inequality is due to the assumption $\widehat{\mu}_{is} \leq \mu_i + \rho_i$, and the last inequality is due to $s \geq N_i \log(T\epsilon_j^2/(VK))/\mathrm{kl}(\mu_i + \rho_i, \mu_1 - \epsilon_j)$ and $b_s \geq 1/N_i$ from (E.18) and (E.19). The rest of proofs are similar to the proofs in Theorem 4.2. Note that when $\mathbb{P}(X_{is} \geq \mu_1 - \epsilon_j) \leq V/(T\epsilon_j^2)$ holds, term $\mathbb{1}\{G_{is}(\epsilon_j) > V/(T\epsilon_j^2)\} = 0$. Now, we check the assumption $\hat{\mu}_{is} \leq \mu_i + \rho_i$ that is needed for (E.20). From Lemma H.1, we have $\mathbb{P}(\widehat{\mu}_{is} > \mu_i + \rho_i) \leq \exp(-s\rho_i^2/(2V))$. Furthermore, it holds that

$$\sum_{s=1}^{\infty} e^{-\frac{s\rho_i^2}{2V}} \leq \frac{1}{e^{\rho_i^2/(2V)} - 1} \leq \frac{2V}{\rho_i^2}, \tag{E.21}$$

where the last inequality is due to the fact $1 + x \leq e^x$ for all $x$. Let $Y_{is}$ be the event that $\widehat{\mu}_{is} \leq \mu_i + \rho_i$ and $m = N_i \log(T\epsilon_j^2/(VK))/\mathrm{kl}(\mu_i + \rho_i, \mu_1 - \epsilon_j)$. We further obtain

$$\mathbb{E}\left[ \sum_{s=1}^{T} \mathbb{1}\{G_{is}(\epsilon) > V/(T\epsilon_j^2)\} \right] \leq \mathbb{E}\left[ \sum_{s=1}^{T} [\mathbb{1}\{G_{is}(\epsilon) > V/(T\epsilon_j^2)\} \mid Y_{is}] \right] + \sum_{s=1}^{T} (1 - \mathbb{P}[Y_{is}])$$

$$\leq \mathbb{E}\left[ \sum_{s=\lceil m \rceil}^{T} [\mathbb{1}\{\mathbb{P}(X_{is} > \mu_1 - \epsilon_j) > V/(T\epsilon_j^2))\} \mid Y_{is}] \right]$$

$$+ \lceil m \rceil + \sum_{s=1}^{T}(1 - \mathbb{P}[Y_{is}])$$

$$\leq \lceil m \rceil + \sum_{s=1}^{T}(1 - \mathbb{P}[Y_{is}])$$

$$\leq 1 + \frac{2V}{\rho_i^2} + \frac{N_i \log(T\epsilon_j^2/V)}{\mathrm{kl}(\mu_i + \rho_i, \mu_1 - \epsilon_j)}, \tag{E.22}$$

where the first inequality is due to the fact that $\mathbb{P}(A) \leq \mathbb{P}(A \mid B) + 1 - \mathbb{P}(B)$, the third inequality is due to (E.20) and the last inequality is due to (E.21). Substituting (E.22) into (E.17), we complete the proof.

# F  Proof of the Asymptotic Optimality of ExpTS$^+$

Now we prove the asymptotic regret bound of ExpTS$^+$ presented in Theorem 5.1.

## F.1  Proof of the Main Result

The proof of the this part shares many elements with finite time regret analysis. In what follows, we bound terms $A$ and $B$, respectively.

**Bounding Term $A$:**  We reuse the Lemma E.1. Then, it only remains term $\sum_{s=1}^{M_j} \mathbb{E}\left[\left(1/G_{1s}(\epsilon) - 1\right) \cdot \mathbb{1}\{\widehat{\mu}_{1s} \in L_s\}\right]$ to be bounded. We bound this term by the following lemma.

**Lemma F.1.** *Let $M_j$, $G_{1s}(\epsilon_j)$, and $L_s$ be the same as defined in Lemma E.1.*

$$\sum_{s=1}^{M_j} \mathbb{E}_{\widehat{\mu}_{1s}}\left[ \left( \frac{1}{G_{1s}(\epsilon_j)} - 1 \right) \cdot \mathbb{1}\{\widehat{\mu}_{1s} \in L_s\} \right] = O(V^2 K(\log\log T)^6 + V(\log\log T)^2 + K).$$

Combining Lemma E.1 and Lemma F.1 together and let $\epsilon = 1/\log\log T$, we have
$$A = O\big((V^2 K(\log\log T)^6 + V(\log\log T)^2 + K\big).$$

**Bounding Term $B$:**  Let $\rho_i = \epsilon_j = 1/\log\log T$. Applying (E.3), we have

$$\mathbb{E}\left[ \sum_{t=K+1}^{T} \mathbb{1}\{A_t = i, E_{i,\epsilon_j}^c(t)\} \right] = O(V\log^2\log T) + \frac{N_i \log(T\epsilon_j^2/V)}{\mathrm{kl}(\mu_i + 1/\log\log T, \mu_1 - 1/\log\log T)}. \tag{F.1}$$

Therefore, we have

$$B = \sum_{i \in S_j} \mathbb{E}\left[ \sum_{t=K+1}^{T} \mathbb{1}\{A_t = i, E_{i,\epsilon_j}^c(t)\} \right]$$

$$\leq O(VK\log^2\log T) + \sum_{i \in S_j} \frac{N_i \log(T/(V\log^2\log T))}{\mathrm{kl}(\mu_i + 1/\log\log T, \mu_1 - 1/\log\log T)}.$$

**Putting It Together:**  Substituting the bound of term $A$ and $B$ into (E.2), we have

$$\sum_{i \in S_j} \mathbb{E}[T_i(T)] = O(V^2 K\log^6\log T + VK\log^2\log T + K) + \sum_{i \in S_j} \frac{N_i \log(T/(V\log^2\log T))}{\mathrm{kl}(\mu_i + 1/\log\log T, \mu_1 - 1/\log\log T)}.$$

Note that for $T \to \infty$, $N_i \to 1$. Therefore,

$$\lim_{T \to \infty} \sum_{i \in S_j} \frac{\mathbb{E}[T_i(T)]}{\log T} = \sum_{i \in S_j} \frac{1}{\mathrm{kl}(\mu_i, \mu_1)}.$$

This completes the proof of the asymptotic regret.

### F.2 Proof of Lemma F.1

The proof of this Lemma shares many elements with the proof of Lemma E.2. We can use the bound of $A_1$ and $A_2$ in Lemma E.2. Let $\epsilon = 1/\log\log T$, we have

$$A_1 + A_2 = O(V(\log\log T)^2).$$

For term $A_3$, from (E.13), we have

$$A_3 \leq K \sum_{s=1}^{M_j} \int_{\mu_1-\epsilon_j-\alpha_s}^{\mu_1-\epsilon_j} p(x) e^{sb_s \cdot \mathrm{kl}(x,\mu_1-\epsilon_j)} \mathrm{d}x.$$

Then, similar to (B.3), we have

$$K \int_{\mu_1-\epsilon_j-\alpha_s}^{\mu_1-\epsilon_j} p(x) e^{sb_s \cdot \mathrm{kl}(x,\mu_1-\epsilon_j)} \mathrm{d}x = O(Ks). \tag{F.2}$$

Let $\epsilon = 1/\log\log T$. By dividing $\sum_{s=1}^{M_j}$ into two terms $\sum_{s=1}^{\lceil 4V(\log\log T)^3 \rceil}$ and $\sum_{s=\lceil 4V(\log\log T)^3 \rceil}^{M_j}$, from (B.4) and (B.5), we have

$$A_3 = O(V^2 K(\log\log T)^6 + VK(\log\log T)^2 + K).$$

Substituting the bound of $A_1$, $A_1$ and $A_3$ to (E.11), we have

$$A = O(V^2 K(\log\log T)^6 + VK(\log\log T)^2 + K).$$

This completes the proof.

## G  Proof of Technical Lemmas

In this section, we present the proofs of the remaining lemmas used in previous sections.

### G.1  Proof of Lemma A.4

Let $\mathrm{kl}_+(x,y) = \mathrm{kl}(x,y)\mathbb{1}(x \leq y)$. We only need to prove

$$\mathbb{P}\left(\exists s \leq f(\epsilon) : \mathrm{kl}_+(\hat{\mu}_{1s}, \mu_1) \geq 4\log(T/(bs))/s\right) = O\left(\frac{bV}{T\epsilon^2}\right).$$

The proof of this step relies on the standard "peeling technique". We have

$$\mathbb{P}\left(\exists s \leq f(\epsilon) : \mathrm{kl}_+(\hat{\mu}_{1s}, \mu_1) \geq 4\log(T/(bs))/s\right)$$

$$\leq \sum_{n=0}^{\infty} \mathbb{P}\left(\exists \frac{f(\epsilon)}{2^{n+1}} \leq s \leq \frac{f(\epsilon)}{2^n} : \mathrm{kl}_+(\hat{\mu}_{1s}, \mu_1) \geq 4\log(T/(bs))/s\right)$$

$$\leq \sum_{n=0}^{\infty} \mathbb{P}\left(\exists \frac{f(\epsilon)}{2^{n+1}} \leq s \leq \frac{f(\epsilon)}{2^n} : \mathrm{kl}_+(\hat{\mu}_{1s}, \mu_1) \geq \frac{4\log(T/(b \cdot f(\epsilon)/2^n))}{M/2^n}\right)$$

$$\leq \sum_{n=0}^{\infty} \exp\left(-\frac{f(\epsilon)}{2^{n+1}} \cdot \frac{4\log(T/(b \cdot f(\epsilon)/2^n))}{f(\epsilon)/2^n} \cdot \right)$$

$$= \sum_{n=0}^{\infty} \exp\left(-2\log(T/(b \cdot f(\epsilon)/2^n))\right)$$

$$\leq \sum_{n=0}^{\infty} \left(\frac{bf(\epsilon)}{T \cdot 2^n}\right)^2, \tag{G.1}$$

where the third inequality is due to Lemma H.1. Note that $f(\epsilon) \leq 32V\log(T\epsilon^2/(bV))/\epsilon^2$. We have

$$\frac{bf(\epsilon)}{T \cdot 2^n} \leq \frac{bf(\epsilon)}{T} \leq 32\log\left(\frac{T\epsilon^2}{bV}\right) \cdot \frac{bV}{T\epsilon^2}.$$

Continue on equation (G.1), we have

$$\sum_{n=0}^{\infty}\left(\frac{bf(\epsilon)}{T\cdot 2^n}\right)^2 \leq \sum_{n=0}^{\infty}\left(\frac{bf(\epsilon)}{T\cdot 2^n}\cdot 32\log\left(\frac{T\epsilon^2}{bV}\right)\cdot\frac{bV}{T\epsilon^2}\right)$$

$$\leq 32^2\sum_{n=0}^{\infty}\left(\frac{bV}{T\epsilon^2\cdot 2^n}\cdot\left(\log\left(\frac{T\epsilon^2}{bV}\right)\right)^2\cdot\frac{bV}{T\epsilon^2}\right)$$

$$\leq 32^2\cdot\frac{2bV}{T\epsilon^2},$$

where the last inequality is due to $(\log(x))^2/x \leq 1$ for $x \geq 1$.

## G.2  Proof of Lemma A.5

We decompose the proof of Lemma A.5 into two cases: Case 1: $p(x)$ is in continues form, and Case 2: $p(x)$ is in discrete form. We first focus on the case that $p(x)$ is in continues form.

Divide the interval $[x_0, x_n]$ into $n$ sub-intervals $[x_0, x_1), [x_1, x_2), \cdots, [x_{n-1}, x_n]$, such that $\mu_1 - \epsilon - \alpha_s = x_0 \leq x_1 \leq \cdots \leq x_{n-1} \leq x_n = \mu_1 - \epsilon - b_s$, $\int_{x_0}^{x_1} q(x)\mathrm{d}x = \int_{x_{i-1}}^{x_i} q(x)\mathrm{d}x$ for all $i \in [n]$. We now define a new function $p_n(x)$. Assume $p_n(x)$ has been defined on $[x_i, x_n]$. We define $p_n(x)$ on $[x_{i-1}, x_i)$ in the following way. We consider two cases.
**Case 1:** $\int_{x_i}^{x_n} p_n(x)\mathrm{d}x + \int_{x_{i-1}}^{x_i} p(x)\mathrm{d}x \geq e^{-s\cdot\mathrm{kl}(x_n,\mu_1)} - e^{-s\cdot\mathrm{kl}(x_{i-1},\mu_1)}$. Then, we define the function $p_n(x) = p(x)$ for $x \in [x_{i-1}, x_i)$.
**Case 2:** $\int_{x_i}^{x_n} p_n(x)\mathrm{d}x + \int_{x_{i-1}}^{x_i} p(x)\mathrm{d}x < e^{-s\cdot\mathrm{kl}(x_n,\mu_1)} - e^{-s\cdot\mathrm{kl}(x_{i-1},\mu_1)}$. Let $\beta = e^{-s\cdot\mathrm{kl}(x_n,\mu_1)} - e^{-s\cdot\mathrm{kl}(x_{i-1},\mu_1)} - \int_{x_i}^{x_n} p_n(x)\mathrm{d}x - \int_{x_{i-1}}^{x_i} p(x)\mathrm{d}x$. Then, define $p_n(x) = p(x) + \beta/(x_i - x_{i-1})$. Hence, for case 2, it holds that

$$\int_{x_{i-1}}^{x_n} p_n(x)\mathrm{d}x = e^{-s\cdot\mathrm{kl}(x_n,\mu_1)} - e^{-s\cdot\mathrm{kl}(x_{i-1},\mu_1)}. \tag{G.2}$$

Let $y_n = x_n = \mu_1 - \epsilon$. For all $i \in [n]$, define $y_i = x_i$ if $\int_{x_i}^{y_n} p_n(x)\mathrm{d}x = e^{-s\cdot\mathrm{kl}(\mu_1-\epsilon-b_s,\mu_1)} - e^{-s\cdot\mathrm{kl}(x_i,\mu_1)}$. Otherwise, define $y_i$ such that

$$\int_{y_i}^{y_n} p_n(x)\mathrm{d}x = e^{-s\cdot\mathrm{kl}(\mu_1-\epsilon-b_s,\mu_1)} - e^{-s\cdot\mathrm{kl}(x_i,\mu_1)}.$$

From the definition, we know

$$x_i \leq y_i. \tag{G.3}$$

Since $p_n(x) \geq p(x)$ holds for any $x \in [x_0, x_n]$ and $g(x) \geq 0$, we have

$$\int_{\mu_1-\epsilon-\alpha_s}^{\mu_1-\epsilon-b_s} p(x)g(x)\mathrm{d}x \leq \int_{\mu_1-\epsilon-\alpha_s}^{\mu_1-\epsilon-b_s} p_n(x)g(x)\mathrm{d}x. \tag{G.4}$$

Note that $g(x)$ is monotone decreasing

$$g'(x) = \left(e^{sb_s\cdot\mathrm{kl}(x,\mu_1-\epsilon)}\right)' = sb_s(\mathrm{kl}(x,\mu_1-\epsilon))'e^{sb_s\cdot\mathrm{kl}(x,\mu_1-\epsilon)}$$

$$= sb_se^{sb_s\cdot\mathrm{kl}(x,\mu_1-\epsilon)}\left(\int_x^{\mu_1-\epsilon}\frac{t-x}{V(t)}\mathrm{d}t\right)'$$

$$= sb_se^{sb_s\cdot\mathrm{kl}(x,\mu_1-\epsilon)}\cdot\int_x^{\mu_1-\epsilon}\frac{-1}{V(t)}\mathrm{d}t$$

$$\leq 0.$$

We have

$$\sum_{i=0}^{n-1}\int_{y_i}^{y_{i+1}} p_n(x)g(x)\mathrm{d}x \leq \sum_{i=0}^{n-1} g(y_i)\int_{y_i}^{y_{i+1}} p_n(x)\mathrm{d}x$$

$$\leq \sum_{i=0}^{n-1} g(x_i) \int_{y_i}^{y_{i+1}} p_n(x)\mathrm{d}x = \sum_{i=0}^{n-1} g(x_i) \int_{x_i}^{x_{i+1}} q(x)\mathrm{d}x$$

$$\leq \sum_{i=0}^{n-1} \int_{x_i}^{x_{i+1}} q(x)g(x)\mathrm{d}x + \sum_{i=0}^{n-1} (g(x_i) - g(x_{i+1})) \int_{x_i}^{x_{i+1}} q(x)\mathrm{d}x$$

$$= \sum_{i=0}^{n-1} \int_{x_i}^{x_{i+1}} q(x)g(x)\mathrm{d}x + (g(x_0) - g(x_n]) \int_{x_0}^{x_1} q(x)\mathrm{d}x$$

$$\leq \sum_{i=0}^{n-1} \int_{x_i}^{x_{i+1}} q(x)g(x)\mathrm{d}x + (g(x_0) - g(x_n))/n. \tag{G.5}$$

In the first inequality, we use the fact that $g(x)$ is monotone decreasing. The second inequality is due to (G.3). The first equality is from the fact that

$$\int_{x_i}^{x_{i+1}} q(x)\mathrm{d}x = e^{-s \cdot \mathrm{kl}(x,\mu_1)} \Big|_{x_i}^{x_{i+1}} = e^{-s \cdot \mathrm{kl}(x_{i+1},\mu_1)} - e^{-s \cdot \mathrm{kl}(x_i,\mu_1)},$$

and the definition of $y_i$ such that

$$\int_{y_i}^{y_{i+1}} p_n(x)\mathrm{d}x = \int_{y_i}^{y_n} p_n(x)\mathrm{d}x - \int_{y_{i+1}}^{y_n} p_n(x)\mathrm{d}x = e^{-s \cdot \mathrm{kl}(x_{i+1},\mu_1)} - e^{-s \cdot \mathrm{kl}(x_i,\mu_1)} = \int_{x_i}^{x_{i+1}} q(x)\mathrm{d}x.$$

The third inequality is due to $\sum_{i=0}^{n-1} \int_{x_{i+1}}^{x_i} q(x)g(x)\mathrm{d}x \geq \sum_{i=0}^{n-1} g(x_i) \int_{x_{i+1}}^{x_i} q(x)\mathrm{d}x$. Now, we focus on bounding term $\int_{x_0}^{y_0} p_n(x)g(x)\mathrm{d}x$. Note that

$$\int_{x_0}^{y_0} p_n(x)g(x)\mathrm{d}x \leq g(x_0) \int_{x_0}^{y_0} p_n(x)\mathrm{d}x.$$

Hence, we only need to bound $\int_{x_0}^{y_0} p_n(x)\mathrm{d}x$. Let

$$n' = \min \left\{ j \in \{0, \cdots, n\} : p_n(x) = p(x) \text{ for all } x \in [x_0, x_j] \right\}.$$

From the definition, for $x < x_{n'}$, $p_n(x) = p(x)$. Besides, for $x \in [x_{n'}, x_{n'+1})$, it must belong to case 2 in the definition of $p_n(x)$. Hence,

$$\int_{x_{n'}}^{x_n} p_n(x)\mathrm{d}x = e^{-s\mathrm{kl}(x_n,\mu_1)} - e^{-s\mathrm{kl}(x_{n'},\mu_1)}.$$

Therefore, $y_{n'} = x_{n'}$. Further, from Lemma H.1, we have

$$\int_{x_0}^{y_{n'}} p_n(x)\mathrm{d}x = \int_{x_0}^{y_{n'}} p(x)\mathrm{d}x \leq \Pr(\widehat{\mu}_{1s} \leq y_{n'}) \leq e^{-s \cdot \mathrm{kl}(y_{n'},\mu_1)}. \tag{G.6}$$

Now, we have

$$\int_{x_0}^{y_0} p_n(x)\mathrm{d}x = \int_{x_0}^{y_{n'}} p_n(x)\mathrm{d}x - \int_{y_0}^{y_{n'}} p_n(x)\mathrm{d}x$$

$$\leq e^{-s\mathrm{kl}(y_{n'},\mu_1)} - \left( \int_{y_0}^{x_n} p_n(x)\mathrm{d}x - \int_{y_{n'}}^{x_n} p_n(x)\mathrm{d}x \right)$$

$$= e^{-s\mathrm{kl}(y_{n'},\mu_1)} - \left( e^{-s\mathrm{kl}(\mu_1-\epsilon,\mu_1)} - e^{-s\mathrm{kl}(x_n,\mu_1)} - e^{-s\mathrm{kl}(\mu_1-\epsilon,\mu_1)} + e^{-s\mathrm{kl}(x_{n'},\mu_1)} \right)$$

$$= e^{-s\mathrm{kl}(x_0,\mu_1)}, \tag{G.7}$$

where the first inequality is due to (G.6), and the last inequality we use the fact $y_{n'} = x_{n'}$. Finally, we have

$$\int_{\mu_1-\epsilon-\alpha_s}^{\mu_1-\epsilon} p(x)g(x)\mathrm{d}x$$

$$\leq \int_{\mu_1-\epsilon-\alpha_s}^{\mu_1-\epsilon} p_n(x)g(x)\mathrm{d}x$$

$$= \sum_{i=0}^{n-1} \int_{y_i}^{y_{i+1}} p_n(x)g(x)\mathrm{d}x + \int_{x_0}^{y_0} p_n(x)g(x)\mathrm{d}x$$

$$\leq \sum_{i=0}^{n-1} \int_{x_i}^{x_{i+1}} q(x)g(x)\mathrm{d}x + (g(x_0) - g(x_n))/n + g(x_0)\int_{x_0}^{y_0} p_n(x)\mathrm{d}x$$

$$\leq \sum_{i=0}^{n-1} \int_{x_i}^{x_{i+1}} q(x)g(x)\mathrm{d}x + (g(x_0) - g(x_n))/n + g(\mu_1 - \epsilon - \alpha_s)e^{-\mathrm{skl}(\mu_1 - \epsilon - \alpha_s, \mu_1)}$$

$$= \int_{\mu_1 - \epsilon - \alpha_s}^{\mu_1 - \epsilon - b_s} q(x)g(x)\mathrm{d}x + g(\mu_1 - \epsilon - \alpha_s)e^{-\mathrm{skl}(\mu_1 - \epsilon - \alpha_s, \mu_1)} + (g(x_0) - g(x_n))/n,$$

where the second inequality is due to (G.5), and the third inequality is due to (G.7). Note that

$$\lim_{n\to\infty} (g(x_0) - g(x_n))/n = 0.$$

Therefore, it holds that

$$\int_{\mu_1 - \epsilon - \alpha_s}^{\mu_1 - \epsilon} p(x)g(x)\mathrm{d}x \leq \int_{\mu_1 - \epsilon - \alpha_s}^{\mu_1 - \epsilon} q(x)g(x)\mathrm{d}x + g(\mu_1 - \epsilon - \alpha_s)e^{-\mathrm{skl}(\mu_1 - \epsilon - \alpha_s, \mu_1)}$$
$$+ (g(x_0) - g(x_n))/n,$$

which completes the Lemma for continues form $p(x)$.

Next, we assume $p(x)$ is in discrete form. Let $z_0, \cdots, z_k$ be all the points such that $p(z_i) > 0$ and $z_i \in [\mu_1 - \epsilon - \alpha_s, \mu_1 - \epsilon]$. We need to prove

$$\int_{\mu_1 - \epsilon - \alpha_s}^{\mu_1 - \epsilon - b_s} q(x)g(x)\mathrm{d}x + e^{-s\cdot\mathrm{kl}(\mu_1 - \epsilon - \alpha_s, \mu_1)} \cdot g(\mu_1 - \epsilon - \alpha_s) \geq \sum_{i=0}^{k} p(z_i)g(z_i).$$

We assume $z_0 \leq z_1 \leq \cdots \leq z_k$. Define $h(z_i) = \int_{z_{i-1}}^{z_i} q(x)\mathrm{d}x$ for $i \in [K]$ and $h(z_0) = \int_{\mu_1 - \epsilon - \alpha_s}^{z_0} q(x)\mathrm{d}x$. We have

$$\int_{\mu_1 - \epsilon - \alpha_s}^{\mu_1 - \epsilon} q(x)g(x)\mathrm{d}x - \sum_{i=0}^{k} p(z_i)g(z_i) \geq \int_{\mu_1 - \epsilon - \alpha_s}^{z_k} q(x)g(x)\mathrm{d}x - \sum_{i=0}^{k} p(z_i)g(z_i)$$

$$\geq \sum_{i=1}^{k} g(z_i)\int_{z_{i-1}}^{z_i} q(x)\mathrm{d}x + g(z_0)\int_{\mu_1 - \epsilon - \alpha_s}^{z_0} q(x)\mathrm{d}x$$

$$- \sum_{i=0}^{k} p(z_i)g(z_i)$$

$$= \sum_{i=0}^{k} (h(z_i) - p(z_i))g(z_i). \tag{G.8}$$

We define $k' = \min\{j \in \{0, \cdots, k\} : p(z_i) - h(z_i) \geq 0 \text{ for all } i \leq j\}$. If such $k'$ does not exist, then $p(z_i) - h(z_i) < 0$ always holds. Hence, $\sum_{i=0}^{k}(h(z_i) - p(z_i))g(z_i) \geq 0$. Otherwise, $k'$ exists. From lemma H.1, we have

$$\sum_{i=0}^{k'} p(z_i) \leq \Pr(\widehat{\mu}_{1s} \leq z_{k'}) \leq e^{-s\cdot\mathrm{kl}(z_{n'}, \mu_1)}. \tag{G.9}$$

Continue on (G.8), we have

$$\sum_{i=0}^{k} (h(z_i) - p(z_i))g(z_i) \geq -\sum_{i=0}^{k'} (p(z_i) - h(z_i))g(z_i)$$

$$\geq -g(\mu_1 - \epsilon - \alpha_s)\sum_{i=0}^{k'} p(z_i) - h(z_i)$$

$$= -g(\mu_1 - \epsilon - \alpha_s)\left(\sum_{i=0}^{k'} p(z_i) - \sum_{i=0}^{k'} h(z_i)\right)$$

$$\geq -g(\mu_1 - \epsilon - \alpha_s)\left(e^{-\mathrm{skl}(z_{k'},\mu_1)} - \int_{\mu_1-\epsilon-\alpha_s}^{z_{k'}} q(x)\mathrm{d}x\right)$$

$$= -g(\mu_1 - \epsilon - \alpha_s) \cdot e^{-\mathrm{skl}(\mu_1-\epsilon-\alpha_s,\mu_1)}, \qquad (\mathrm{G.10})$$

where the first inequality is from the definition of $k'$, the second inequality is due to that $g(\cdot)$ is monotone decreasing, the first equality is due to $p(z_i) - h(z_i) \geq 0$ for all $i \in \{0, 1, \cdots, k'\}$, the third inequality is due to (G.9), and the last equality is due to

$$\int_{\mu_1-\epsilon-\alpha_s}^{z_{k'}} q(x)\mathrm{d}x = e^{-\mathrm{skl}(x,\mu_1)}\bigg|_{\mu_1-\epsilon-\alpha_s}^{z_{k'}} = e^{-\mathrm{skl}(z_{k'},\mu_1)} - e^{-\mathrm{skl}(\mu_1-\epsilon-\alpha_s,\mu_1)}.$$

Combining (G.9) and (G.10), we complete the proof.

# H   Useful Inequalities

**Lemma H.1** (Maximal Inequality [32])**.** *Let $N$ and $M$ be two real numbers in $\mathbb{R}^+ \times \overline{\mathbb{R}^+}$, let $\gamma > 0$, and $\widehat{\mu}_n$ be the empirical mean of $n$ random variables i.i.d. according to the distribution $\nu_{b'^{-1}(\mu)}$. Then, for $x \leq \mu$,*

$$\mathbb{P}(\exists N \leq n \leq M, \widehat{\mu}_n \leq x) \leq e^{-N \cdot \mathrm{kl}(x,\mu)},$$
$$\mathbb{P}(\exists N \leq n \leq M, \widehat{\mu}_n \leq x) \leq e^{-N(x-\mu)^2/(2V)}. \qquad (\mathrm{H.1})$$

*Meanwhile, for every $x \geq \mu$,*

$$\mathbb{P}(\exists N \leq n \leq M, \widehat{\mu}_n \geq x) \leq e^{-N(x-\mu)^2/(2V)}. \qquad (\mathrm{H.2})$$

**Lemma H.2** (Tail Bound for Gaussian Distribution)**.** *For a random variable $Z \sim \mathcal{N}(\mu, \sigma^2)$,*

$$\frac{e^{-z^2/2}}{z \cdot \sqrt{2\pi}} \geq \mathbb{P}(Z > \mu + z\sigma) \geq \frac{1}{\sqrt{2\pi}}\frac{z}{z^2+1}e^{-\frac{z^2}{2}}. \qquad (\mathrm{H.3})$$

*Besides, for $0 \leq z \leq 1$,*

$$\mathbb{P}(Z > \mu + z\sigma) \geq \frac{1}{\sqrt{8\pi}}e^{-\frac{z^2}{2}}.$$

*Proof.* (H.3) is from Abramowitz and Stegun [2]. For the second statement, we have that for $0 \leq z \leq 1$,

$$\mathbb{P}(Z > \mu + z\sigma) \geq \mathbb{P}(Z > \mu + \sigma) \geq \frac{1}{\sqrt{8\pi}}e^{-\frac{z^2}{2}},$$

where the last inequality is due to (H.3). $\qquad \square$