# OpenReview forum: "Finite-Time Regret of Thompson Sampling Algorithms for Exponential Family Multi-Armed Bandits"
_NeurIPS.cc/2022/Conference — NeurIPS 2022 Accept_

### Official Review · Reviewer_vJ2N · 2022-07-04

**Rating:** 6
**Confidence:** 4
**Soundness:** 4 excellent
**Presentation:** 4 excellent
**Contribution:** 4 excellent

**Summary:**

The paper contributes to the growing body of works, where new `optimistic' TS algorithms are designed with sampling distributions that are updated in a non-Bayesian fashion but yields better regret bounds. This paper presents two variations of the Thompson Sampling (TS) algorithm ExpTS and ExpTS$^+$ to solve a K-armed bandit problem. Instead of considering the Bayesian posterior as the sampling distribution, they construct a new sampling distribution that is updated using past rewards and the number of trials for each arm. The sampling distribution in ExpTS is designed to avoid the under-estimation of the optimal arm and over-estimation of the sub-optimal arms. ExpTS$^+$ is like ExpTS but with a greedy step to choose an action from ExpTS. Under the assumption that the rewards are modeled as a distribution from the exponential family of distributions with finite variance, they are able to show that the frequentist regret bound of
1. ExpTS is asymptotically optimal, sub-UCB, and minimax optimal up to a factor of  $\sqrt{\log K}$, and
2. ExpTS$^+$  is minimax optimal and asymptotically optimal.
Moreover, the author improves upon the existing regret bounds for Beta-Bernoulli and Gaussian-Gaussian bandits with their new regret decomposition.

**Questions:**

Major:
79 -It is not clear what is making ExpTS+ minimax optimal but not sub-UCB. Please elaborate with more clarity.

Table 1: Incorrect row 2 (refer Theorem 1.2 in [1] )

94-97: Are these lines correct? The lower bound for TS for Gaussian reward ($O(\sqrt{KT\log K})$) was already established in Theorem 1.4 in [1] (and [2]).

135-136- The statement is confusing. One cannot have Gaussian posterior with a Bernoulli reward.

A.3: I didn't follow the second inequality in A.3. Why there is T in the denominator of the first expression? Please elaborate.

528:  I am not sure, I followed the argument here. Please elaborate

A.7:  How the first term in the third inequality is bounded by $V/\epsilon^2$ and the second term is bounded by the second term in the 4th inequality using 3.3. Please elaborate.

571: I don't think I follow the inequality described here to prove A18.

Minor:

116: Asymptotic - > Asymptotically

130: Beta not beta

146: measure 'of' $p(\theta)$ as ..

299: :loose


Ref..
[1]. Shipra Agrawal and Navin Goyal. Near-optimal regret bounds for Thompson sampling. Journal of the ACM (JACM), 64(5):30, 2017.

**Ethics Review Area:**

["I don’t know"]

**Limitations:**

The authors briefly address some of the limitations of their work in the conclusion.

**Strengths And Weaknesses:**

1. I think the paper is well-written, and the ideas presented are novel and clearly described. In particular, the authors explain the importance of their approach and analysis in Sec. 4.1 and the second para. of Sec. 5.
2. The construction of the sampling distribution (in sec 4.2) with all the nice properties is novel and the key contribution of this paper.
3. The author improves upon the existing regret bounds for Beta-Bernoulli and Gaussian-Gaussian bandits with their new regret decomposition.
4. The proofs in the Appendix A are structured well and looks correct to me, however the authors may address some of my question to enhance clarity. (The length of the paper (including Appendices with proofs) is too long to review for a conference paper.)

---

> ### Author Response · Authors · 2022-08-02
> **Response to Reviewer vJ2N (Part 2)**
>
>
> ---
> **Q6:** 528: I am not sure, I followed the argument here. Please elaborate.
>
> **A6:** Firstly, if $A$ and $B$ are independent and $C\subseteq A\cap B$, then $\mathbb{P}(C)\leq \mathbb{P}(A\cap B)=\mathbb{P}(A)\cdot \mathbb{P}(B)$. In Line 528, let $C=\{A_t=i,E_{i,\epsilon}(t) \ \text{occurs}\}$,  $A=\{A_t'=i,E_{i,\epsilon}(t) \ \text{occurs}\}$ and $B= \{\theta_{1}(t)\leq \mu_1-\epsilon\}$. Then $A$ and $B$ are conditionally independent given $\mathcal{F}_{t-1}$.
>
> Besides, if $C$ happens, then $A_t=i$ and $\theta_{i}(t)\leq \mu_1-\epsilon$. This implies the $\theta_{1}(t)\leq \mu_1-\epsilon$ (otherwise, we will have $A_t\neq i$). Therefore, if $C$ happens, we must have $A_t'=i, E_{i,\epsilon}(t) \ \text{occurs}$ and $\theta_{1}(t)\leq \mu_1-\epsilon$. Therefore, $C\subseteq A\cap B$ and
>
> $ \mathbb{P}(A_t=i, E_{i,\epsilon}(t) \ \text{occurs} \mid \mathcal{F}_{t-1})$
>
>  $=\mathbb{P}(C \mid \mathcal{F}_{t-1})$
>
>   $ \leq  \mathbb{P}(A\cap B \mid \mathcal{F}_{t-1})$
>
>   $\leq \mathbb{P}(A \mid \mathcal{F}_{t-1} )$
>
>  $\quad \cdot\mathbb{P}(B \mid \mathcal{F}_{t-1})$
>
>   $= \mathbb{P}(A_t'=i,E_{i,\epsilon}(t) \ \text{occurs}\mid \mathcal{F}_{t-1}) $
>
>   $\quad \cdot \mathbb{P}(\theta_{1}(t)\leq \mu_1-\epsilon \mid \mathcal{F}_{t-1})$
>
>    $= \mathbb{P}(A_t'=i,E_{i,\epsilon}(t) \ \text{occurs}\mid \mathcal{F}_{t-1})$
>
>   $\quad \cdot \big(1-\mathbb{P}(\theta_{1}(t)> \mu_1-\epsilon \mid \mathcal{F}_{t-1})\big)$.
>
> ---
> **Q7:** A.7: How the first term in the third inequality is bounded by $V/\epsilon^2$ and the second term is bounded by the second term in the 4th inequality using 3.3. Please elaborate.
>
> **A7:** The first question is due to the fact that $M=\lceil 16V\log(T\epsilon^2/V)/\epsilon^2\rceil \geq 8V\log(T\epsilon^2/V)/\epsilon^2$ and thus
> \begin{align*}
>     T\cdot e^{-M(\mu_1-(\mu_1-\epsilon/2))^2/(2V)}=T\cdot e^{-M\epsilon^2/(8V)}\leq \frac{V}{\epsilon^2}.
> \end{align*}
>
> For the second question,  $\theta_{1}(t)$ is sampled from  $\mathcal{P}(\hat\mu_{1T_{1}(t-1)},T_{1}(t-1))$ and from the property of sampling distribution $\mathcal{P}$, we have
> $\sum_{t:T_{1}(t-1)>M}\mathbb{P}(\theta_{1}(t)\leq \mu_1-\epsilon\mid \hat\mu_{1T_{1}(t-1)}\geq \mu_1-\epsilon/2)$
>
> $=\sum_{t:T_{1}(t-1)>M}1/2e^{-T_{1}(t-1)b_{T_{1}(t-1)}\text{kl}(\hat\mu_{1T_{1}(t-1)},\mu_1)}$
>
> $\leq T\cdot e^{-\frac{M}{2}\text{kl}(\hat\mu_{1T_{1}(t-1)},\mu_1)}$
>
> $\leq T\cdot e^{-\frac{M}{2}\text{kl}(\mu_1-\epsilon/2,\mu_1)}$
>
> $\leq T\cdot e^{-M\epsilon^2/(16V)}$
>
> $\leq \frac{V}{\epsilon^2}$,
>
> where first equality is due to $\theta_{1}(t)\sim \mathcal{P}(\hat\mu_{1T_{1}(t-1)},T_{1}(t-1))$ and (4.5), the first inequality is due to the fact that $b_{s}\leq 1/2$ for any $s>1$ and $T_{1}(t-1)>M$, the second inequality is due to $\hat\mu_{1T_{1}(t-1)}\geq \mu_1-\epsilon/2$ and (3.4), the third inequality is due to (3.3), and the last inequality is due to $M\geq {16V\log (T\epsilon^2/V)}/{\epsilon^2}$.
>
> We have revisied the proof in the revision.
>
> ---
> **Q8:** 571: I don't think I follow the inequality described here to prove A18.
>
> **A8:** We guess you mean A17.  There are some typos. Thank you for pointing out this. We have fixed it and revised the proofs as follows. From Line 570 and 571, we have $s\geq N\log(T\epsilon^2/V)/\text{kl}(\mu_i+\rho_i,\mu_1-\epsilon)$ (from the assumption of (A.17)). For term $b_s$, we have $b_s=1-1/s\geq 1/2$ (we have pulled each arm once at Line 2 of Algorithm 1, thus $s$ starts from $2$).
> Note that
> \begin{align}
> \frac{1}{N}=\max\bigg\\{{1-\text{kl}(\mu_i+\rho_i,\mu_1-\epsilon)/\log(T\epsilon^2/V)},\frac{1}{2}\bigg\\}.
> \end{align}
> For case $1/N=1/2$, we have $b_s\geq 1/2=1/N$. For case $1/N={1-\text{kl}(\mu_i+\rho_i,\mu_1-\epsilon)/\log(T\epsilon^2/V)}$, we have
> \begin{align}
> b_s\geq 1-1/s
>    \geq 1-\text{kl}(\mu_i+\rho_i,\mu_1-\epsilon)/(N\log(T\epsilon^2/V))
>    \geq 1-\text{kl}(\mu_i+\rho_i,\mu_1-\epsilon)/(\log(T\epsilon^2/V))
>    = 1/N,
> \end{align}
> where the second inequality is due to $s\geq N\log(T\epsilon^2/V)/\text{kl}(\mu_i+\rho_i,\mu_1-\epsilon)$, the third inequality is due to $N>1$.
>
> **Q9:** Minor: 116: Asymptotic - > Asymptotically
>                130: Beta not beta
>                146: measure 'of'  as ..
>                299: loose
>
> **A9:** Thanks for pointing out the typos, and we have fixed them in the revision.

---

> > ### Comment · Reviewer_vJ2N · 2022-08-06
> > **Proof of Lemma A1**
> >
> > First, I would like to appreciate the authors for their time and effort in responding to all my concerns and incorporating them into the revised version of their manuscript.
> >
> > However, I have a minor question in the proof of Lemma A1:
> >
> > Ref. last two inequalities in line 548 (A.6) for case-1 and 2nd and 3rd inequality of line 550 for case 2.
> >
> > In both the steps above, going from the first inequality to the second requires handling a summation over a random quantity $T_1(t-1)\leq M$ (case 1) and  $T_1(t-1) \geq M$. In both cases, one cannot just ignore the randomness in $T_1(t-1)$ and take it out of the expectation.
> >
> > I am not sure if I am missing some details here. Please elaborate.

---

> > > ### Author Response · Authors · 2022-08-07
> > > **Re: Proof of Lemma A1**
> > >
> > > Thanks for your further questions.
> > >
> > >
> > > The last two inequalities in line 548 of our previous version (which is now line 554 of our new version) are reproduced as follows.
> > >
> > > $\mathbb{E}\bigg[\sum_{t:T_{1}(t-1)<M} \bigg(\frac{1}{G_{1T_{1}(t-1)}(\epsilon)}-1\bigg)\mathbb{P}(A_t=1\mid\mathcal{F}_{t-1}) \bigg]$
> > >
> > > $\leq \mathbb{E}\bigg[\sum_{t:T_{1}(t-1)\leq M} \bigg(\frac{1}{G_{1T_{1}(t-1)}(\epsilon)}-1 \bigg)\cdot \mathbb{1}\\{A_t=1\\} \bigg] \qquad (*)$
> > >
> > > $\leq \mathbb{E}\bigg[\sum_{s=1}^{M}\bigg(\frac{1}{G_{1s}(\epsilon)}-1 \bigg)\bigg]$.
> > >
> > > The first inequality is actually an equality, since $\mathbb{E}[\mathbb{1}\\{A_t=1\\}]=\mathbb{P}(A_t=1)$. Note that due to the indicator function, the summand in $(*)$ is not zero only when $\mathbb{1}\\{A_t=1\\}=1$. And $\mathbb{1}\\{A_t=1\\}=1$ further means that we have pulled the best arm (arm 1) at time $t$. Therefore, the summation over all $T_1(t-1)$ conditional on $\mathbb{1}\\{A_t=1\\}=1$ is equivalent to the summation over $s$, which is the number of pulls of arm 1. These arguments are standard and also can be found in Equation (36.6) in [30] (Bandit Algorithms).
> > >
> > > The second inequality on line 550 of our previous version (which is now line 556 of our new version) is reproduced as follows
> > >
> > > $\mathbb{E}\bigg[\sum_{t:T_{1}(t-1)>M}^{T}\mathbb{1}\\{\theta_{1}(t)<\mu_1-\epsilon\\}\bigg]$
> > > $\leq T\cdot \mathbb{P}(\exists s>M: \hat\mu_{1s}<\mu_1-\epsilon/2)$
> > > $+\mathbb{E}\bigg[\sum_{t:T_{1}(t-1)<M} \mathbb{1}\\{\theta_{1}(t)<\mu_1-\epsilon \\} \mid \hat\mu_{1T_{1}(t-1)\geq \mu_1-\epsilon/2}\bigg]\qquad (**)$.
> > >
> > > Here we decompose the term into two events. Event one: there exists a $t$ with $T_{1}(t-1)>M$, $\hat\mu_{1T_{1}(t-1)}<\mu_1-\epsilon/2$. Event two: for all $T_{1}(t-1)>M$,  $\hat\mu_{1T_{1}(t-1)}\geq\mu_1-\epsilon/2$.
> > >
> > > In the third inequality on line 550 of our previous version (which is now line 556 of our new version), there is a typo where we missed an expectation symbol, which does not affect any of our proof. For a better understanding of readers, we added an additional explanation in our revision to show why the inequalities hold:
> > > we have
> > >
> > > $\mathbb{E}\bigg[\sum_{t:T_{1}(t-1)>M}\mathbb{P}(\theta_{1}(t)\leq \mu_1-\epsilon\mid \hat\mu_{1T_{1}(t-1)}\geq \mu_1-\epsilon/2) \bigg]$
> > >
> > > $=\mathbb{E}\bigg[\sum_{t:T_{1}(t-1)>M}1/2e^{-T_{1}(t-1)b_{T_{1}(t-1)}\text{kl}(\hat\mu_{1T_{1}(t-1)},\mu_1-\epsilon)}\bigg]$
> > >
> > > $\leq \mathbb{E}\bigg[\sum_{t:T_{1}(t-1)>M} e^{-\frac{M}{2}\text{kl}(\hat
> > >     \mu_{1T_{1}(t-1)},\mu_1-\epsilon)}\bigg] $
> > >
> > > $\leq \mathbb{E}\bigg[\sum_{t:T_{1}(t-1)>M} e^{-\frac{M}{2}\text{kl}(\mu_1-\epsilon/2,\mu_1-\epsilon)}\bigg]$
> > >
> > > $\leq T\cdot e^{-M\epsilon^2/(16V)}$
> > >
> > > $\leq \frac{V}{\epsilon^2},$
> > >
> > > where the first equality is due to $\theta_{1}(t)\sim \mathcal{P}(\hat\mu_{1T_{1}(t-1)},T_{1}(t-1))$ and (4.5), the first inequality is due to the fact that $b_{s}\geq 1/2$ for any $s>1$ and $T_{1}(t-1)>M$, the second inequality is due to $\hat\mu_{1T_{1}(t-1)}\geq \mu_1-\epsilon/2$ and (3.4), the third inequality is due to (3.3), and the last inequality is due to $M\geq {16V\log (T\epsilon^2/V)}/{\epsilon^2}$.

---

> > > > ### Comment · Reviewer_vJ2N · 2022-08-08
> > > > **Proof of Lemma A1**
> > > >
> > > > Thank you for addressing my comment.
> > > >
> > > > I am okay with the argument in (\*). However, I still think the second term in (\*\*) is incorrect.
> > > >
> > > > If I understand the authors argument in (\*\*) correctly, the authors decompose the events on the set $\{ \exists t \in \{T_1(t-1)>M\} : \hat \mu_{1T_1(t-1)} < \mu_1 -\epsilon/2 \}$ and its complement, which is $\{ \forall t \in \{T_1(t-1)>M\} : \hat \mu_{1T_1(t-1)} \geq \mu_1 -\epsilon/2 \}$
> > > >
> > > > So, the second term in (\*\*) is a bound on $\mathbb{E}\left[ \left( \sum_{t:T_1(t-1)>M}^T \mathbb{1}(\theta_1 < \mu_1 - \epsilon)  \right) \mathbb{1}\left(\forall t \in \{T_1(t-1)>M\} : \hat \mu_{1T_1(t-1)} \geq \mu_1 -\epsilon/2  \right) \right]$. I am not sure how the authors compute this bound. Even if I assume that  (\*\*) is correct, then how conditioning on the event $\{\hat \mu_{1T_1(t-1)} \geq \mu_1 -\epsilon/2\} $, enables them to compute $\mathbb{E}\left[  \sum_{t:T_1(t-1)>M}^T \mathbb{P}\left(\theta_1 < \mu_1 - \epsilon | \hat \mu_{1T_1(t-1)} \geq \mu_1 -\epsilon/2    \right) \right]$ as a bound on the second term in (\*\*).
> > > >
> > > > I also noted in the revised manuscript that the authors didn't elaborate their argument in the proof of Lemma A1, as they are doing in the response to my comments. Since, there is no page limit for appendix, I would request authors to provided as much explanation as possible for their proofs (even if it is obvious to many readers who are familiar with the subject).

---

> > > > > ### Author Response · Authors · 2022-08-09
> > > > > **Re: Further questions on the proof of Lemma A1**
> > > > >
> > > > > Thanks for your further questions. We added more details in our paper about the steps omitted in our proof. In particular, as is pointed out by you, after the decomposition, the second term should be
> > > > > $\mathbb{E}\Big[\Big(\sum_{t:T_{1}(t-1)>M}^T \mathbb{1}\\{\theta_1(t)<\mu_1-\epsilon \\} \Big) \mathbb{1}\big \\{\forall t\in \{t\mid T_{1}(t-1)>M\}:\hat\mu_{1T_{1}(t-1)}\geq \mu_1-\epsilon/2\big \\} \Big]$.
> > > > >
> > > > > We further bounded it in the following way:
> > > > >
> > > > > $\quad \mathbb{E}\Big[\Big(\sum_{t:T_{1}(t-1)>M}^T \mathbb{1}\\{\theta_1(t)<\mu_1-\epsilon \\} \Big)\mathbb{1}\big\\{\forall t\in \\{t\mid T_{1}(t-1)>M\\}:\hat\mu_{1T_{1}(t-1)}\geq \mu_1-\epsilon/2\big\\} \Big]$
> > > > >
> > > > > $=\mathbb{E}\Big[\sum_{t:T_{1}(t-1)>M}^T \Big(\mathbb{1}\\{\theta_1(t)<\mu_1-\epsilon\\}\cdot \mathbb{1}\big\\{\forall t\in \\{t \mid T_{1}(t-1)>M\\}: \hat\mu_{1T_{1}(t-1)}\geq \mu_1-\epsilon/2\big\\}\Big) \Big]$
> > > > >
> > > > > $\leq\mathbb{E}\Big[\sum_{t:T_{1}(t-1)>M}^T \big(\mathbb{1}\\{\theta_1(t)<\mu_1-\epsilon\\}\cdot \mathbb{1}\big\\{\hat\mu_{1T_{1}(t-1)}\geq \mu_1-\epsilon/2\big\\}\big) \Big]$
> > > > >
> > > > > $=\mathbb{E}\Big[\sum_{t:T_{1}(t-1)>M}^T \mathbb{1}\\{\theta_1(t)<\mu_1-\epsilon,\hat\mu_{1T_{1}(t-1)}\geq \mu_1-\epsilon/2\\} \Big]$,
> > > > >
> > > > > where the inequality is due to the fact $\mathbb{1}\big\\{\forall t\in T_{1}(t-1)>M: \hat\mu_{1T_{1}(t-1)}\geq \mu_1-\epsilon/2\big\\}\leq\mathbb{1}\big\\{\hat\mu_{1T_{1}(t-1)}\geq \mu_1-\epsilon/2\big\\}$ for any $t\in \\{t\mid T_{1}(t-1)>M\\}$. The above result is now presented as the second term in (**). Note that there was a typo about the conditional expectation in our previous presentation, which we have fixed in our revision.
> > > > >
> > > > > Based on this result, we can further have the following result:
> > > > >
> > > > > $\quad \mathbb{E}\Big[\sum_{t:T_{1}(t-1)>M}^T \mathbb{1}\big\\{\theta_1(t)<\mu_1-\epsilon,\hat\mu_{1T_{1}(t-1)}\geq \mu_1-\epsilon/2\big\\} \Big]$
> > > > >
> > > > > $=\mathbb{E}\Big[\sum_{t:T_{1}(t-1)>M}^T \mathbb{E}\Big[\mathbb{1}\big\\{\theta_1(t)<\mu_1-\epsilon,\hat\mu_{1T_{1}(t-1)}\geq \mu_1-\epsilon/2\big\\} \mid T_{1}(t-1)\Big]\Big]$
> > > > >
> > > > > $=\mathbb{E}\Big[\sum_{t:T_{1}(t-1)>M}^T \mathbb{P}(\theta_1(t)<\mu_1-\epsilon,\hat\mu_{1T_{1}(t-1)}\geq \mu_1-\epsilon/2) \Big]$
> > > > >
> > > > > $\leq \mathbb{E}\bigg[\sum_{t:T_{1}(t-1)>M}^T \mathbb{P}(\theta_1(t)<\mu_1-\epsilon \mid \hat\mu_{1T_{1}(t-1)}\geq \mu_1-\epsilon/2) \bigg]$,
> > > > >
> > > > > where the inequality is due to  $\mathbb{P}(A, B)=\mathbb{P}(B)\cdot \mathbb{P}(A\mid B)\leq \mathbb{P}(A\mid B)$. Then the proof proceeds with the explanation in our previous response to you.
> > > > >
> > > > > We have also added all the clarification made in our responses to you to the revision now. Thanks for the suggestion.

---

> ### Author Response · Authors · 2022-08-02
> **Response to Reviewer vJ2N (Part 1)**
>
> Thanks for your helpful comments. Based on the comments, we have thoroughly revised the paper, with major changes highlighted in ${\color{blue}{blue}}$. We now address your questions point by point as follows.
>
> ---
> **Q1:** 79 -It is not clear what is making ExpTS+ minimax optimal but not sub-UCB. Please elaborate with more clarity.
>
> **A1:** According to the first equality in Line 741, we can show that the finite-time bound for ExpTS+ is
> \begin{align}
>     R_{\mu}(T)=O\bigg(\sum_{i>1}\Delta_i+\sum_{i>1}\frac{{\color{red} K}V\log(T\Delta_i^2/(VK))}{\Delta_i}\bigg).
> \end{align}
> Note that the factor $4\sqrt{VKT}$ in the frist equality of Line 741 of our original version is removed because it only appears when there exists an arm $i$ such that its gap satisfies $\Delta_i=O(\sqrt{VK/T})$ and when this happens, the regret in the above equality is $O(\sqrt{VKT})$, which is in the same order as $4\sqrt{VKT}$. This regret is not sub-UCB due to the additional factor $\color{red} K$ in the regret.
>
> Recall that the sampling distribution for ExpTS+ is $\mathcal{P}^+(\mu,n)$. The parameter $\theta$ is sampled from $\mathcal{P}^+$ in the following way: $\theta=\mu$ with probability $1-1/K$, and $\theta\sim \mathcal{P}(\mu,n)$ with probability $1/K$. When the best arm is underestimated and its empirical mean is lower than the samples from the suboptimal arms, to pull the best arm in the current step,  we need to choose the $\theta\sim \mathcal{P}(\hat\mu_{1}(t),T_{1}(t))$ (otherwise $\theta=\hat\mu_1(t)$, which is lower than the samples from suboptimal arms). Compared with sampling $\theta\sim \mathcal{P}$ in ExpTS, the probability that we choose $\theta$ from $\mathcal{P}$ is reduced by $K$ times in $\mathcal{P}^+$, which will lead to an additional factor $K$ in the regret bound.
>
> ---
> **Q2:** Table 1: Incorrect row 2 (refer Theorem 1.2 in [1])
>
> **A2:** The second row is for Theorem 1.3 in [5].
>
> ---
> **Q3:** 94-97: Are these lines correct? The lower bound for TS for Gaussian reward ($\Omega(\sqrt{KT\log K})$) was already established in Theorem 1.4 in [1] (and [2]).
>
> **A3:** In Line 94-97, we say
>
> * Jin et al. [18] (Lemma 5 in their paper) show that for Gaussian reward distributions, Gaussian-TS has a regret bound at least in the order of $\Omega(\sqrt{KT\log T})$ if the standard regret decomposition in the existing analysis of Thompson sampling [5, 28, 18] is adopted.
>
> Our statement shows that if the standard regret decomposition (Eq. (4.1)) is used, the lower bound ($\Omega(\sqrt{KT\log K})$) cannot be matched. We also note that Theorem 1.3 of [5] (Near-optimal Regret bounds for Thompson Sampling) can achieve the worst-case regret $O(\sqrt{KT\log K})$ with Gaussian posterior (Theorem 1.3). However, they achieve this by using Gaussian posterior for Bernoulli reward models, which further loses the asymptotic optimality. This does not contradicts with our statement, since our statement only holds for Gaussian rewards with Gaussian posteriors.  In our regret analysis, we use a new regret decomposition and thus can prove the regret of Gaussian-TS (Bernoulli-TS) is $O(\sqrt{KT\log K})$  and is asymptotically optimal.
>
> ---
> **Q4:** 135-136- The statement is confusing. One cannot have Gaussian posterior with a Bernoulli reward.
>
> **A4:** The reward distribution and posterior can be different in Thompson sampling. When the reward distribution is Bernoulli, we can also use the Gaussian prior in TS and hence the posterior distribution is also Gaussian. Please see Algorithm 2 in [5].
>
> Algorithm 2 in [5] is proposed for improving the worst-case regret. The worst-case regret for Algorithm 2 in [5] is $O(\sqrt{KT\log K})$. However, using Gaussian posterior for Bernoulli rewards, Algorithm 2 in [5]  loses asymptotic optimality. In our paper, we show that Bernoulli-TS (using Beta prior) itself has regret $O(\sqrt{KT\log K})$ and is also asymptotically optimal.
>
>
> ---
> **Q5:** A.3: I didn't follow the second inequality in A.3. Why there is T in the denominator of the first expression? Please elaborate.
>
> **A5:** Thank you for pointing out this typo. We have removed it in the revision.

---

### Official Review · Reviewer_XukM · 2022-07-10

**Rating:** 3
**Confidence:** 4
**Soundness:** 2 fair
**Presentation:** 3 good
**Contribution:** 2 fair

**Summary:**

This paper considers stochastic sampling in the multi-armed bandit problem. This problem involves K arms and the goal of the algorithm is to draw the arm of the largest mean as many times as possible. A class of algorithms called Thompson sampling, which conducts a greedy assuming that the posterior is correct, is known to be efficient. This paper considers a version of Thompson sampling with an incorrect posterior.

To argue the advantages of the proposed algorithm, this paper considers three criteria for good algorithms. Namely,
* Asymptotic optimality
* Finite-time guarantee (sub-UCB, minimax)
* anytime algorithm

The proposed algorithm ExpTS is described in Table 1, which samples from the posterior based on a distribution that is inspired by the large deviation asymptotics (4.3). ExpTS+ is a version of ExpTS that mixes greedy and ExpTS.

Class of distributions:
This paper considers a one-parameter exponential family distribution (if I understand correctly, the setting is the same as [Cappe et al. 2013]).

[Cappe et al. 2013] "Kullback-Leibler upper confidence bounds for optimal sequential allocation" Olivier Cappe, Aurelien Garivier, Odalric-Ambrym Maillard, Remi Munos, Gilles Stoltz

The paper is well-written. In my view, there are several issues in this paper that I discuss below. In particular, some of the setup and terms are different from the ones generally used in the community (and wrong in some ways). Besides, the algorithm modifies the posterior without addressing the corresponding anti-concentration of the original TS, which does not contribute to the understanding of the Thompson sampling itself. Given these, I think the paper requires some revision.

Issues
* Incorrect mathematical definition: I guess "finite-time" is used in a wrong way. If I understand correctly, sub-UCB (1.2) bound and O(\sqrt{KT}) regret bounds are not finite-time (Landau notation is asymptotic). Usually finite-time analysis refers to the bound where all the terms are explicitly written as functions of model parameters. Moreover, I do not think most (all?) of TS analysis does not have a sub-UCB bound in the sense of (1.2). For example, if there is a term that is 1/Delta_i^2, then it is not sub-UCB because it diverges faster than (1.2) when we fix T and takes a small Delta).
* The tradeoff relation is not very clear: the asymptotic rate on the top of log T is different in (1.1) and (1.2), which are not in a tradeoff relation. A more natural bound is a finite-time version of (1.1). There may be a tradeoff between minimax regret and the relative-divergence bound of (1.1).
* Comparison with existing results: [23] consider TS for 1-parameter exponential family and showed its asymptotical optimality (which I think is one of the closest papers to this). They stated that "Our result is asymptotic, but the only stage where the constants are not explicitly derivable from knowledge of F, T , and θ0 is in Lemma 9." I guess this bound is from a limited understanding of the anti-concentration property of TS. Authors should mention whether this bound can be made finite-time or if there need a modification such as ExpTS that is targeted to improve the anti-concentration property.
* No simulation: The computational aspect of the proposed algorithm is not mentioned (nor simulations), which limits the utility of the proposed algorithm. The sampling method from the distribution is mentioned (L258-L266) and it looks like the simulation is doable.

## At the end of the discussion phase:

A least agreement is that the minimax optimality is the most important part of this paper. I still think there are many concerns remain in the current version of paper, which I write in the following.

For clarification, the most relevant papers are (references are current version)
* [25] "Thompson sampling for 1-dimensional exponential family bandits." Korda et al. -> Asymptotic optimality of Thompson sampling for 1-dim exponential family
* [20] "MOTS: Minimax Optimal Thompson Sampling" Jin et al. -> Asymptotic + minimax optimality of a version of TS for Gaussian bandits
* [5] "Near-optimal regret bounds for Thompson sampling" Agrawal et al. -> TS is not minimax optimal up to $\sqrt{\log K}$

Given these results, the novelty of this paper is
* A version of TS, applicable to V-bounded 1-dim exponential family that is asymototical+minimax optimal
* Finite-time (+ sub-UCB) analysis of it.

I consider the contributions are incremental given such an asymptotical analysis is already known in [25] for TS and minimax improvement is previously done in [20] for Gaussian by "inflating" posterior just like this paper, but it is okay if it were properly presented and complemented by experimental results.

For me, there remain concerns on the paper. Major ones of them are as follows.

* Confusion on the analysis and performance of the algorithm: Table 1 is ill-defined if I understand each notion correctly. The authors place "Minimax Ratio" and "Sub-UCB" (even though these are defined in terms of asymptotics) inside "Finite-Time Regret". In fact, this is not even a comparison of algorithms. Finite-time or asymptotic is the property of *an analysis* (i.e., giving all constants explicitly or not), whereas all other properties are of *an algorithm*.

* Limited characterization of the minimax property. [5] considers a misspecified instance where the rewards are fixed rather than drawn from the anticipated distributions, so this is not a lower bound in the true sense. A bit larger bound of $O(\sqrt{KT \log T})$ derived from the process of upper-bounding the regret; the remark "L223: using Gaussian posteriors will unavoidably suffer
from the lower bound of $\Omega(\sqrt{KT \log T})$" needs a fix (this is not a lower bound).

* No simulations are provided: p2 motivates the finite-time sub-UCB bound is important in the practical value of $T$, but there is no simulation provided even though "We show that sampling from P is tractable (Line 277)". I think deriving a minimax example where this algorithm is expected to shine is easy ($\mu_1 = \sqrt{K/T}, \mu_2-\mu_K = 0$?).

* Possibly no tradeoffs. They consider "there is a tradeoff between minimax optimal regret and sub-UCB regret" but open. If I understand correctly, if there is a tradeoff between $A$ and $B$, then $A \cap B = \emptyset$ (or show some evidence that $A \cap B$ is unlikely). If it is open then it is not a tradeoff. Regarding the boundedness of $V$, their explanation does not make sense at least for the exponential distribution.

* Lack of comparison on the existing theoretical results: Limited discussion on connecting the results with relevant papers. Regarding whether the finite-time analysis of TS can be done for [25] (only single term is asymptotic in [25]), they commented "The idea of [25] is totally different from ours." and did not provide any discussion on the most closely related paper. During the answer A9 (optimal algorithm for 1-parameter exponential family), they did not aware of existing literature in 2015.

**Questions:**

* Does the original TS has plausible anti-concentration? Lemma 9 in [23] is essentially unremovable (which motivates the introduction of ExpTS) or not? Do ExpTS and original TS match for some limit?
* Tradeoff relation: Among the properties considered in this paper (asymptotic optimality, finite-time bounds, anytime property), which of them are in the tradeoff relations?
* How well ExpTS perform in practice (with some synthetic/real data simulations)?

* Is there any reason that one cannot demonstrate the same rate as the asymptotic rate?
>L227: In this paper, we study the general exponential family of reward distribution, which has no closed form. Thus we cannot obtain a tight concentration bound for µˆ1s as in special cases such as Gaussian or Bernoulli rewards.
* I think having closed form is not directly related to the tightness of the concentration inequality. (probably the author cannot derive the concentration inequality of KL-divergence rate for some hardness)

**Limitations:**

* Ethically sound.
* No simulations on the performance of the proposed method.

**Strengths And Weaknesses:**

Strengths
* Algorithm and related analysis: Showed that sampling from a (mock-)posterior can achieve optimality as well as finite-time exponential concentration, not only Bernoulli and Gaussian but also some class of exponential family.
* Minimax optimality: Discussion on minimax regret of TS for exponential distribution bandits

Weaknesses
* Characterization needs some fix
* Limited contribution to the understanding of TS for exponential bandits, on top of related work [18,23].
* Some gap between what the community wants and what this paper has done

---

> ### Author Response · Authors · 2022-08-02
> **Response to Reviewer XukM (Part 2)**
>
> ---
> **Q7:** Does the original TS has plausible anti-concentration? Do ExpTS and original TS match for some limit?
>
> **A7:** We are not aware of any anti-concentration bounds for the general exponential family of reward distributions.  ExpTS is always better than the original TS in [4] and [5] (see Table 1 for details). However, in this paper we have improved the regret bound for Gaussian-TS and Bernoulli-TS. As shown in our Theorem 4.2, Gaussian-TS matches ExpTS for Gaussian rewards; as shown in Theorem 4.4, Bernoulli-TS matches ExpTS for Bernoulli rewards.
>
>
> ---
> **Q8:** Tradeoff relation: Among the properties considered in this paper (asymptotic optimality, finite-time bounds, anytime property), which of them are in the tradeoff relations?
>
> **A8:** There is no tradeoff among these properties. They are different metrics that are used in different applications. Moreover, ExpTS is anytime and can simultaneously achieve asymptotic optimality, sub-UCB, and minimax optimality up to a factor $\sqrt{\log K}$ for one exponential family of reward distributions.
>
>
> ---
> **Q9:** Is there any reason that one cannot demonstrate the same rate as the asymptotic rate?
>
> **A9:** To the best of our knowledge, no algorithm can give regret bound
>     $R_{\mu}(T)=O\bigg(\sum_{i>1}\frac{\Delta_i\cdot \log T}{\text{kl}(\mu_i,\mu_1)}\bigg)$ for general exponential family of reward distributions.
> The reason is that the regret bound can be written as $O\bigg(\sum_{i>1}\frac{\Delta_i\log T}{\text{kl}(\mu_i,\mu_1-\epsilon_i)}\bigg)$, where the $\epsilon_i$ is the estimation error of the best arm. Then, for asymptotic regret, we can set $\epsilon_i=1/\log \log T$. For $T\rightarrow \infty$, $\epsilon_i=0$ and the regret is $O\bigg(\sum_{i>1}\frac{\Delta_i\log T}{\text{kl}(\mu_i,\mu_1)}\bigg)$. However, for finite-time analysis, $\epsilon_i$ cannot be ignored. Usually, we choose $\epsilon_i=\Delta_i/c$, where $c$ is some constant. However, choosing different constant $c$ will lead to the different regret. Therefore, for simplicity, we would like to use lower bound of $\text{kl}(x,y)$, i.e., $(x-y)^2/(2V)$ to replace. As a result, we consider the sub-UCB $R_{\mu}(T)=O\bigg(\sum_{i>1}\frac{V \log T}{\Delta_i}\bigg)$, following the convention in the literature ([27, Refining the confidence level for optimistic bandit strategies.]   cited in our paper).
>
>
> ---
> **Q10:** I think having closed form is not directly related to the tightness of the concentration inequality. (probably the author cannot derive the concentration inequality of KL-divergence rate for some hardness)
>
> **A10:** We are not aware of any tight anti-concentration bounds that hold for general exponential family of distributions. We will appreciate it a lot if any of these results could be suggested.

---

> > ### Comment · Reviewer_XukM · 2022-08-04
> > **On A9**
> >
> > > A9: no algorithm can give regret bound
> >
> > For some classes of the one-parameter exponential family, the KL-UCB paper (AS version, below) has this bound. It would be great if you point out the difference between the setting of this paper and the KL-UCB paper.
> >
> > KULLBACK-LEIBLER UPPER CONFIDENCE BOUNDS FOR OPTIMAL SEQUENTIAL ALLOCATION Olivier Cappé, Aurélien Garivier, Odalric-Ambrym Maillard, Rémi Munos and Gilles Stoltz

---

> > > ### Author Response · Authors · 2022-08-07
> > > **Re: On A9**
> > >
> > > Thank you for pointing out this Annals of Statistics version [1] of the KL-UCB algorithm (reference [14] in our paper), which we were not aware of before! The nonasymptotic bound of Theorem 2 in [1] is $\sum_{i>1}\frac{\Delta_i}{\text{kl}(\mu_i,\mu_1)}+O((\log T)^{4/5}\log \log T)$, which is achieved under the assumption that the reward distribution is supported in $[0,1]$. Our paper considers the general exponential family reward distribution, which might not have a finite bounded support.
> > >
> > > [1] KULLBACK-LEIBLER UPPER CONFIDENCE BOUNDS FOR OPTIMAL SEQUENTIAL ALLOCATION Olivier Cappé, Aurélien Garivier, Odalric-Ambrym Maillard, Rémi Munos and Gilles Stoltz
> > >
> > > We have added the discussion about their work in our revised paper.

---

> > > > ### Comment · Reviewer_XukM · 2022-08-07
> > > > **Re: On A9**
> > > >
> > > > Theorem 1 in [1] is for some class of one-parameter exponential family, non-asymptotic (whereas Thm 2 is for any dist with bounded support). If you allow asymptotics, then Lai and Robbins 1985 has such a divergence-based bound. I am not fully sure how general they are and how you define a general distribution, though...

---

> > > > > ### Author Response · Authors · 2022-08-07
> > > > > **Re: On A9**
> > > > >
> > > > > Thanks for your further question. For Theorem 1 in [1], it is not clear whether it achieves the regret $O(\sum_{i>1}\frac{\Delta_i\log T}{\text{kl}(\mu_i,\mu_1)})$ mentioned by your previous comment, since the regret bound in Theorem 1 includes a term $\sum_{i>1}8\sigma^2_{i,\star}\big(\text{kl}'(\mu_i,\mu_1)/\text{kl}(\mu_i,\mu_1) \big)^2$, where $\text{kl}'(\cdot,\mu_1)$ denotes the derivative of $\text{kl}(\cdot,\mu_1)$ and $\sigma^2_{i,\star}=\max\{Var(\nu_{\theta}):\mu_i\leq \mathbb{E}(\nu_{\theta})\leq \mu_1\}$. The order of $\text{kl}'(\mu_i,\mu_1)$ is dependent of the specific distribution and might be very large. Therefore, it is hard to compare the terms
> > > > > $\sum_{i>1}\frac{\Delta_i\log T }{\text{kl}(\mu_i,\mu_1)}$ and  $\sum_{i>1}8\sigma^2_{i,\star}\big(\text{kl}'(\mu_i,\mu_1)/\text{kl}(\mu_i,\mu_1) \big)^2$ in general. For the results in Theorem 2 of [1], we have added the discussion in our revised version (Related Work: Line 120-122), which only holds for bounded reward.
> > > > >
> > > > >
> > > > > We should clarify that Lai and Robbins 1985 ([26] in our paper) is for asymptotical setting and for regret lower bounds. Theorem 2 of [1] is for any distribution in $[0,1]$, which includes nonparametric distributions. Our paper focuses on the parametric distribution but may not be supported in $[0,1]$. Therefore, the settings in Theorem 2 of [1] and our paper are different. We would like to highlight that the main contribution of our paper is to prove a simultaneously asymptotically optimal, sub-UCB, and minimax optimal regret of Thompson sampling type algorithms, which is very different from the setting and the goal of [1].

---

> ### Author Response · Authors · 2022-08-02
> **Response to Reviewer XukM (Part 1)**
>
> Thank you for the helpful comments. Based on the comments, we have  thoroughly revised the paper, with major changes highlighted in ${\color{blue}{blue}}$. Before answering your questions, we would like to highlight our contributions, where you might have some misunderstandings.
> 1. We propose a variant of Thompson Sampling, called ExpTS. ExpTS simultaneously achieves asymptotic optimality, sub-UCB, and minimax optimality up to a factor $\sqrt{\log K}$ for one exponential family of reward distributions.
> 2. We propose another variant of Thompson Sampling, called ExpTS+, which is simultaneously minimax optimal and asymptotically optimal.
> 3. For original Thompson Sampling, we prove that Bernoulli-TS and Gaussian-TS have the worst-case regret $O(\sqrt{KT\log K})$. Similar to ExpTS+, we can add the greedy step to Gaussian-TS (or Bernoulli-TS), and the resulted algorithms Gaussian-TS+ (Bernoulli-TS+) will be minimax optimal and asymptotically optimal simultaneously.
>
> Now, we address your questions point by point.
>
> ---
> **Q1:** The algorithm modifies the posterior without addressing the corresponding anti-concentration of the original TS, which does not contribute to the understanding of the Thompson sampling itself.
>
> **A1:** We also studies the original TS, see contribution 3. Here, we want to highlight that proving Thompson Sampling without inflating the posterior distribution (will lose asymptotic optimality) achieving the worst-case regret $O(\sqrt{KT \log K})$ is an open problem. This is considered as one of the main contributions of our paper.
>
> ---
> **Q2:** Incorrect mathematical definition: finite-time (Landau notation is asymptotic).
>
> **A2:** Following the convention of the bandit/RL literature [27, 28], we say a regret bound is asymptotic if it only holds when $T$ approaches $+\infty$. For instance, the inequality in (1.1) is only true when we take the limit. A finite-time regret bound means that the derivation of it does not use any condition that $T$ goes to infinity, but instead treats $T$ as an arbitrarily fixed finite number. The Landau notation is only used in our paper to state the order of the regret and compare it with existing results.
>
>
> ---
> **Q3:** I do not think most (all?) of TS analysis does not have a sub-UCB bound in the sense of (1.2). For example, if there is a term that is $1/\Delta_i^2$, then it is not sub-UCB because it diverges faster than (1.2) when we fix T and takes a small Delta).
>
> **A3:** Some existing TS algorithms can achieve the sub-UCB regret, as clearly shown in our Table 1. In particular, TS in references [4] and [5] achieves the sub-UCB. Our main motivation is to design an algorithm that simultaneously achieves minimax optimality, asymptotic optimality, and sub-UCB, which is not achieved by any TS algorithm in the literature.
>
> ---
> **Q4:** The tradeoff relation is not very clear: the asymptotic rate on the top of $\log T$ is different in (1.1) and (1.2), which are not in a tradeoff relation.
>
> **A4:** There is no tradeoff for (1.1) and (1.2). As we mentioned in **A2**, the asymptotic regret and the sub-UCB (finite-time) regret are only for different settings in different problems. We prove that ExpTS simultaneously achieves asymptotic optimality (1.1) and sub-UCB  (1.2).
>
>
> ---
> **Q5:** Authors should mention whether the bound in [23] can be made finite-time or if there need a modification such as ExpTS that is targeted to improve the anti-concentration property.
>
> **A5:** The idea of [23] is totally different from ours. In [23], the constant that depends on the problem instance can not be explicitly derived. Since our regret decompositions are different, it is unclear whether it can be converted to a tight finite-time regret bound using our anti-concentration property.
>
>
> ---
> **Q6:** The computational aspect of the proposed algorithm is not mentioned (nor simulations), which limits the utility of the proposed algorithm.
>
> **A6:** Our work is mainly focused on the theoretical optimality of Thompson sampling type algorithms. It would be a very interesting future direction to investigate the empirical performance of ExpTS with different choices of sampling distributions.

---

> > ### Comment · Reviewer_XukM · 2022-08-04
> > **On A2, A3**
> >
> > Thank you for answering each part of the paper.
> >
> > > A2
> > > A finite-time regret bound means that the derivation of it does not use any condition that $T$ goes to infinity, but instead treats $T$ as an arbitrarily fixed finite number.
> >
> > We agree on this definition. However, Landau's notation is very different from this statement. $O(f(T))$ is "there exists $T_0$ such that all $T > T_0$, which describes nothing on finite $T < T_0$.
> >
> > > A3
> >
> > For example, [4] (arxiv: 1209.3353) has $1/\Delta^4$ term (p7 after "Here, order notation is hiding functions"...) and is not $O(\log T/\Delta + \Delta)$ (when we fix $T$ as constant and $\Delta \rightarrow +0$). Sorry if I misunderstood or checked an obsolete version.

---

> > > ### Author Response · Authors · 2022-08-07
> > > **Re: On A2, A3**
> > >
> > > Thank you for your further questions. We clarify them as follows.
> > >
> > > ---
> > > **Q:** However, Landau's notation is very different from this statement. $O(f(T))$ is "there exists $T_0$ such that all $T>T_0$, which describes nothing on finite $T<T_0$.
> > >
> > > **A:** We want to first clarify that we use "finite-time" to differentiate the setting from "asymptotic" following numerous bandit/RL papers in the literature. Specifically, the asymptotic regret bound $\lim_{T\rightarrow\infty}R_{\mu}(T)/\log T=\sum_{i>1}\frac{\Delta_i}{\text{kl}(\mu_i,\mu_1)}$ only holds for the case when $T\rightarrow\infty$. In contrast, our results hold for any $T$, which could be an arbitrarily fixed finite number. We will explain it by showing our proof as follows.
> > >
> > > All of our proofs for finite-time bounds explicitly show the constants in the regret bounds. Take the proof of the finite-time regret in Theorem 4.2 as an example, which is presented in Appendix A.1. The proof of Theorem 4.2 and all of its supporting lemma explicitly show the constant factors except for a few steps where we omit the constant and write it using the $O(\cdot)$ notation for simplicity. By considering all these omitted constants, we have the following exact result:
> > > $R_{\mu}(T)\leq C\sum_{i:\Delta>
> > > \lambda}\bigg(\Delta_i+\frac{V\log (T\Delta_i^2/V)}{\Delta_i} \bigg)+\max_{i:\Delta_i\leq\lambda}\Delta_i T,$
> > > where $\lambda\geq 16\sqrt{V/T}$ and $C$ is a constant that is independent of the problem parameters (indeed, a loose calculation shows that $C=2000$ suffices). Therefore, our result is indeed finite-time. We will remark this in our final version.
> > >
> > > ---
> > > **Q:** For example, [4] (arxiv: 1209.3353) has  term $1/(\Delta_i)^4$ (p7 after "Here, order notation is hiding functions"...) and is not  $O(\Delta+\log T/\Delta)$ (when we fix  as constant and ). Sorry if I misunderstood or checked an obsolete version.
> > >
> > >
> > > **A:** Thanks for pointing out this. The algorithm in [4] we cited is TS with Beta prior. In [4], the regret on page 7 indeed has a term $1/(\Delta_i)^4$. However, this is the intermediate step in their proof of the problem-dependent bound (Theorem 1), which is not sub-UCB. On the other hand, they also provide a problem-independent bound (Theorem 2), of which the proof presented on page 8 shows that the regret is $\mathbb{E}[R(T)]=O(\sum_{i\neq 1}\log T/\Delta_i)$, which is sub-UCB. Moreover, the algorithm we cited in [5] is TS with Gaussian prior (Theorem 1.3 in their paper).  Their paper does not have results showing that TS with Gaussian prior achieves the sub-UCB. However, their proof of Theorem 1.3 in their revised version (page 18) might imply the sub-UCB regret. We have revised our paper accordingly.

---

> ### Author Response · Authors · 2022-08-08
> **Any further questions?**
>
> Dear Reviewer, we would like to know whether our responses have addressed your concern. If you have any further questions, we are more than happy to clarify and address them. If you are satisfied with the current response and revised paper, we hope you can consider raising your score for our paper, which is very important for us. Thank you.

---

> > ### Comment · Reviewer_XukM · 2022-08-08
> > **Re: Any further questions?**
> >
> > Thank you for the reminder. Below are my responses to your reply. I will write my remaining concerns in what follows.
> >
> > > A2
> >
> > It is just wrong to use Landau notation and claim it as finite time-bound. I recommend you to use notations like [29] (1).
> >
> > > A3
> >
> > You are right, thanks!
> >
> > > A9
> >
> > I agree that the bound includes a 1/Delta^4 term.

---

> > > ### Author Response · Authors · 2022-08-09
> > > **Re: Any further questions?**
> > >
> > > Thanks for your suggestion on A2. We have revised our paper accordingly.

---

> > ### Comment · Reviewer_XukM · 2022-08-08
> > **Some of remaining concerns**
> >
> > Let me describe some of my major remaining concerns.
> >
> > * Sub-UCB as a characterization
> >
> > Honestly speaking, I did not get clear-cut reasoning on why this paper cares much about Sub-UCB criteria. I can understand the other characterizations -- asymptotic optimality, being finite-time, minimax ratio, and being anytime. The bound $C (\log T/\Delta + \Delta)$ - having possibly sub-optimal $\log T$ terms and a reasonable non-$T$ term seems not a good characterization. For example, being sub-UCB and being $(\log T)/(2 \Delta) + 1/\Delta^2$ (i.e. better leading constant on $\log T$ with possibly large $1/\Delta^2$ term), looks better than sub-UCB for me. Honestly speaking, many existing papers (such as [4,8]) do not care much about being sub-UCB.
> >
> > Related to this, does the minimax ratio and sub-UCB have a tradeoff relation? I mean, relation between ExpTS ($\sqrt{\log K}$ minimax with Sub-UCB) and ExpTS+ ($1$ minimax without Sub-UCB).
> >
> > * Anti-concentration (4.4)-(4.5), and (A1)
> >
> > It would be great if you elaborate on how these apply to true Thompson sampling. For example, can you derive (or state negatively) similar inequality on the posterior of the original 1-parameter exponential TS? This (and together with the lack of mentioning on [25]) is what I meant "does not contribute to the understanding of the Thompson sampling", but sorry if I am not very clear.
> >
> > * 1-parameter Exponential family
> >
> > It would be great if you elaborate what classes of one-parameter exponential family is (and is not) included in this paper. For example, the class of exponential family dealt with in this paper is wide/narrow/incompatible with existing papers, such as [25,26] and KL-UCB paper (AS, discussed here).

---

> > > ### Author Response · Authors · 2022-08-09
> > > **Re: Some of remaining concerns**
> > >
> > > Thank you for the helpful comments. Based on the comments, we have thoroughly revised the paper. We address your concerns point by point as follows.
> > >
> > > ---
> > > **Q1:** is Sub-UCB a good characterization?
> > >
> > > **A1:** We believe that sub-UCB is a complement metric for regret bound, and the reasons are as follows:
> > >
> > > 1. If we want a reasonable worst-case regret bound, then the finite-time regret cannot be much better than sub-UCB up to a $\log$ factor. (this argument could be found on page 2 in [29, Refining the confidence level for optimistic bandit strategies]).
> > >
> > > 2. KL-UCB$^{++}$ [32, A minimax and asymptotically optimal algorithm for
> > >  stochastic bandits] is asymptotically optimal and minimax optimal, but not sub-UCB. Moreover, there is an instance (page 8 of [29]) such that the regret of KL-UCB$^{++}$ is $K$ times larger than the regret of any algorithms that satisfy sub-UCB criteria. Therefore, sub-UCB is a fair complementary metric for the instance-dependent finite-time regret bound.
> > >
> > > 3. We agree that $\sum_{i>1}(\log T)/(2\Delta_i)+\sum_{i>1}1/\Delta_i^2$ is a reasonably good regret, since the constant $1/2$ is small and will have a good performance for the case with a large $\Delta_i$. However, for bandit instances with small $\Delta_i=\sqrt{K/T}$ for all $i>1$ (which is a common case used in the derivation of the minimax optimal regret),  the above regret reduces to the order of $\Theta(T)$, while the sub-UCB regret would be $O(\sqrt{KT\log T})$, which is much smaller.
> > >
> > > In any case, the sub-UCB criteria is only one of the goals for our algorithm to achieve. Our algorithm also aims to achieve the asymptotically optimal regret and the minimax optimal regret.
> > >
> > >
> > > ---
> > > **Q2:** Tradeoff relation between  ExpTS and ExpTS$^+$.
> > >
> > > **A2:** For ExpTS and ExpTS$^{+}$, there is a tradeoff between minimax optimal regret and sub-UCB regret. We have mentioned in our conclusion "it would be an interesting future direction to design a sampling distribution such that TS is asymptotically optimal, minimax optimal, and matches the sub-UCB criteria at the same time".
> > >
> > > ---
> > > **Q3:** Anti-concentration (4.4)-(4.5). Can you derive (or state negatively) similar inequality on the posterior of the original 1-parameter exponential TS? This (and together with the lack of mentioning on [25]) is what I meant "does not contribute to the understanding of the Thompson sampling", but sorry if I am not very clear.
> > >
> > >  **A3:**  Deriving similar inequality on the posterior of the original 1-parameter exponential TS is a new problem. We are not aware of any anti-concentration bounds for the posteriors of 1-parameter exponential family of reward distributions.
> > >
> > > We would like to clarify that our results also contribute to the understanding of Thompson Sampling. We show that Thompson Sampling without inflating the posterior distribution (will lose asymptotic optimality) achieves the worst-case regret $O(\sqrt{KT \log K})$ and maintains the asymptotic optimality. The previous work [5] for original Thompson Sampling achieves the worst-case regret bound $O(\sqrt{KT \log K})$ but sacrifices the asymptotic optimality. This is considered as one of the main contributions of our paper.
> > >
> > > ---
> > > **Q4:** 1-parameter Exponential family,  It would be great if you elaborate what classes of one-parameter exponential family is (and is not) included in this paper. For example, the class of exponential family dealt with in this paper is wide/narrow/incompatible with existing papers, such as [25,26] and KL-UCB paper (AS, discussed here).
> > >
> > > **A4:** The definition of 1-parameter exponential family in our paper is $p_{\theta}(x)=\exp(x\theta-b(\theta)+c(x))$, which is the same as  the definition in [15,17,19,32] as well as AS [12]. The 1-parameter exponential family considered in [25] ($p_{\theta}(x)=\exp(T(x)\theta-b(\theta)+c(x))$) is more general than that in the aforementioned papers (see page 4 in [25]). [26] considers parametric distributions that satisfy some mild conditions, which is also more general than ours. Moreover, [12] also considered the reward distribution supported in [0,1], which is not compatible to ours.
> > >
> > > We have added the above discussions on Line 180-186 in our paper.

---

### Official Review · Reviewer_mKp1 · 2022-07-10

**Rating:** 8
**Confidence:** 4
**Soundness:** 4 excellent
**Presentation:** 4 excellent
**Contribution:** 4 excellent

**Summary:**

This work studies the regret of Thompson sampling (TS) for exponential family reward distributed multi-armed bandits, by addressing all its theoretical properties.

After a careful summary of their work, an informative related work Section 2 and necessary preliminaries in Section 3, the authors argue in Section 4.1. that the existing finite-time regret TS analysis based on the under-estimation of the optimal arm can be improved by introducing a lower confidence bound on such under-estimation.

To that end, they present a new decomposition of the regret that allows for a new approach to algorithm analysis and design. Specifically, they propose in Section 4.2 a sampling distribution that results in a variant of TS (denoted Exp-TS), with its subsequent frequentist regret analysis provided in Section 4.3.

The analysis simultaneously yields both the finite-time regret bound and the asymptotic regret bound: they prove that Exp-TS is the first algorithm to achieve the sub-UCB criteria, asymptotic optimality, and minimax optimality simultaneously for exponential family reward distributions.

In addition, they are able to show that when the exponential reward distribution has a closed form (e.g., Gaussian and Bernoulli), the posteriors in standard Thompson sampling relate to the proposed sampling distribution and therefore, obey asymptotic and finite-time regrets as well.

Finally, they introduce in Section 5 a new sampling distribution that adds a greedy exploration step to the sampling distribution used in ExpTS to remove the extra logarithm term in the worst-case regret of ExpTS, and show that this new algorithm ExpTS$^+$ is minimax optimal. To do so, a procedure to mitigate the over-estimation of sub-optimal arms is proposed.

**Questions:**

The authors claim that the proposed sampling distribution in Equation 4.3 "_seems complicated for a general exponential family reward distribution, even though they only need the sampling distribution to satisfy a nice tail bound derived from this reward distribution._" Given that they show in Section 4.4. that usual conjugate priors in Gaussian and Bernoulli distributed rewards do meet such requirements, could the authors discuss what priors (or which properties of the priors) on other exponential-family rewards would meet the necessary distribution requirements?

**Limitations:**

Given the theoretical nature of this work, there is minimal immediate impact on society, and limitations are somehow implicit in the modeling and theoretical assumptions.

**Strengths And Weaknesses:**

This work is very relevant as it studies a rich family of bandit reward distributions, i.e., the exponential family of distributions that covers commonly used ones (Bernoulli, Gaussian, Gamma, and Exponential).

The novelty and significance of this work's contributions are:

- A proof that ExpTS is an algorithm that, for exponential family of reward distributions, achieves the sub-UCB criteria, and is simultaneously minimax optimal (up to a factor of $\sqrt{log K})$ and asymptotically optimal. In addition, they show how a greedy extension of ExpTS is actually able to remove the extra $\sqrt{log K}$ minimax optimality factor.

- A novel proof that both TS with Gaussian rewards and Gaussian posteriors, and Bernoulli rewards with Beta posteriors, are asymptotically optimal and minimax optimal up to a factor of $\sqrt{log K}$

- A novel, yet conceptually simple set of techniques, to analyze not only the frequentist regret of existing algorithms, but to also enable the proposal and analysis of new algorithms. Specifically, a regret decomposition that is conditioned on a lower confidence bound introduced in Equation 4.2

---

> ### Author Response · Authors · 2022-08-02
> **Response to Reviewer mKp1**
>
> Thank you for your encouraging and supportive comments. We address your question as follows.
>
> ---
> **Q1:** The authors claim that the proposed sampling distribution in Equation 4.3 "seems complicated for a general exponential family reward distribution, even though they only need the sampling distribution to satisfy a nice tail bound derived from this reward distribution." Given that they show in Section 4.4. that usual conjugate priors in Gaussian and Bernoulli distributed rewards do meet such requirements, could the authors discuss what priors (or which properties of the priors) on other exponential-family rewards would meet the necessary distribution requirements?
>
> **A1:** This is a very interesting problem. While the concentration bounds are already obtained for all exponential-family rewards, we are not aware of any tight anti-concentration bounds for other special exponential-family distributions. We believe that if the anti-concentration and concentration bounds are tight for other conjugate priors, then our techniques can be applied there and lead to a tighter finite-time regret bound. It is an important and open problem to prove a tight anti-concentration bound for general one-exponential family of distributions.

---

> > ### Comment · Reviewer_mKp1 · 2022-08-04
> > **Thank you for your informative responses**
> >
> > I thank the authors for taking the time to carefully respond to my/our comments/questions.
> >
> > As such, I would encourage the authors to incorporate their response and discussion to question Q1 (also related to Q7 by Reviewer XukM) on why there is a need for tight anti-concentration bounds for general one-exponential family of distributions to extend the presented algorithm and results beyond the cases studied in this work, which also helps pinpoint its significance.

---

> > > ### Author Response · Authors · 2022-08-07
> > > **Re: Thank you for your informative responses**
> > >
> > > Thank you for your further reply and question. In our regret decomposition Eq. (4.2), we need to upper bound the following term:
> > > $\Delta_i\sum_{s=1}^T E_{\hat\mu_{1s}}$
> > >  $\bigg[\bigg(\frac{1}{G_{1s}(\epsilon)}-1\bigg)\cdot \mathbb{1}(\hat\mu_{1s}\geq Low_s)  \bigg]  \quad\qquad (*)$.
> > >
> > > There are two parts that incur errors for bounding $(*)$.
> > >
> > > First, $1/G_{1s}(\epsilon)$ denotes the number of samples we need to sample from the sampling distribution such that at least one sample is larger than $\mu_1-\epsilon$. When $\hat\mu_{1s}\leq \mu_1-\epsilon$, we need an anti-concentration to lower bound $\mathbb{P}(\theta_{1}(t)\geq \mu_1-\epsilon)=G_{1s}(\epsilon)$.
> > >
> > > Second, note that in Equation $(*)$, the expectation is taken over the randomness of $\hat\mu_{1s}$. However, for general one exponential family distributions, the closed form of the distribution of $\hat\mu_{1s}$ is hard to obtain. In our paper, we instead use the concentration bound to approximate the distribution of $\hat\mu_{1s}$ (refer to Lemma A.5), which will incur additional error for lower bounding $G_{1s}(\epsilon)$.
> > >
> > > Therefore, to account for these extra errors in bounding (*), we need a tight anti-concentration bound.

---

### Official Review · Reviewer_vJHc · 2022-07-14

**Rating:** 7
**Confidence:** 4
**Soundness:** 3 good
**Presentation:** 4 excellent
**Contribution:** 3 good

**Summary:**

The paper proposes a new Thompson sampling algorithm by designing proper sampling distributions for exponential family reward distributions. Three contributions: (1) new algorithm (2) showing that the standard TS enjoys a $\sqrt{\log(K)}$ minimax ratio (without inflating the posterior distribution) and (3) a simple greedy trick to achieve a constant minimax ratio.

**Questions:**

* L168: missing a square for (mu-mu')
* L169-170: I did not follow why we are setting V(.) = 1/4. I mean, we don't get to choose V(.); it is given from the definition of a specific exponential family distribution?
* L209-210: I would remove the word 'a regret' there since it sounds like it is about the regret lower bound rather than the lower bound on $D_i$.
  * Furthermore, looking into Lemma 5 of Jin et al., it seems that D_i = \Omega(\sqrt{KT\log(T)}) is not correct (and rather a typo). It seems we need to say K D_i = \Omega(\sqrt{KT\log(T)}) ?
* It'd be great to have a figure showing plots of PDFs for the sample distribution.
* what's the role of b_n? Anything goes wrong if we set $b_n=1$
* How does the empirical performance of ExpTS fair with TS?
* L254: "when n is large, we will choose b_n to be close to 0" ⇒ this contradicts to b_n in Theorem 4.1?
* L276-277: to claim that ExpTS is comparable to the best known UCB, we need to define the scope. For Gaussian rewards, AdaUCB has a strictly better guarantee than ExpTS.

**Strengths And Weaknesses:**

Originality: high

Quality: high

Clarity: high

Significant: medium-high

The key contribution, to me, is the novel analysis described around Eq 4.2. In fact, proving/disproving that the standard Thompson sampling (without inflating the posterior distribution, which breaks the asymptotic optimality) has the minimax rate of $O(\sqrt{\log(K)})$ was an open problem. Note Agrawal & Goyal (2017) have shown this rate with Gaussian TS but they in fact inflate the posterior variance by a constant factor.

On clarity, I appreciate that the authors point out specific section/formula numbers from the references; not many people do this these days.

While the sampling distribution is novel, I am not entirely convinced that it is necessary, but at least it is a nice addition to the library of variations of TS.

---

> ### Author Response · Authors · 2022-08-02
> **Response to Reviewer vJHc**
>
> Thank you for the supportive and helpful comments. Based on the comments, we have  thoroughly revised the paper, with major changes highlighted in ${\color{blue}{blue}}$. We address your questions point by point as follows.
>
> ---
> **Q1:** L168: missing a square for $(\mu-\mu')$
>
> **A1:** We have fixed it in the revision.
>
> ---
> **Q2:** L169-170: I did not follow why we are setting $V(.) = 1/4$. I mean, we don't get to choose $V(.)$; it is given from the definition of a specific exponential family distribution?
>
> **A2:** $V$ is the upper bound of the variances of the reward distributions. For a Bernoulli distribution with parameter $p$, its variance is $p(1-p)$. Then，$p(1-p)\leq 1/4$ and achieves its maximum at $p=1/2$. Hence, we choose $V=1/4$ as the upper bound for the variance.
>
> ---
> **Q3:** L209-210: I would remove the word 'a regret' there since it sounds like it is about the regret lower bound rather than the lower bound on $D_i$.
>
> **A3:** We agree. We remove the word 'a regret' and claim "it is a lower bound of $D_i$" in the revision.
>
> ---
> **Q4:** Furthermore, looking into Lemma 5 of Jin et al., it seems that $D_i = \Omega(\sqrt{KT\log(T)})$ is not correct (and rather a typo). It seems we need to say $K D_i = \Omega(\sqrt{KT\log(T)})$ ?
>
> **A4:** Thanks for pointing out the typo. We have fixed it in the revision.
>
> ---
> **Q5:** It'd be great to have a figure showing plots of PDFs for the sample distribution.
>
> **A5:** The sampling distribution $\mathcal{P}(\mu,n)$ is adaptive to parameters $\mu$, $n$ and $\text{kl}(\cdot)$. Therefore, it is hard to plot the PDFs of a general distribution $\mathcal{P}$. For Gaussian rewards, the sampling distribution is similar to Rayleigh distribution. To explain, for Gaussian reward distribution with variance $V$, we have
> \begin{align*}
> \mathcal{P}(\mu,n)=\frac{nb_n|x-\mu|}{2V}e^{-nb_n(u-x)^2/(2V)}.
> \end{align*}
> The PDF of Rayleigh distribution with parameter $\sqrt{V/(nb_n)}$ is $$\frac{nb_nx}{V}e^{-nb_nx^2/(2V)}, \quad x>0.$$
> Hence, in this case the sampling distribution $\mathcal{P}$ is a Rayleigh distribution if it goes along the $x$ axis by a distance of $\mu$ and then is restricted to $x \geq 0$.
>
> ---
> **Q6:** What's the role of $b_n$? Anything goes wrong if we set $b_n=1$.
>
> **A6:** For the finite-time regret, there is no problem with $b_n=1$. The finite-time regret bounds ((4.6) and (4.7)) also hold for $b_n=1$.
>
> For the asymptotic regret, it might not be optimal. The reason is as follows. If $b_n=1$,  Lemma A.2 used for the finite-time analysis shows that the regret has a term $O(V(\log (T\epsilon^2/V)/\epsilon^2)$. However, this is not asymptotically optimal. To solve this problem, we introduce the parameter $b_n$ with $b_n<1$. With the help of $b_n$, we decrease the term $O(C_1\log T)$ in Lemma A.2 to the term $O(C_2(\log \log T)^6)$  in Lemma B.1, where $C_1$, $C_2$ are constants independent of $T$. Then, $O(C(\log \log T)^6)$ will be dominated by $\log T$ and disappear in the asymptotic regret.
>
> For Gaussian-TS and Bernoulli-TS, we do not need $b_n$.
>
> Remark: $b_n$ plays a similar role as the method used in [5] (Near-Optimal Regret Bounds for Thompson Sampling) and [17] (MOTS: Minimax Optimal Thompson Sampling), which inflates the variance of the sampling distribution. However, they are for different purposes. Inflating sampling distribution in [5] and [17] is for proving the finite-time bound, while ours is for the asymptotic regret bound. Besides, inflating the sampling distribution in [5] and [17] loses the asymptotic optimality. In our paper, the use of $b_n$ does not break the asymptotic optimality  when $n$ approaches infinity, $b_n=1$.
>
> ---
> **Q7:** How does the empirical performance of ExpTS fair with TS?
>
> **A7:** Our work is mainly focused on the theoretical optimality of Thompson sampling type algorithms. It would be a very interesting future direction to investigate the empirical performance of ExpTS with different choices of sampling distributions.
>
> ---
> **Q8:** L254: "when n is large, we will choose $b_n$ to be close to 0" ⇒ this contradicts to b_n in Theorem 4.1?
>
> **A8:** This is a typo. It should be "When $n$ is larger, we will choose $b_n$ to be close to $1$." We have fixed it in the revision.
>
> ---
> **Q9:** L276-277: to claim that ExpTS is comparable to the best-known UCB, we need to define the scope. For Gaussian rewards, AdaUCB has a strictly better guarantee than ExpTS.
>
> **A9:** We agree that we need to add a specific scope to compare our algorithm to UCB type algorithms. We rephrase it as "Compared with state-of-the-art MAB algorithms listed in Table 2, ExpTS is comparable to the best known UCB algorithms that work for exponential family of reward distributions".

---

> > ### Comment · Reviewer_vJHc · 2022-08-08
> > **thanks for your response.**
> >
> > I appreciate your detailed response.

---

### Comment · Area_Chair_c3kB · 2022-08-09
**Question from AC**

Since the evaluation of this paper is quite diverse, I briefly read the paper and I came up with one question (sorry if it is already discussed somewhere and I missed it). This paper and author-reviewer discussions stress that the result holds for general exponential families as far as it has a bounded variance.
Does this mean that the variance must be uniformly bounded by $V$ for all distributions in the model? If so, I don't have any idea of examples of an exponential family with this property except Bernoulli and Gaussian and at least gamma and exponential distributions are not covered. Are there other practical examples of this property? Or we have to consider the exponential family with a bounded parameter space?

---

> ### Author Response · Authors · 2022-08-10
> **Re: Question from AC**
>
> Thank you for reading our paper and for your comments.
>
> For proving the asymptotic optimality, a bounded variance is not necessary. For proving the worst-case regret, all existing references [4,5,6,7,20,29,32] require the variance to be bounded. In specific, we follow the setting (bounded variance $V$) in [32], which also studies the worst-case regret for exponential family bandits.
>
> The reason is as follows. The minimax optimal regret would become meaningless if we allow variance $=\infty$, since it is defined for the worst-case bandit instance. If we allow the variance to be unbounded, then there exists a worst-case instance, where the variance of at least one arm is $\infty$. In this case, we cannot determine whether this arm is the best arm within finite-time steps $T$, which therefore leads to a linear regret $O(T)$.
>
> For exponential distribution and Gamma distribution, it suffices to assume the parameter space is bounded. In particular, for an exponential distribution with parameter $\lambda$, its variance is $1/\lambda^2$. For Gamma distribution $\Gamma(k,\lambda)$ with parameter $\lambda$ (fixed $k$), its variance is $k\lambda^2$. And thus assuming the parameter space is bounded is enough to have bouned variances.

---

> > ### Comment · Reviewer_XukM · 2022-08-10
> > **Re: Question from AC**
> >
> > > we cannot determine whether this arm is the best arm within finite-time steps
> >
> > In many of the exponential family distributions (such as normal and exponential), having a higher variance makes distinguishing two arms easier (larger KL divergence).

---

> > > ### Author Response · Authors · 2022-08-10
> > > **Re: Question from AC**
> > >
> > > > In many of the exponential family distributions (such as normal and exponential), having a higher variance makes distinguishing two arms easier (larger KL divergence).
> > >
> > > This is false. The KL divergence for two Gaussian distribution $\mathcal{N}(\mu,V)$ and $\mathcal{N}(\mu',V)$ is $\text{kl}(\mu,\mu')=\frac{(\mu-\mu')^2}{2V}$. Therefore, a higher variance makes the KL divergence smaller.

---

> > > > ### Comment · Reviewer_XukM · 2022-08-10
> > > > **Re: Question from AC**
> > > >
> > > > > The KL divergence for two Gaussian distribution $\mathcal{N}(\mu, V)$ and $\mathcal{N}(\mu', V)$.
> > > >
> > > > Thanks for the correction. Yeah, in the case of equal variance normal, a higher variance makes KL divergence smaller. How about the case of exponential distribution?

---

> > ### Comment · Area_Chair_c3kB · 2022-08-10
> > **thank you for the response**
> >
> > Thank you for the clarification.
> > I understood that the authors' standpoint that the bounded variance is justified through the existing work, and exponential and gamma distributions can also be covered by restriction of the parameter space.
> > (I refrain from writing my own opinion here for neutrality as an AC.)

---

### Meta-Review · Area_Chair_c3kB · 2022-08-24

**Recommendation:** Accept
**Confidence:** Less certain

**Metareview:**

This paper provides a good contribution on Thompson sampling for exponential families, while several concerns on the presentation and technical points are raised. After the discussions with reviewers and my own reading of the paper, I judged that the issues on the presentation is not so much serious and considering the strong results of the paper I determined to recommend acceptance. Still, I partly agree with the opinion that technical novelty is not enough or well presented, and I strongly encourage the authors to seriously address the raised concerns in the final version.

Following is my own comment to the paper.
- In the discussion phase the authors explained that the algorithm can also consider the exponential families other than Bernoulli or Gaussian distributions by restricting the parameter space. Though it is supported through existing work, some problems become significantly easy when we are allowed to consider a compact parameter space in my experience. Though the boundedness of the space is not explicitly used, I believe that this limitation on the model must be at least explicitly clarified (or practical examples of models without restriction other than Bernoullis or Gaussians should be given). In practice we would not know the exact space and it would make another problem of whether we can take the space conservatively. The response on the necessity of bounded $V$ is not convincing to me and at least not formally explained.
- One of the biggest reasons of success of TS would be its easy implementation. In this viewpoints the proposed algorithm does not seem to be practical and the original motivation of using TS seems to be somewhat weakened. In particular, in the exponential families KL-based algorithms are easily implemented. Though this kind of algorithms sometimes requires to solve the inverse of KL, the instance to be solved converges rapidly and the actual number of iterations for computing the inverse becomes very small. On the other hand, the proposed algorithm requires to solve randomly sampled instance and iterations at each round seems to become considerably large. It is fine so far if the TS-based algorithm is essentially necessary to achieve bounds of the paper, but it does not seem to be explained well and I expect that the motivation of using the TS-based algorithm despite its computational burden is clarified (other than its technical interestingness).

**Award:**

No

---

### Decision · Program_Chairs · 2022-09-14

Accept